# Provable Policy Optimization for Reinforcement Learning from Trajectory Preferences with an Unknown Link Function

## Abstract

The link function, which characterizes the relationship between the preference for two trajectories and their cumulative rewards, is a crucial component in designing RL algorithms that learn from preference feedback. Most existing methods, both theoretical and empirical, assume that the link function is known (often a logistic function based on the Bradley-Terry model), which is arguably restrictive given the complex nature of preferences, especially those of humans. To avoid mis-specification, this paper studies preference-based RL with an unknown link function and proposes a novel zeroth-order policy optimization algorithm called `ZSPO`. Unlike typical zeroth-order methods, which rely on the known link function to estimate the value function differences and form an accurate gradient estimator, `ZSPO` only estimates the *sign* of the value function difference. It then constructs a parameter update direction that is *positively correlated* with the true policy gradient, eliminating the need to know the link function exactly. Under mild conditions, `ZSPO` provably converges to a stationary policy with a polynomial rate in the number of policy iterations and trajectories per iteration. Empirical evaluations further demonstrate the robustness of `ZSPO` under link function mis-specifications.

## 1 Introduction

Reinforcement learning (RL), as a paradigm for online sequential learning via interactions with the environment (Sutton et al., 1998), has achieved success in many fields (Kohavi et al., 2009; Christiano et al., 2017; Kiran et al., 2022; Ouyang et al., 2022). An RL problem is typically formulated as an episodic stochastic Markov decision process (MDP), where at each step, the agent observes the current state, takes an action, and then receives a numerical reward to reflect the action's quality (Bellman, 1958; Puterman, 2014). The desired behavior featuring the optimal policy is learned by maximizing the return (cumulative reward) of episodes. It is believed that for each RL problem, a *true* reward function exists, but often not known. Learning the reward, also known as the inverse reinforcement learning (IRL) problem, is extremely non-trivial (Ng & Russell, 2000). In reality, a hand-crafted reward *proxy* is designed by domain experts in the hope that learning from the proxy would induce the same behavior as learning from the true reward function (Hadfield-Menell et al., 2017; Kwon et al., 2023). However, this is often not the case where the reward proxy is likely to induce undesirable agent behaviors known as reward hacking (Skalse et al., 2022). In general, it is difficult to design a good reward proxy for complex RL environments that is both goal-achieving and easy to optimize.

**RL from Preference.** In recent years, reinforcement learning from human feedback (RLHF) has been proposed to avoid reward proxy design in many areas (Kaufmann et al., 2023), where the easiest to collect and most commonly used form of human feedback is the preference over a pair of trajectories. In these settings, the agent will not receive the numerical reward. Instead, it regularly queries preferences on pairs of trajectories from a noisy comparison oracle and then uses preferences to infer the quality of the policy. Two main approaches have been studied for policy optimization from preferences: (i) reward inference and (ii) direct policy optimization. The first approach (Christiano et al., 2017) learns an intermediate reward model compatible with the preferences to approximate the true reward function through a maximum likelihood loss, and then optimizes the learned reward via standard RL algorithms such as `PPO` (Schulman et al., 2017). The quality of the learned policy heavily depends on the quality of the reward model, which usually suffers from insufficient state-action

coverage, overfitting, and poor evaluation without the ground truth (Casper et al., 2023). The second approach (Rafailov et al., 2023) avoids these drawbacks by directly optimizing the policy from preferences, which delivered promising results both theoretically and empirically.

**Link Function.** Most algorithms assume the preference is generated via a known process. For example, the Bradley-Terry model (Bradley & Terry, 1952) widely used in the literature assumes the probability that a trajectory $\tau_1$ is preferred over $\tau_0$ is a logistic function of the return difference:

$$\mathbb{P}(\tau_1 \succ \tau_0) = (1 + \exp(-\gamma[r(\tau_1) - r(\tau_0)]))^{-1}, \tag{1}$$

where $r(\tau_1)$ and $r(\tau_0)$ are returns of the two trajectories and $\gamma$ is a known constant representing the expertise. Importantly, the maximum likelihood loss for reward model learning and the loss for policy optimization algorithms such as `DPO` (Rafailov et al., 2023) are both constructed explicitly using the Bradley-Terry relation. One may replace the logistic function with another admissible function $\sigma(\cdot) : \mathbb{R} \mapsto [0, 1]$, referred to as the *link* function, but the expression needs to be known. Moreover, these preference-based methods like `DPO` often require an explicit expression between reward and the optimal policy, a special structure only valid for bandits or deterministic MDPs, limiting their applicability to general RL problems despite their success in some sub-fields. For general MDPs and link functions, a recent work (Zhang & Ying, 2025) shows the effectiveness of a zeroth-order approach called `ZPG` from preferences for general MDPs and link functions, which again assumes the link function is known to estimate the trajectory reward difference.

Given the complex nature of preferences, such as variability in time and population, which inspired the rich literature on sensory, social choice, and behavior (Azari et al., 2012; Greene, 2010; Lawless & Heymann, 2010; Meilgaard et al., 1999), it is not adequate to characterize the preference with a single known function. Most preference-based RL methods suffer from preference mis-specification, similar to classic RL suffering from reward mis-specification. Therefore, it remains an open question:

> Without knowing the link function exactly, can we still design provable policy optimization algorithms to learn from preferences that work for general RL problems?

Our paper answers this question. We relax the requirement only to assume that the preference is positively related to the rewards, while the exact formula is unknown and subject to variability. Inspired by `ZPG`, this paper proposes a new zeroth-order policy optimization method. The key idea is to estimate the *sign* of the value function difference instead of the exact value function difference from preferences over trajectories, for which the link function is not needed. The algorithm then uses the sign to identify a direction for policy improvement with a provable convergence guarantee.

**Related Works.** The study of preference learning without a known link function has a long history, where dueling bandit (Bengs et al., 2021) and heuristic evolution algorithms (Busa-Fekete et al., 2014) are among the early attempts. However, most of these methods are often suitable only in tabular problems or are inefficient without theoretical guarantees. Most recent works (Christiano et al., 2017) overlooked the role of the preference model and the link function. Some Works (Munos et al., 2024; Azar et al., 2024) also studied policy optimization with an unknown link function from the viewpoint of a dynamic game, assuming no relation between preferences and rewards. Consequently, the learned policy is usually quite pessimistic and is far from the optimal policy due to the limitations of preferences (Knox et al., 2024; Zhang et al., 2024a). Our work is most related to (Zhang & Ying, 2025), where our proposed `ZSPO` improves over their `ZPG` to eliminate the need for the link function.

## 1.1 OUR CONTRIBUTIONS

We propose *Zeroth-Order Sign Policy Optimization* (`ZSPO`) from trajectory preference. Under mild conditions and the correct choice of the hyperparameters, `ZSPO` enjoys the following convergence rate (in terms of the gradient norm) to a stationary policy:

$$\sqrt{d} \cdot \widetilde{\mathcal{O}}\left(\sqrt{\frac{H}{T}} + \frac{1}{\min\{\sigma'(0), 1\} N^{1/4}} + \sqrt{\varepsilon_D^*}\right),$$

where $T$ is the number of policy iterations, $H$ is the episode length, $N$ is the number of trajectory batches for comparison in each iteration. $\sigma'(0)$ is the derivative of the link function at the origin, which characterizes the expertise of the preference oracle. As the oracle has more expertise, $\sigma'(0)$ becomes larger to constitute a faster convergence rate. $\varepsilon_D^*$ captures the distinguishability by the preference

oracle when the batch size is $D$. The first term matches the rate for classic zeroth-order methods. The second term captures the error of estimating the expected preference from finite evaluators. The third term characterizes the hardness to infer the value function sign through trajectory preferences, which decreases as the batch size $D$ increases. To the best of our knowledge, in utility-based preference models (Bengs et al., 2021; Wang et al., 2023) for general MDPs with an unknown link function, `ZSPO` is the *first* policy optimization algorithm with a provable convergence guarantee.

**Remark.** The focus of this paper is to answer the fundamental question associated with the unknown link function in theory and develop provable algorithms targeting broader preference-based RL, i.e., with stochastic transitions, and not tied to LLMs. The empirical validation of their practicality in these specific domains will be our future work, which we also briefly discuss in this paper.

## 2 PRELIMINARIES

We first introduce the preliminaries of the problem. For a scalar $a$, $\text{sign}[a]$ denotes its sign. For two vectors $\boldsymbol{a}$ and $\boldsymbol{b}$, $\langle \boldsymbol{a}, \boldsymbol{b} \rangle$ denotes the inner product. $\mathbb{E}_{\text{x}}[\cdot]$ denotes the expectation taken over x.

**Episodic RL:** We consider an episodic MDP instance $\mathcal{M} = (\mathbb{S}, \mathbb{A}, H, \boldsymbol{P}, \boldsymbol{\mu}_0)$, where $\mathbb{S}$ is the state space and $\mathbb{A}$ is the action space (both may be uncountable). $H$ is the planning horizon, $\boldsymbol{P} = \{\boldsymbol{P}_h\}_{h=1}^H$ is the set of transition kernels, and $\boldsymbol{\mu}_0$ is the initial distribution of states. The agent interacts with the environment in episodes. At the beginning of each episode, the agent chooses a policy $\pi$, a set of functions $\{\pi_h : \mathbb{S} \to \mathcal{P}(\mathbb{A})\}_{h=1}^H$, where $\mathcal{P}(\mathbb{A})$ denotes the set of all probability distributions over $\mathbb{A}$. Then, nature samples an initial state $s_1$ from the initial distribution $\boldsymbol{\mu}_0$. At each step $h$, the agent takes an action $a_h$ sampled from the distribution $\pi_h(s_h)$ after observing the state $s_h$. The environment consequently moves to a new state $s_{h+1}$ sampled from the distribution $\boldsymbol{P}_h(\cdot|s_h, a_h)$ without revealing any reward feedback. We use $\tau = \{(s_h, a_h)\}_{h=1}^H$ to denote a trajectory. We assume the expected return of $\tau$ is a function $r(\tau)$ which maps any trajectory to a value in $[0, H]$, which is more general than classic MDPs where the return is the sum of per-step rewards. For any given policy $\pi$, we define the value function $V_1^\pi(s)$ as the expected return of trajectories starting from $s$ and using policy $\pi$:

$$V_1^\pi(s) = \mathbb{E}_\pi\left[r(\tau) \,|\, s_1 = s\right] = \mathbb{E}\left[r(\tau) \,|\, s_1 = s, \{a_1, \cdots, a_H\} \sim \pi\right].$$

We define the expected value function over the initial state distribution $\boldsymbol{\mu}_0$ as $V(\pi) = \mathbb{E}_{s \sim \boldsymbol{\mu}_0}[V_1^\pi(s)]$.

**Policy Parameterization.** The agent's policy is parameterized by a policy network as a function class $\mathcal{N} = \{\pi_{\boldsymbol{\theta}} : \mathbb{S} \times [H] \to \mathcal{P}(\mathbb{A}) | \boldsymbol{\theta} \in \mathbb{R}^d\}$, which takes a state $s$ and a decision-making step $h$ as input and then outputs the probability distribution of the next action. Here $\boldsymbol{\theta}$ is the parameter of the policy network. Each parameter $\boldsymbol{\theta}$ induces a policy which we abuse the notation to denote as $\pi_{\boldsymbol{\theta}}$.

**Preference Feedback.** The agent has access to a preference oracle, for example, a black-box mechanism, a human evaluator, or a language model. In each episode, the agent can choose two batches of trajectories $\mathcal{D}_0 = \{\tau_{0,i}\}_{i=1}^D$ and $\mathcal{D}_1 = \{\tau_{1,i}\}_{i=1}^D$ to query the oracle to obtain a one-bit feedback $o \in \{0, 1\}$, where $D$ is the batch size. If $o = 1$, the oracle prefers $\mathcal{D}_1$, and if $o = 0$, the oracle prefers $\mathcal{D}_0$. The feedback $o$ is generated according to a preference model characterized by an *unknown* link function $\sigma : \mathbb{R} \to [0, 1]$ of the average reward difference between trajectories:

$$\mathbb{P}(\mathcal{D}_1 \succ \mathcal{D}_0) = \sigma(\bar{r}(\mathcal{D}_1) - \bar{r}(\mathcal{D}_0)) = \sigma\left(\frac{1}{D}\sum_{i=1}^D r(\tau_{i,1}) - \frac{1}{D}\sum_{i=1}^D r(\tau_{i,0})\right), \tag{2}$$

where $\bar{r}(\cdot)$ denotes the average return of a batch. If $\sigma(\cdot)$ is a logistic function, it becomes the Bradley-Terry model. In reality, most preference oracles like humans may not accurately aggregate the returns of a large batch, so a smaller $D$ is preferred. Generally, for reasonable and learnable link functions, we expect the preference probability to be positively correlated with the average reward difference, so we assume the following fundamental regularity of the link function, which is commonly noted in dueling bandits (Bengs et al., 2021) and preference-based RL (Wang et al., 2023).

**Assumption 1** *The link function* $\sigma : [-H, H] \to [0, 1]$ *is strictly monotonically increasing with* $\sigma(0) = 1/2$ *and* $\sigma(-x) = 1 - \sigma(x)$.

We aim to design a policy-optimization algorithm from preferences to find a parameter $\boldsymbol{\theta} \in \mathbb{R}^d$ that maximizes the value function, i.e., $\max_{\boldsymbol{\theta} \in \mathbb{R}^d} V(\pi_{\boldsymbol{\theta}})$.

---

**Algorithm 1** Zeroth-Order Sign Policy Optimization from Trajectory Preference

---

**Require:** initialize the actor-network parameter $\boldsymbol{\theta}_1$, learning rate $\{\alpha_t\}_{t=1}^T$, perturbation distance $\{\mu_t\}_{t=1}^T$, size of trajectory batches $D$;

1: **for** iteration $t = 1 : T$ **do**
2:     sample a random vector $\boldsymbol{v}_t$ from a normal distribution $\mathcal{N}(\boldsymbol{0}, \boldsymbol{I}_d)$;
3:     obtain perturbed parameter $\boldsymbol{\theta}_t' = \boldsymbol{\theta}_t + \mu_t \boldsymbol{v}_t$;
4:     **for** $n = 1 : N$ **do**
5:         sample a batch of $D$ trajectories $\mathcal{D}_{n,0} \sim \pi_{\boldsymbol{\theta}_t}$;
6:         sample a batch of $D$ trajectories $\mathcal{D}_{n,1} \sim \pi_{\boldsymbol{\theta}_t'}$;
7:         query the preference oracle with the two batches $(\mathcal{D}_{n,1}, \mathcal{D}_{n,0})$ and obtain results $o_{t,n}$;
8:     estimate the ascent direction with a majority vote as follows:

$$\hat{\boldsymbol{g}}_t = \text{sign}\left[\sum_{n=1}^N \left(o_{t,n} - \frac{1}{2}\right)\right]\boldsymbol{v}_t;$$

9:     update the actor network $\boldsymbol{\theta}_{t+1} = \boldsymbol{\theta}_t + \alpha_t \hat{\boldsymbol{g}}_t$;

---

## 3   ZEROTH-ORDER SIGN POLICY OPTIMIZATION FROM PREFERENCE

We propose ZSPO for preference learning with an unknown link function. The algorithm is summarized in Algorithm 1. At each policy iteration, say round $t$, it consists of the following steps:

1. Perturb the current policy $\pi_{\boldsymbol{\theta}_t}$ with a randomly sampled vector $\boldsymbol{v}_t$ from a standard normal distribution and distance $\mu_t$ to obtain the perturbed policy $\pi_{\boldsymbol{\theta}_t'}$ (line 2-3).

2. Sample $N$ pairs of trajectory batches with size $D$ under both policies $\pi_{\boldsymbol{\theta}_t}$ and $\pi_{\boldsymbol{\theta}_t'}$ (line 4-6).

3. For each pair of batches, query the oracle for preference feedback (line 7).

4. Use the majority vote over the preference of $N$ pairs to estimate ascent direction $\hat{\boldsymbol{g}}_t$ (line 8).

5. Update the current policy $\pi_{\boldsymbol{\theta}_t}$ with learning rate $\alpha_t$ and ascent direction $\hat{\boldsymbol{g}}_t$ (line 10).

Two main components are used to build ZSPO: (i) estimate the sign of the value function difference between the current policy $\pi_{\boldsymbol{\theta}_t}$ and the perturbed policy $\pi_{\boldsymbol{\theta}_t'}$, which is controlled by the perturbation distance $\mu_t$ at each iteration, and (ii) use the sign of the value function difference to construct a gradient estimator $\hat{\boldsymbol{g}}_t$ that has a positive correlation with the policy gradient $\nabla_{\boldsymbol{\theta}} V(\pi_{\boldsymbol{\theta}_t})$ in expectation, and then use gradient ascent to find the optimal policy. We illustrate both aspects in more detail.

**Policy Optimization from Signed Feedback.** Suppose we have a policy oracle that can compare the value function of $\pi_{\boldsymbol{\theta}_t}$ and $\pi_{\boldsymbol{\theta}_t'}$ and obtain $\text{sign}[V(\pi_{\boldsymbol{\theta}_t'}) - V(\pi_{\boldsymbol{\theta}_t})]$. Then, we can construct the gradient direction estimator $\hat{\boldsymbol{g}}_t$ from the perturbation direction $\boldsymbol{v}_t$ as: $\hat{\boldsymbol{g}}_t = \text{sign}[V(\pi_{\boldsymbol{\theta}_t'}) - V(\pi_{\boldsymbol{\theta}_t})]\boldsymbol{v}_t$. Intuitively, $\hat{g}_t$ aligns with the gradient $\nabla_{\boldsymbol{\theta}} V(\pi_{\boldsymbol{\theta}_t})$: suppose the perturbation distance $\mu$ is small, so under mild conditions, we can linearize the value function difference around the neighborhood of $\boldsymbol{\theta}_t$:

$$V(\pi_{\boldsymbol{\theta}_t'}) - V(\pi_{\boldsymbol{\theta}_t}) \approx \langle \nabla_{\boldsymbol{\theta}} V(\pi_{\boldsymbol{\theta}_t}), \boldsymbol{\theta}_t' - \boldsymbol{\theta}_t \rangle = \mu \langle \nabla_{\boldsymbol{\theta}} V(\pi_{\boldsymbol{\theta}_t}), \boldsymbol{v}_t \rangle. \tag{3}$$

Therefore, the sign of the value function difference can be approximated as follows:

$$\text{sign}[V(\pi_{\boldsymbol{\theta}_t'}) - V(\pi_{\boldsymbol{\theta}_t})] \approx \text{sign}[\langle \nabla_{\boldsymbol{\theta}} V(\pi_{\boldsymbol{\theta}_t}), \boldsymbol{v}_t \rangle]. \tag{4}$$

In other words, if the sign of the value function difference is positive, then the perturbation vector $\boldsymbol{v}_t$ is likely to have a positive inner product with the gradient $\nabla_{\boldsymbol{\theta}} V(\pi_{\boldsymbol{\theta}_t})$, and if the sign of the value function difference is negative, $-\boldsymbol{v}_t$ will be positively aligned with the gradient, which ensures a convergence dynamic similar to stochastic gradient. This function difference sign approach is unconventional in the zeroth-order literature, which we further discuss in appendix D.1.

**Value Function Preference Approximation.** The value-function-based preference oracle is usually unrealistic ($D = +\infty$ in equation 2). Therefore, we use batched trajectory preferences to estimate the value function difference sign with a majority vote rule. Specifically, we ask the preference oracle to compare different pairs of trajectory batches generated from the two policies with a proper batch size. Then, we perform a majority vote on which policy has a higher value function and take the

policy with more votes. The majority vote rule helps tackle the unknown link function setting and resembles the preference based on value functions under mild conditions.

**Link Function for Reward Estimation.** The link function $\sigma(\cdot)$ plays an important role in preference learning problems. In reward inference, the agent fine-tunes the reward model parameters to maximize the likelihood of the preference outcomes for the offline dataset. This can only be achieved when the preference generation mechanism, i.e., the link function $\sigma(\cdot)$, is known. DPO uses almost the same idea, and additionally views the reward function as an intermediate step, expressed as a function of the optimal policy $\pi^*$. Similarly, ZPG uses the inverse link function $\sigma^{-1}(\cdot)$ to recover the reward difference of trajectories to estimate the zeroth-order gradient. Almost all algorithms in the literature explicitly use the link function, and therefore, become inapplicable when the link function $\sigma(\cdot)$ is unknown, It becomes difficult, if not impossible, to recover the true reward function from preference.

**Link Function Agnostic Sign Estimation.** The main reason ZSPO can be applied with unknown link functions is that ZSPO does not attempt to recover the full numerical reward information from preference. Specifically, for each trajectory pair $(\tau_1, \tau_0)$, ZPG from (Zhang & Ying, 2025) queries each trajectory pair with multiple oracles to estimate the preference probability $\mathbb{P}(\tau_1 \succ \tau_0)$, and then plug it into the inverse link function $\sigma^{-1}(\cdot)$ to estimate the return difference $r(\tau_1) - r(\tau_0)$. This step is required for value function difference estimation and policy gradient approximation. On the other hand, ZSPO in this paper only estimates the sign of the reward difference and then uses a majority vote rule to reconstruct the sign of the value function difference. The information is much more compressed, but is much easier to recover from preferences, which does not require knowledge of the link function. In the meantime, this piece of information turns out to be sufficient to infer the landscape of the value function and guarantees the convergence of policy gradient algorithms.

## 4 MAIN RESULTS

In this section, we theoretically analyze the performance of ZSPO. We first introduce the assumptions on the landscape of value functions, the preference model, and the distinguishability with preferences.

### 4.1 DEFINITIONS AND ASSUMPTIONS

We impose the following assumption on the link function $\sigma(\cdot)$, satisfied by the BT model. A weaker version is justified in (Wang et al., 2023) as the minimal requirement to learn the optimal policy.

**Assumption 2** *The link function $\sigma(\cdot)$ is L-smooth with a positive derivative at the origin, i.e.,*

$$|\sigma'(x) - \sigma'(y)| \leq L|x - y|, \quad \forall x, y \in [-H, H], \quad and \quad \sigma'(0) > 0.$$

We also require the landscape of the value function and the policy network to be "regular", and impose the following standard smoothness assumption in nonconvex optimization (Liu et al., 2019; Bernstein et al., 2018; Reddi et al., 2018) and reinforcement learning (Zhang & Ying, 2025). Notice that linearly realizable MDPs (Weisz et al., 2023; Li et al., 2021), including linear MDPs (Jin et al., 2020), naturally satisfy this assumption when the policy parameterization $\pi_{\boldsymbol{\theta}}$ is smooth.

**Assumption 3** *The value function $V(\pi_{\boldsymbol{\theta}})$ for the network parameter $\boldsymbol{\theta}$ is L-smooth on $\mathbb{R}^d$, i.e.,*

$$\|\nabla_{\boldsymbol{\theta}} V(\pi_{\boldsymbol{\theta}_1}) - \nabla_{\boldsymbol{\theta}} V(\pi_{\boldsymbol{\theta}_2})\|_2 \leq L\|\boldsymbol{\theta}_1 - \boldsymbol{\theta}_2\|_2, \quad \forall \boldsymbol{\theta}_1, \boldsymbol{\theta}_2.$$

For simplicity, we use $L$ to represent the upper bound of the smoothness constants in both assumptions. If the perturbed parameter $\boldsymbol{\theta}'_t$ is close to the original parameter $\boldsymbol{\theta}_t$, the value function of the two policies will also be close. However, in this case, the preference oracle may have difficulty finding the better policy from comparing trajectories with a finite batch size $D$. Let $\varsigma(x) = \sigma(x) - 1/2$ be the (preference) *deviation* function we define distinguishability as follows:

**Definition 1 (Distinguishability)** *For any RL problem $\mathcal{M}$ and deviation function $\varsigma(\cdot)$, define the preference distinguishability $\varepsilon_D^*$ under batch size $D$ to be the maximum constant $\varepsilon$, such that for any two policies $\pi_0$ and $\pi_1$ with $V(\pi_1) - V(\pi_0) \geq \varepsilon$, we have:*

$$\mathbb{E}_{\mathcal{D}_0 \sim \pi_0, \mathcal{D}_1 \sim \pi_1} \left[ \varsigma \left( \bar{r}\left(\mathcal{D}_1\right) - \bar{r}\left(\mathcal{D}_0\right)\right)\right] \geq \frac{1}{2}\varsigma\left(\frac{V(\pi_1) - V(\pi_0)}{2}\right).$$

*where $\mathcal{D}_0$ and $\mathcal{D}_1$ are trajectory batches generated by $\pi_0$ and $\pi_1$ respectively with $|\mathcal{D}_0| = |\mathcal{D}_1| = D$.*

When two policies with a value function difference smaller than $\varepsilon_D^*$ are compared, the preference oracle may not distinguish the better policy, which reveals a fundamental limit of learning from preference with an unknown link function $\sigma(\cdot)$. So, to effectively compare the policy $\pi_{\boldsymbol{\theta}_t}$ and its perturbation, we need to control the distance $\mu_t$ and make sure they are distinguishable. We can derive an upper bound for $\varepsilon_D^*$ as follows.

**Proposition 1** *For any RL problem $\mathcal{M}$ and any deviation function $\varsigma(\cdot)$ that satisfy assumption 1 and 2, the distinguishability $\varepsilon_D^*$ under batch size $D$ satisfies $\varepsilon_D^* = \widetilde{\mathcal{O}}(H/\sqrt{D})$.*

The proof is based on concentration and is deferred to the appendix section E. This shows that when the batch size is large, the average rewards of each batch $\mathcal{D}_0$ and $\mathcal{D}_1$ are concentrated around the value function, so the preference over the trajectory batches is almost the preference over the policies. More discussions of the limit of distinguishability are provided in the appendix section D.2.

### 4.2 Convergence Rate

In this section, we present the theoretical guarantees under the assumptions in the previous sections. We aim to learn an $\epsilon$-stationary policy $\pi_{\boldsymbol{\theta}}$ with $\|\nabla_{\boldsymbol{\theta}} V(\pi_{\boldsymbol{\theta}})\|_2 \leq \epsilon$, and study the convergence rate.

**Theorem 1** *Choose the perturbation distance to be time homogeneous, i.e., $\mu_t = \mu$ and the learning rate to be $\alpha_t = \Theta(\sqrt{H/dt})$. If we randomly pick $\boldsymbol{\theta}_R$ from $\{\boldsymbol{\theta}_1, \boldsymbol{\theta}_2, \cdots, \boldsymbol{\theta}_T\}$ with $\mathbb{P}(\boldsymbol{\theta}_R = \boldsymbol{\theta}_t) = \alpha_t / \sum_{i=1}^T \alpha_i$, then the convergence rate of ZSPO satisfies:*

$$\mathbb{E}\left[\|\nabla_{\boldsymbol{\theta}} V(\pi_{\boldsymbol{\theta}_R})\|_2\right] = \widetilde{\mathcal{O}}\left(\sqrt{\frac{Hd}{T}} + \frac{1}{\mu}\left(\varsigma^{-1}\left(\sqrt{\frac{2}{N}}\right) + \varepsilon_D^*\right) + \mu d\right).$$

The complete proof is provided in the appendix section F. We first illustrate the insight of the convergence rate, the choice of the hyperparameters, and the technical novelties and challenges.

**Insights.** The convergence rate of ZSPO has three components: the convergence rate of zeroth order optimization, the preference distinguishability $\varepsilon_D^*$, and the majority vote approximation error:

$$\underbrace{\sqrt{\frac{Hd}{T}} + \mu d}_{\text{Zeroth-Order Optimization}} + \underbrace{\frac{\varepsilon_D^*}{\mu}}_{\text{Distinguishability}} + \underbrace{\frac{1}{\mu}\varsigma^{-1}\left(\sqrt{\frac{2}{N}}\right)}_{\text{Majority Vote Approximation Error}} .$$

The first term resembles zeroth-order stochastic gradient descent (Nesterov & Spokoiny, 2017), stochastic coordinate descent (Cai et al., 2021), and sign gradient descent (Liu et al., 2019). If we choose $\mu = \mathcal{O}(1/\sqrt{dT})$ as in the literature, this term matches the state-of-the-art $\mathcal{O}(\sqrt{d/T})$ result for non-convex smooth function optimization. The second term comes from the distinguishability limit of the preference oracle: when the current policy $\boldsymbol{\theta}_t$ is close to stationary, i.e., the gradient norm is smaller than $\varepsilon_D^*/\mu$, the perturbed policy and the current policy have similar value functions with difference smaller than $\varepsilon_D^*$ according to equation 3, which becomes indistinguishable. One could also view the parameter $\boldsymbol{\theta}_R$ learned ZSPO as the policy most preferred by the oracle in the $\varepsilon_D^*$-neighborhood of a stationary policy. The third term comes from approximating the expected preference probability with a majority vote. As the number of batches $N$ increases, the approximation error would decrease since $\varsigma^{-1}(\sqrt{2/N}) \to \varsigma^{-1}(0) = 0$, since the majority vote becomes more accurate and reflects the population-level preference.

**Optimizing the Convergence Rate Upper Bound.** The rate of convergence in Theorem 1 trades off the zeroth-order optimization error with both the distinguishability and the majority vote approximation error. Optimizing the bound results in the tightest characterization as follows:

**Corollary 1** *If we choose $\alpha_t = \Theta(\sqrt{H/dt})$ and $\boldsymbol{\theta}_R$ the same way as in Theorem. 1, and if the perturbation distance $\mu$ satisfies: $\mu^2 = \Theta(d^{-1}\max\{\varsigma^{-1}(\sqrt{2/N}), \varepsilon_D^*\})$, Then, we have:*

$$\mathbb{E}\left[\|\nabla_{\boldsymbol{\theta}} V(\pi_{\boldsymbol{\theta}_R})\|_2\right] = \sqrt{d} \cdot \widetilde{\mathcal{O}}\left(\sqrt{\frac{H}{T}} + \left[\varsigma^{-1}\left(\sqrt{\frac{2}{N}}\right)\right]^{1/2} + \sqrt{\varepsilon_D^*}\right).$$

However, to achieve this rate, we need to fine-tune the hyperparameter $\mu$ in practice or require the knowledge of $\varsigma(\cdot)$ and $\varepsilon_D^*$. Nonetheless, some insights are offered. First, $\mu$ cannot be too large because the zeroth-order estimator of the ascent direction is only accurate when the perturbation distance is small, as shown in non-convex optimization (Nesterov & Spokoiny, 2017; Liu et al., 2019). Second, $\mu$ also cannot be arbitrarily small as chosen in vanilla zeroth-order optimization algorithms because, in that case, the perturbed policy $\pi_{\boldsymbol{\theta}_t'}$ may be indistinguishable by the preference oracle, and the convergence is not guaranteed. The moderate perturbation requirement is similar to ZPG (Zhang & Ying, 2025), because in both algorithms, the gradient bias is amplified by the inverse of perturbation distance. However, the reasons are different: in ZSPO, the bias comes from the distinguishability of preference and approximating the population-level preference via majority vote, and in ZPG, it comes from recovering the reward difference using a non-linear link function.

**Preference Oracle Quality.** The convergence rate in Theorem 1 depends on the preference model, i.e., the deviation function $\varsigma(\cdot)$, which constitutes the majority vote error. If the preference oracle is more sensitive to distinguish candidates with similar average returns, $\varsigma(\cdot)$ is closer to a step function. Then, the majority vote error will decrease faster, resulting in a better convergence rate. On the other hand, for the same pair of trajectories, we can also query multiple different preference oracles (multiple evaluators) to provide preferences and then aggregate the results via another majority vote. This is equivalent to querying a preference oracle with a more step-like deviation function, i.e., with more expertise, and a better convergence rate is anticipated.

**Practical Choice of Perturbation.** To obtain a practical choice of perturbation distance $\mu$ which does not rely on unknown quantities such as $\varepsilon_D^*$ and $\varsigma(\cdot)$, we seek to construct an upper bound for the optimal choice of $\mu$. The distinguishability $\varepsilon_D^*$ can be bounded with Proposition 1, and the majority vote approximation error is bounded under the smoothness assumption of the link function, i.e., $\varsigma^{-1}(\sqrt{2/N}) = \mathcal{O}(1/\sqrt{N})$. Then, we have the following corollary:

**Corollary 2** *Choose $\alpha_t = \Theta(\sqrt{H/dt})$ and $\mu^2 = \Theta(d^{-1}\max\{1/\sqrt{N}, H/\sqrt{D}\})$, If we randomly pick $\boldsymbol{\theta}_R$ the same way as Theorem 1, then the convergence rate of ZSPO satisfies:*

$$\mathbb{E}\left[\|\nabla_{\boldsymbol{\theta}}V(\pi_{\boldsymbol{\theta}_R})\|_2\right] = \sqrt{d} \cdot \widetilde{\mathcal{O}}\left(\sqrt{\frac{H}{T}} + \frac{1}{\min\{\sigma'(0), 1\}\,N^{1/4}} + \frac{\sqrt{H}}{D^{1/4}}\right).$$

The proof is in appendix G. The canonical Bradley-Terry model has derivative $\sigma'(0) = 1/4$. It implies that we need to choose the batch size $D$ as large as possible (within the capacity of the preference oracle) to maintain distinguishability when the parameter $\boldsymbol{\theta}_t$ is around the neighborhood of convergence. We also choose the number of batches $N$ to be large so that the majority vote result is accurate and reflects the value function sign. Finally, we prefer preference oracles with more capacity and a larger $\sigma'(\cdot)$, which reflects the sharpness of oracles towards trajectories with similar returns.

### 4.3 TECHNICAL CHALLENGES AND PROOF NOVELTIES

In this section, we provide a proof roadmap for Theorem 1 and discuss its novelty compared to results for zeroth-order optimization (Nesterov & Spokoiny, 2017; Liu et al., 2019).

**Smoothing Function Framework.** Assume the perturbation distance $\mu_t = \mu$ is time-homogeneous. Classic proofs in the zeroth-order optimization literature, including the convergence of ZPG, make use of a smoothing function $V_\mu(\pi_{\boldsymbol{\theta}_t})$ whose derivative is the expectation of the gradient estimator $\hat{\boldsymbol{g}}_t$. For example, in zeroth-order stochastic gradient descent or sign gradient descent, the smoothing function is defined as the expected value function of the perturbed parameter: $V_\mu(\pi_{\boldsymbol{\theta}}) = \mathbb{E}_{\boldsymbol{v}}\left[V(\pi_{\boldsymbol{\theta}+\mu\boldsymbol{v}})\right]$, where $\boldsymbol{v}$ follows some distribution in $\mathbb{R}^d$. Moreover, when $\mu$ is small, the smoothing function will behave almost the same as the original value function. Then, using the smoothing value function as the Lyapunov function and combining it with the smoothness assumption, we can obtain the following inequality, neglecting problem-independent constants:

$$V_\mu(\boldsymbol{\theta}_t) - V_\mu(\boldsymbol{\theta}_{t+1}) \leq -\alpha_t \underbrace{\|\nabla_{\boldsymbol{\theta}}V_\mu(\pi_{\boldsymbol{\theta}_t})\|_2^2}_{\text{Drift}} + \alpha_t \underbrace{\langle\nabla_{\boldsymbol{\theta}}V_\mu(\pi_{\boldsymbol{\theta}_t}), \nabla_{\boldsymbol{\theta}}V_\mu(\pi_{\boldsymbol{\theta}_t}) - \hat{\boldsymbol{g}}_t\rangle}_{\text{1st Order: bias}}$$

$$+ \alpha_t^2 \underbrace{\|\hat{\boldsymbol{g}}_t - \nabla_{\boldsymbol{\theta}}V_\mu(\pi_{\boldsymbol{\theta}_t})\|_2^2}_{\text{2nd Order: var}}.$$

Since in the classic setting, the expectation of $\hat{g}_t$ is the gradient for the smoothing function, the first order term `bias` is $0$ in expectation, and the second order term `var` is much smaller than the drift term when the learning rate $\alpha_t$ is small. Taking a telescoping sum over both sides and dividing by the sum of learning rates, we can obtain a bound for the gradient of the smoothing function $V_\mu(\pi_\theta)$, which can be transferred to the bound for the original value function $V(\pi_\theta)$ when $\mu$ is small.

**Proof Roadmap for `ZSPO`.** However, it is difficult to construct a smoothing function for the ascent direction estimator $\hat{g}_t$ in `ZSPO`, since it involves feedback with an unknown link function and thus can only preserve the information of the value function sign. The smoothing function framework cannot be directly applied, and we use the value function as the Lyapunov function. Nonetheless, this would lead us to the following drift analysis, neglecting problem-independent constants:

$$V(\pi_{\theta_t}) - V(\pi_{\theta_{t+1}}) \leq - \alpha_t \underbrace{\mathrm{sign}\left[V(\pi_{\theta_t'}) - V(\pi_{\theta_t})\right]\langle \nabla_\theta V(\pi_{\theta_t}), v_t\rangle}_{A_1} + \alpha_t^2 \underbrace{\mathbb{E}[\|v_t\|_2^2]}_{=d}$$

$$- \alpha_t \underbrace{\left(\mathrm{sign}\left[\sum_{n=1}^{N} o_{t,n} - \frac{1}{2}\right] - \mathrm{sign}\left[V(\pi_{\theta_t'}) - V(\pi_{\theta_t})\right]\right)\langle \nabla_\theta V(\pi_{\theta_t}), v_t\rangle}_{A_2}.$$

Ideally, we envision $A_1$ to constitute a negative drift since the sign of the value function difference resembles the sign of $\langle \nabla_\theta V(\pi_{\theta_t}), v_t\rangle$ due to the linear approximation in equation 3, i.e.,

$$\mathrm{sign}\left[V(\pi_{\theta_t'}) - V(\pi_{\theta_t})\right] \approx \mathrm{sign}\left[\mu\langle \nabla_\theta V(\pi_{\theta_t}), v_t\rangle\right] = \mathrm{sign}\left[\langle \nabla_\theta V(\pi_{\theta_t}), v_t\rangle\right], \qquad (5)$$

and therefore the expectation of $D_1$ will be approximated as follows:

$$\mathbb{E}[A_1] \approx \mathbb{E}\left[\mathrm{sign}\left[\langle \nabla_\theta V(\pi_{\theta_t}), v_t\rangle\right]\langle \nabla_\theta V(\pi_{\theta_t}), v_t\rangle\right] = \mathbb{E}\left[|\langle \nabla_\theta V(\pi_{\theta_t}), v_t\rangle|\right],$$

which is proportional to the expected norm of gradient $\mathbb{E}[\|\nabla_\theta V(\pi_{\theta_t})\|_2]$ from Khintchine's inequality (Vershynin, 2018). Then, a negative drift is constructed, which allows us to bound the value function difference as follows:

$$\mathbb{E}[V(\pi_{\theta_t}) - V(\pi_{\theta_{t+1}})] \lesssim -\alpha_t\sqrt{\frac{2}{\pi}}\mathbb{E}[\|\nabla_\theta V(\pi_{\theta_t})\|_2] + \alpha_t^2 d + \mathbb{E}[A_2].$$

However, the sign between the value function difference and the inner product $\langle \nabla_\theta V(\pi_{\theta_t}), v_t\rangle$ is not guaranteed to be the same due to higher-order terms in the linear approximation error. To ensure both terms have the same sign and make equation 5 exactly an equality, the magnitude of the first-order inner product should dominate the sum of higher-order errors, which requires the value function difference to be large enough. Therefore, to analyze this error resulting from the sign of linear approximation, we need to condition on the sampled vector $v_t$ and divide the event space into events with high value difference and events with a low value difference, and then analyze the error in each event separately. The characterization of $A_2$, i.e., the approximation error of using the preference to estimate the sign of the value function difference, mainly follows a classic concentration argument. However, due to the existence of the distinguishability limit $\varepsilon_D^*$, the signs of both values only coincide with one another when the function difference is large, similar to the error in $A_1$. Therefore, we again need to perform an event separation to bound the approximation error.

## 5 EMPIRICAL EVALUATION WITH LINK FUNCTION MIS-SPECIFICATION

To validate our theoretical results, and to evaluate the practicality of `ZSPO` in real-world tasks, we conducted empirical experiments on a set of robotic RL tasks in `Gymnasium` (Towers et al., 2024).

**Environments.** We considered three simulated robotic environments, two of which use the `MuJoCo` physics engine, i.e., `CartPole`, `HalfCheetah`, and `Hopper`. To isolate and measure the influence of a mis-specified link function, we used a synthetic oracle with a linear link function, where the expected preference probability of two trajectories equals a capped linear link function (Chen & Frazier, 2017; Bengs et al., 2021) of the cumulative environmental reward difference, i.e., for two trajectories $\tau_0$ and $\tau_1$, we have:

$$\mathbb{P}(\tau_1 \succ \tau_0) = \max\left\{\min\left\{\gamma\left[r(\tau_1) - r(\tau_0)\right] + \frac{1}{2}, 1\right\}, 0\right\},$$

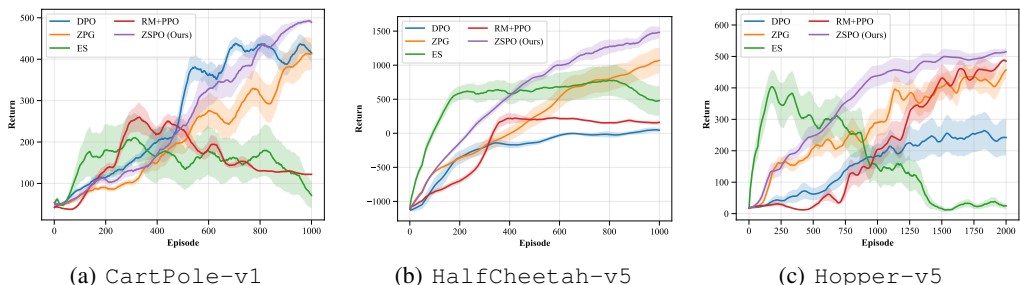

|   | (a) CartPole-v1 | (b) HalfCheetah-v5 | (c) Hopper-v5 |

Figure 1: Averaged running return (mean ± std) during training with a link function mis-specification. The returns are smoothed with a proper window size for clear presentation.

| Environments | CartPole-v1 | HalfCheetah-v5 | Hopper-v5 |
|---|---|---|---|
| DPO | $438.1 \pm 16.6$ | $60.0 \pm 33.8$ | $264.4 \pm 60.0$ |
| RM+PPO | $260.4 \pm 35.8$ | $226.54 \pm 21.4$ | $488.3 \pm 3.6$ |
| ES | $210.5 \pm 58.9$ | $778.6 \pm 224.5$ | $403.8 \pm 52.0$ |
| ZPG | $413.6 \pm 40.8$ | $1068.9 \pm 187.0$ | $457.1 \pm 27.7$ |
| ZSPO(Ours) | $\mathbf{493.1 \pm 4.4}$ | $\mathbf{1484.4 \pm 86.4}$ | $\mathbf{514.9 \pm 10.9}$ |

Table 1: Averaged return for the best policy obtained (mean ± std), where the boldfaced entries represent the best results in each environment.

where $\gamma$ characterizes the expertise of the preference oracle similar to the Bradley-Terry model in equation 1. However, the true underlying link function is unknown to the algorithms. To study the robustness of algorithms in empirical tasks and investigate the limit of the proposed ZSPO, we chose $D = 1$, i.e., we only compared pairs of trajectories instead of trajectory batches. More details on the implementation and additional experiment results are provided in Appendix C.

**Algorithms and Baselines.** We used a fully-connected neural network with two hidden layers of 64 neurons as the actor network of all algorithms. We compared ZSPO to four baseline algorithms: (1) RM+PPO (Christiano et al., 2017), which trains a reward model first and then uses PPO over the reward model to train the actor network without regularization to the reference policy, (2) Online DPO (Dong et al., 2024; Guo et al., 2024), which optimizes the DPO loss and updates the reference policy for each policy iteration, (3) ZPG (Zhang & Ying, 2025), which uses the link function inverse to estimate the zeroth-order policy gradient, and (4) ES as evolution strategy (Busa-Fekete et al., 2014; Salimans et al., 2017), where the agent constantly "mutates" to obtain new policies and maintains the one most preferred by the preference oracle. For baseline algorithms such as DPO and ZPG that require a known link function, they presume the Bradley-Terry model with a logistic link function. For algorithms that either require a critic network or a reward model, we used another fully-connected neural network with a similar structure to the actor network. In all experiments, we used $N = 1$ to test the robustness of all algorithms, especially the proposed ZSPO, to noisy feedback in practical tasks. Between policy updates, each algorithm only rolled out one pair of trajectories and obtained preference feedback for this pair. We reported the average return curves during training over 10 random seeds for all algorithms in Fig. 1. To study the quality of the obtained policy, we reported the average return of the best policy obtained by each algorithm in Tab. 1.

**Learned Policy.** We first compare the average return of the final learned policies as shown in Fig. 1, and the average return of the best policy obtained via early stopping, as shown in Tab. 1. We observe that for both metrics, ZSPO achieved the highest average return in all three environments compared to all four baseline algorithms, including algorithms that require extra structures such as the reward model or the critic network. Specifically, ZSPO almost achieves an average return of 500 over multiple runs in CartPole, which is the highest possible trajectory reward in this environment. This shows that ZSPO has stably learned the optimal policy. For the other two environments, ZSPO also achieves an average return that is significantly higher than the baselines. This demonstrates the practicality of ZSPO in complex tasks and its robustness to the preference model mismatch,

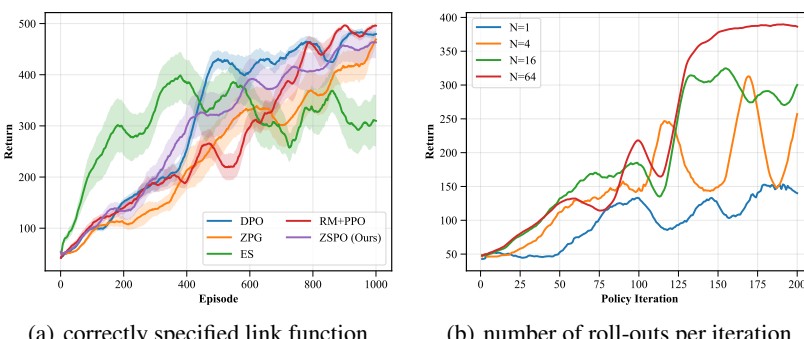

(a) correctly specified link function       (b) number of roll-outs per iteration

Figure 2: Ablation studies on `CartPole-v3`: (a) the average returns under the Bradley-Terry preference and (b) the average return of `ZSPO` with different numbers of roll-out trajectories.

even with limited trajectories and preference feedback. It is also interesting to observe that `ZSPO` learned a policy with the best performance even under a single trajectory comparison, i.e., $N = 1$ and $D = 1$, despite our theoretical results requiring large $N$ and $D$. We remark that the empirical results do not contradict with our theoretical results because (i) the tested environment transitions are mainly deterministic, resulting in a low variance in policy evaluation, so we do not need a large $N$ for variance reduction; and (2) the trajectory reward is regular enough so that the distinguishability $\varepsilon_1^*$ is small to ensure consistency.

**Training Dynamic.** We then zoom in on the training dynamics shown in Fig. 1 to understand the influence of link function mismatch. Specifically, the `DPO` loss is not the correct loss with a link function mis-specification, so `Online DPO` exhibits unstable or slow training dynamics in more complex MoJoCo environments. `RM+PPO` suffers from the same reason, where the reward model does not accurately learn the true environmental reward function due to the link function mis-specification. Due to this reason, it may have a slow training dynamic, or get stuck on highly sub-optimal policies, or simply cannot maintain the best performance achieved in previous policy iterations. Therefore, the best policies learned by these two methods are inferior to `ZSPO` since the intermediate reward, either implicitly or explicitly learned from the preference with a mis-specification, deviates from the true reward function, and thus shifts the learned policy. The link function mis-specification also influences the convergence of `ZPG`. The sign of the value function difference recovered by `ZPG` is still accurate, but the difference is not estimated accurately and could be arbitrary in essence. Therefore, compared to `ZSPO`, `ZPG` fails to use the correct distance to move into the ascent direction, resulting in overshoots and undershoots in the training dynamic.

**Ablation Studies and Connections to Theory.** We first studied the performance of algorithms under the Bradley-Terry preference mechanism, where the true preference probability comes from a logistic function, to identify the impact of link function mis-specification and the robustness of `ZSPO` across link functions. By comparing Fig. 2(a) with Fig. 1(a), we observed that all baselines suffer performance loss when we switched from a correctly specified link function to a mis-specified link function. It is also observed that `ZSPO` maintains a robust performance across different link functions. We then studied the performance of `ZSPO` under different $N$, the number of trajectory roll-outs per iteration, to validate the variance reduction. It is observed that with a larger $N$, the performance of `ZSPO` improved much quicker, which is consistent with our theoretical results.

## 6 CONCLUSION

In this paper, we studied preference-based RL with an unknown link function. We developed `ZSPO`, which estimates the sign of the value function difference from preferences and constructs an ascent direction from it. `ZSPO` has a provable convergence guarantee with polynomial sample and preference-query complexities, validated also by numerical experiments. The future direction involves combining `ZSPO` with successful zeroth-order algorithms such as `MeZO` (Malladi et al., 2023) for LLMs to evaluate its practicality in more complex real-world RL tasks.

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

APPENDIX TABLE OF CONTENTS

## A  STATEMENTS

**Ethical Statement.** The contribution of the paper focuses on theoretically understanding the fundamental limits of preference-based reinforcement learning when the mechanism of preference generation, characterized by the link function, is unknown. The scope of this paper is mainly theoretical, and the preferences are modeled as random variables with a proper distribution. The theoretical findings of this paper point out insights towards real-world applications which has an impact in many aspects, but the authors feel none of them should be emphasized as potential negative outcomes.

**Reproducibility Statement.** The main focus of this paper is theoretical, where proofs of all results are provided in the appendix. Empirical experiments serve mainly as a proof of concept. We used a publicly available RL environment (Towers et al., 2024) and a toy `GridWorld` example in the appendix to evaluate the empirical performance of our proposed method. The important details of the implementation are demonstrated in section 5 and section C respectively. Reproducing the results, given the simplicity of the algorithm with the details stated, should require minimal effort. The source codes are submitted with the paper for reproducibility and will be publicly available.

**LLM Use Statement.** The LLMs are only used in this paper to correct grammar mistakes. LLMs do NOT contribute significantly at the level of a contributing author.

# B RELATED WORKS

In this section, we review the literature on preference-based RL and zeroth-order optimization that is relevant to our paper. One could refer to (Kaufmann et al., 2023) and (Casper et al., 2023) for a thorough survey on these topics.

**Empirical Studies on RL from Preference.** Reinforcement learning from human feedback has been used in various fields such as training robotics (Christiano et al., 2017) and large language models (Ouyang et al., 2022), where the preference is used to align their behavior with human interest. The typical pipeline consists of three steps: (i) pre-train a actor network with supervised learning, (ii) generate multiple pairs of trajectories to query human experts for preference and then use the human feedback to train a reward model to infer the true reward of the RL task, and (iii) use off-the-shell classic RL algorithms to train the actor network with the help of the reward model. So far, most works in the literature of empirical RLHF follow this pipeline and focus on either improving the quality of the reward model (Gao et al., 2023; Wirth et al., 2016), or developing better RL algorithms tailored for learning the optimal policy from an imperfect reward proxy in each application field (Guo et al., 2025; Rae et al., 2021; Ahmadian et al., 2024). However, it is commonly observed that training agents from reward models is prone to reward hacking (Skalse et al., 2022) and overfitting (Zhu et al., 2024), and therefore, direct RLHF approaches such as DPO (Rafailov et al., 2023; 2024; Dong et al., 2024; Xiong et al., 2024) and SLiC-HF (Zhao et al., 2023) are also studied to avoid reward model training, which utilizes a direct relationship between the optimal policy and the reward function in some specific settings such as contextual bandits or deterministic MDPs with KL regularization. These approaches, as pointed out in (Zhang & Ying, 2025), are difficult to generalize to settings beyond LLMs with stochastic transitions. An essential step in the aforementioned works that relates the human preference with the true reward of each state-action pair is to make use of the Bradley-Terry model (Bradley & Terry, 1952):

$$\mathbb{P}\left(\tau_1 \succ \tau_0\right) = \frac{1}{1 + \exp\left(r(\tau_0) - r(\tau_1)\right)}.$$

Therefore, one could formulate the likelihood of observing the human feedback when the reward is unknown, i.e., a cross-entropy loss. Then, by minimizing this loss function, we could either estimate the reward function through approximations or utilize the relation between the optimal policy and the reward function and directly learn the optimal policy. However, when used in real tasks where the human preference does not exactly follow the Bradley-Terry model, these approaches suffer from model misspecification, and the performance loss could be non-negligible.

**Provable Preference-Based RL with Known Link Function.** Even though empirical studies have been conducted for some time, it was not until recent years that provable preference-based RL algorithms were studied. For both reward inference and direct methods, previous works have attempted to characterize their provable performance and provide insights for developing better algorithms. In (Wang et al., 2023) and (Du et al., 2024), a preference-to-reward intermediate module was considered to infer the reward function from preference feedback with an MLE loss. Both value-based and policy-based RL algorithms are analyzed when learning from the approximated reward. Both works convey the message that with a reward model, preference-based RL should not be significantly harder than standard RL. Similar analysis has also been conducted in contemporary theoretical preference-based RL papers such as (Saha et al., 2023; Zhan et al., 2024a;b; Zhu et al., 2023; Kong & Yang, 2022; Wu & Sun, 2024) for offline, online, and hybrid RL problems. The analysis usually characterizes the error of the reward model parameters using the concentration property of the MLE estimator and then integrates the error into the sample complexity proof for standard RL algorithms. On the other hand, the analysis of direct approaches receives less attention due to the non-standard analysis framework of each method (Li et al., 2023; Chakraborty et al., 2024; Kausik et al., 2024). (Azar et al., 2024) attempts to analyze the theoretical performance of DPO, but only shows the existence of loss function optima. For deterministic MDPs, (Xie et al., 2024) combines DPO with optimistic exploration to provide a provable convergence to the KL-constrained optimal policy. (Xu et al., 2020) and (Zhang et al., 2024a) reduce the tabular MDP problem into a sequence of dueling bandit problems and provide both instance-independent and instance-dependent sample complexity guarantees. For general MDPs with infinite state-action pairs, (Zhang & Ying, 2025; Tang et al., 2024b) establishes the relation between human preference and zeroth-order gradient to design an algorithm with a provable convergence guarantee. However, the theoretical guarantees of the works mentioned above require the preference to be either generated from the Bradley-Terry

model or a preference model with a known link function. This assumption is likely to fail in reality, and the convergence guarantee may not be meaningful.

**RL with Unknown Relation between Reward and Preference.** Some previous researches also recognize the complex nature of humans and studies preference-based RL without exactly knowing how preference is generated. In this setting, the classic MLE loss cannot be formulated, and it is difficult to infer the complete reward information from preference data. Most studies around this topic change the definition of optimality. They define the optimal policy as the one most preferred by evaluators instead of the policy with maximum cumulative reward. Specifically, in dueling bandits literature, (Yue & Joachims, 2009; 2011) specifies a preference over all pairs of arms and assumes properties such as transitivity and triangle inequality so that one will be able to learn the most preferred arm via comparisons (Bengs et al., 2021). In the RL setting, (Azar et al., 2024) and (Tang et al., 2024a) propose to optimize a general non-decreasing loss function of the population-level preference probability instead of the cumulative reward, and (Chen et al., 2022) proposes to learn the mapping between trajectory pairs and the preference. Recently, a line of work named Nash learning from preference feedback (Munos et al., 2024) has been considered, where they define the best policy in a game-theoretic manner to avoid assumptions such as transitivity. In their definition, the best policy is the one that achieves the largest population-level preference probability when competing against any other policy using preference feedback as a mechanism. Works have extended this idea both theoretically (Ye et al., 2024; Zhou et al., 2025) and empirically (Rosset et al., 2024; Zhang et al., 2024c). However, all previous works in this field do not aim to learn the reward maximization policy. As shown by (Zhang & Ying, 2025) and this paper, the reward maximization policy could be different from the policy most preferred by the preference oracle. Therefore, completely ignoring the reward structure of the RL problem may result in performance loss since real human feedback can be misleading or indifferent among candidates. Moreover, the performance gap between the policy learned by these works and the optimal return is not characterized.

**Zeroth-Order Optimization and Evolutionary Strategy.** When the first-order information, such as the gradient, cannot be easily obtained due to the limitation of the optimization problem itself or the high computation requirement, zeroth-order methods are usually considered for both convex and non-convex problems (Ghadimi & Lan, 2013; Nesterov & Spokoiny, 2017), where the authors used a two-point method to estimate the gradient through the function value difference and then proceed with stochastic gradient descent. Variants of the zeroth-order stochastic gradient descent have also been studied (Liu et al., 2018a;b; Cai et al., 2021; Gao et al., 2018) with block-coordinate descent and variance reduction tricks. These methods have also been used in optimizing the loss function in LLMs (Malladi et al., 2023; Zhang et al., 2024b) as well. However, in ordinary zeroth-order problems, the function value can be queried or estimated in the presence of noise, which is used to approximate the gradient via difference, i.e.,

$$\nabla f(\boldsymbol{x}) \approx \mathbb{E}_{\boldsymbol{v}} \left[ \frac{f(\boldsymbol{x} + \mu \boldsymbol{v}) - f(\boldsymbol{x})}{\mu} \right],$$

where $f$ is the loss function, $\boldsymbol{x}$ is a point, $\boldsymbol{v}$ is a randomly sampled vector from some symmetric distribution, and $\mu$ is a perturbation distance which should be chosen small. This idea is also adopted in (Zhang & Ying, 2025) for developing preference-based RL algorithms. However, in our setting with an unknown link function, we have no access to estimate the function value, and therefore, this approach can not be directly applied. One approach to circumvent this dilemma is to use the evolutionary strategy (Rechenberg, 1973), which does not estimate the gradient to proceed with gradient descent, but to update the current policy with a perturbed policy that has a better performance. As long as there is a way to verify the superiority of the perturbed policy versus the current policy, the value function difference would be of no significance. The evolutionary strategies have also been studied in classic RL (Salimans et al., 2017; Conti et al., 2018), preference-based RL (Busa-Fekete et al., 2014; Akrour et al., 2011), and language model optimizations (Malladi et al., 2023) as well. However, the theoretical guarantees of evolutionary algorithms are much less clear compared to zeroth-order optimization, and provable algorithms are underdeveloped. Another approach that does not rely on the value function difference is to replace gradient descent with sign gradient descent (Bernstein et al., 2018; Liu et al., 2019). But to estimate the element-wise sign of the gradient vector without using the value function difference, one needs to perturb each entry of the policy parameter one by one to obtain a perturbed policy for each entry, and compare the performance of each policy pair. This proves to be difficult when dealing with complex policy approximations such as neural networks. Our work is inspired by both fields and closely related to zeroth-order optimization

with one-bit feedback (Cheng et al., 2020; Cai et al., 2022; Zhang & Li, 2024), which receives much less attention compared to classic settings. However, these works assume a perfect preference oracle, but in our paper, we estimate the sign of the value function difference from noisy feedback and then plug it into the zeroth-order gradient descent framework.

**Link Function and Preference Model.** It has been a long-standing effort to understand the rationale for how humans make decisions, and establish models to predict human behaviors (Thurstone, 1927; Train, 2009; Greene, 2000; Meilgaard et al., 1999; Lawless & Heymann, 2010) from both social science, economics, and behavioral science fields. Despite the complexity of humans, the most dominant models that have been adopted in the literature are the random utility model in social choice theory (Azari et al., 2012), developed as early as the 1920s (Thurstone, 1927). The random utility model assumes each person is associated with a utility function for all candidates and will choose the action that maximizes the utility. Therefore, how the utility of each person is generated or distributed, which is characterized by the link function (Bengs et al., 2021), gives rise to different preference models. Even though the logistic link function, i.e., the Bradley-Terry model (Bradley & Terry, 1952), is mostly adopted in the literature due to the closed expression and easy-to-manipulate nature, other preference models such as the Probit model (Thurstone, 1927), the Cauchy model, the complementary log-log model, and the Weibull model (Greene, 2000) have also been studied in the literature. Any cumulative distribution function for continuous distributions would be a valid link function, and the best model to describe human behavior is yet debatable. Therefore, in this paper, we explore the common traits of admissible preference models and develop algorithms applicable to any preference model without knowing the link function.

## C    DETAILS OF EMPIRICAL EXPERIMENTS

In this section, we first describe the important details of the experiments conducted in section 5. Then, we perform additional empirical experiments on a Stochastic `GridWorld` environment (Zhang & Ying, 2025) to evaluate the influence of stochastic transitions and demonstrate the robustness of `ZSPO` in general RL problems.

### C.1    DETAILS OF ROBOTIC EXPERIMENTS BASED ON GYMNASIUM

We first present the implementation details of the experiments conducted in 5 for three robotic tasks from the `Gymnasium` benchmark: `HalfCheetah-v5`, `CartPole-v1`, and `Hopper-v5`. The benchmark has been widely used to evaluate RL algorithms in complex environments. The detailed information about these three environments could be found in Towers et al. (2024). We set the planning horizon $H$ to be the default value of each environment, i.e., $H = 500$ for `CartPole-v1`, $H = 1,000$ for both `HalfCheetah-v5` and `Hopper-v5`, and then we report the cumulative return of each episode.

**Preference Feedback.** In all three environments, we assume agents have access to multiple simulated preference oracles (representing different human evaluators or language model judges in the real world) that compare a pair of trajectories, i.e., $D = 1$ for the preference. Each oracle will generate preferences over two trajectories based on a linear link function on the trajectory returns. For example, for two trajectories $\tau_0$ and $\tau_1$, the linear model would have a preference probability as follows:

$$\mathbb{P}(\tau_1 \succ \tau_0) = \max\left\{\min\left\{\gamma\left[r(\tau_1) - r(\tau_0)\right] + \frac{1}{2}, 1\right\}, 0\right\},$$

where $\gamma$ is a constant characterizing the expertise of the preference oracle. Then, each preference oracle will sample a Bernoulli random variable with probability of success being $\mathbb{P}(\tau_1 \succ \tau_0)$ and use the result as the preference feedback to agents. For the expertise constant $\gamma$, we choose $\gamma = 10^{-3}$ for `CartPole-v1` and `HalfCheetah-v5`, and we use $\gamma = 5 \times 10^{-4}$ for `Hopper-v5` to ensure moderate randomness and uncertainty in preferences between trajectories. We set the number of preference oracles to be 100 so that all baseline algorithms can aggregate the feedback from different oracles to ensure a stable learning dynamic. We remark that for `ZSPO`, using multiple preference oracles with a smaller $\gamma$ is equivalent to using a single preference oracle with a large $\gamma$ due to the majority vote nature of the sign mechanism. Therefore, `ZSPO` does not unfairly benefit from using multiple preference oracles. In fact, some information, such as the fraction of preference oracles

preferring one trajectory over the other, is used by many baselines like `DPO` and `RM+PPO`, but not `ZSPO`.

**Policy Network.** The policy network that agents train to make decisions is a fully connected neural network with two hidden layers. Each hidden layer has $64$ neurons, and depending on the input and output dimensions, the total number of parameters is around $6,000$. We conducted a full training on the actor network and did not freeze parameters. We used the hyperbolic tangent function as an activation function and initialized the parameters of the neural network from a standard normal distribution. Then, we normalized the output of each layer to balance the smoothness constant for each parameter. For environments with finite action spaces like `CartPole-v1`, we used a softmax layer to convert logits to the policy, i.e., the probability of taking each action. For environments with continuous action spaces like `HalfCheetah-v5` and `Hopper-v5`, we used a normal distribution as the policy class, where the mean is the output logits and the standard deviation is another set of trainable parameters. The `RM+PPO` baseline requires two extra architectures, a reward model and a critic network. For the reward model, we also used a fully connected neural network with two hidden layers of $64$ neurons, similar to the size of the actor, where the input is the concatenation of a state vector and an action vector, and the output is a numeric reward. The state critic network has a similar structure as well, i.e., two hidden layers with $64$ neurons, where the input is a state vector and the output is a numeric value. To reduce computation and more efficiently demonstrate the performance difference between algorithms, in the `CartPole-v1` environment, we pre-trained the policy network using the environmental reward until it lasted for around $40$ time steps, i.e., with a cumulative reward of around $40$. Then, we warm-started all algorithms from the same pre-trained policy network. Notice that the maximum possible reward is $500$, which is much larger than the starting point, and `ZSPO` almost reaches the maximum reward in all runs. We believe the warm-start should not affect the implications and takeaways of the empirical studies. For other environments, we start training all algorithms from the initialization policy.

**Training and Interaction Protocol.** To have a fair comparison, we let online preference-based algorithms conduct $500$ policy iteration steps on the `CartPole-v1` and the `HalfCheetah-v5` environments, i.e., $T = 500$, and conducted $1,000$ policy iterations on the `Hopper-v5` environment, i.e., $T = 1,000$. This is because the `Hopper-v5` environment is much noisier since the agent could very easily fall to the ground and terminate the episode. For each policy iteration, we let algorithms sample a pair of trajectories and obtain the pair-wise preferences from all oracles. However, we let the baseline `RM+PPO` and `DPO` conduct multiple rounds of loss optimization in each policy iteration. The `RM+PPO` algorithm is not fully online and requires additional samples to train the reward model. To ensure fairness, we allow it to sample half of the total number of trajectories ahead of time to train the reward model, and then use the same number of policy iterations and sampled trajectories in the actor network training phase. That is to say, we separated the total number of samples into two parts of equal size: one for training the reward model and one for training the policy network. We used $10$ different random seeds to test the robustness of all algorithms, i.e., $[33, 81, 34, 44, 42, 41, 31, 173, 139, 83]$. The running return with a window size from $25$ to $35$ is reported in Fig. 1, and the return of the best policy for these ten seeds is also demonstrated in Tab. 1.

**Hyperparameters and Implementation of `ZSPO`.** Two hyper-parameters need to be fine-tuned in practice for `ZSPO`: the learning rate $\alpha_t$ and the perturbation distance $\mu_t$. Depending on the environments being tested, we set the perturbation distance $\mu_t$ to be time-homogeneous and vary from $0.1$ to $0.3$. In each policy iteration, `ZSPO` perturbs the actor network based on a high-dimensional normal noise and then samples one trajectory for the perturbed and unperturbed actor network, respectively. To improve the empirical performance of `ZSPO`, demonstrate its flexibility for modern optimization methods, and showcase its practical potential, we incorporate `ZSPO` with the `AdamW` optimizer with a starting learning rate around $0.01$, depending on the environments being tested. We first use `ZSPO` to estimate the ascent direction $\hat{g}_t$ as in Algorithm 1, and then replace the gradient in the `AdamW` optimizer with the estimation $\hat{g}_t$. After that, we use the `AdamW` optimizer to conduct gradient ascent. This simple modification helps `ZSPO` to reuse past estimations of the ascent direction and improve its sample complexity compared to the vanilla stochastic ascent paradigm.

**Implementation of Baseline Algorithms.** For the `ZPG` baseline, to ensure fairness among comparisons, we perturbed the actor network and sample trajectories the same way as `ZSPO`. We also incorporated the `AdamW` optimizer with the zeroth-order gradient estimator and fine-tuned the perturbation distance and the starting learning rate. For the `ES` baseline, we only need to fine-tune the perturbation distance. Then, we perturbed the actor network and sampled trajectories the same way as

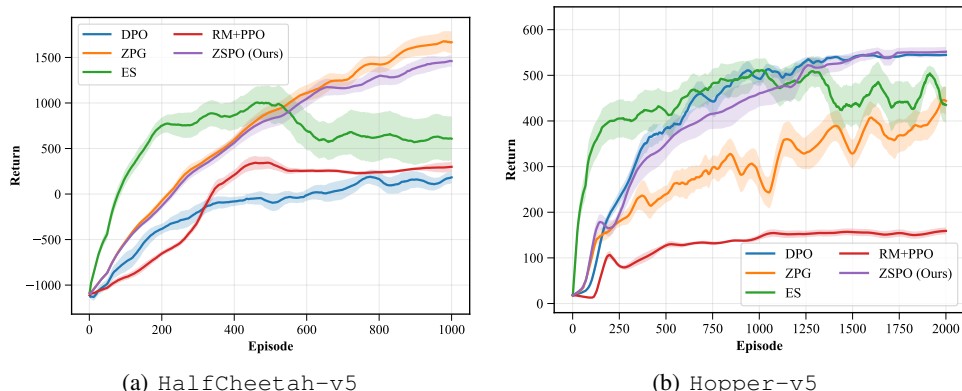

(a) `HalfCheetah-v5`  (b) `Hopper-v5`

Figure 3: Ablation studies on `HalfCheetah-v5` and `Hopper-v5` with correctly specified link function. Average running returns (mean $\pm$ std) is reported.

`ZSPO`. For the `Online DPO` baseline in each policy iteration, we used the current policy to sample a pair of trajectories and obtained the fraction of preference oracles preferring one trajectory over the other. Then, we constructed the DPO loss using this fraction instead of the pure zero-one feedback to reduce variance. We used the `AdamW` optimizer and fine-tuned the starting learning rate. For the `RM+PPO` baseline, we first trained an exploratory policy to collect trajectories and preferences to train the reward model. Notice that, similar to our `Online DPO` implementation, we used the fraction of preference oracles preferring one trajectory over the other instead of the zero-one feedback to construct the likelihood loss to train the reward model. The method of obtaining feedback from multiple humans, instead of one, to learn the reward model is exactly the classic method proposed in Christiano et al. (2017) as one of the earliest papers studying RLHF. Then, we used the reward model and fine-tuned the learning rate of `PPO`, with the clip variant. The other hyperparameters of `PPO` were set similarly to the standard benchmarks in `Gymnasium`.

**Additional Ablation Studies on Link Function.** To facilitate the importance and emphasize the effect of link function mis-specification in real-world RL tasks, we conducted further ablation studies on `HalfCheetah-v5` and `Hopper-v5` where the preferences are generated from the Bradley-Terry model, i.e., the same link function used by baseline algorithms. It could be observed that the performance of baselines improves with the correctly specified preference model, and some even perform better than `ZSPO`. This demonstrates the universality of influences in mis-specified link functions, similar to mis-specified reward functions. Moreover, it also strengthens our argument that `ZSPO` is a robust algorithm under different link functions.

## C.2 Stochastic GridWorld Environment Experiments

To demonstrate the hardness of MDPs with stochastic transitions and test the empirical performance of `ZSPO` under link function mismatch in such environments, we considered a stochastic `GridWorld` environment in (Zhang & Ying, 2025) with $H = 10$. We tested the empirical performance under different unknown link functions, i.e., logistic and linear (Chen & Frazier, 2017; Bengs et al., 2021), similar to our main robotic experiments in Sec. 5.

**Environment.** The environment has $5 \times 5$ blocks, denoted as $(1, 1)$ to $(5, 5)$. For each block, with probability $1/2$, a random reward is sampled from a standard normal distribution and is assigned as the reward of the state. So on average, half of the blocks will have a reward $0$. Each episode consists of $H = 10$ steps, and at the start of each episode, an agent is positioned in block $(3, 3)$, i.e., the center of the `GridWorld` environment. At each step, the agent can choose to go up, down, left, or right. However, the action may be changed due to environmental disturbances. Each state has a disturbance action probability distribution over the four actions, which is randomly generated. For each action taken, with probability $1/2$ the state will transition according to the selected action, and with probability $1/2$ the action will be re-chosen according to the disturbance action probability distribution. The motivation for imperfect control arises naturally from designing agents for turbulent

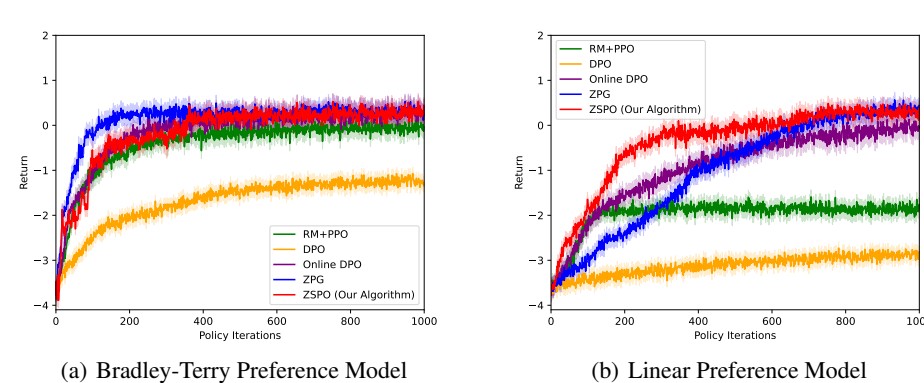

(a) Bradley-Terry Preference Model          (b) Linear Preference Model

Figure 4: GridWorld: (a) comparison of `ZSPO` and baselines without link function mismatch, and (b) comparison of `ZSPO` and baselines with link function mismatch. Results are averaged over $10^3$ repetitions, and shaded areas are $95\%$ confidence intervals.

environments, where wind or a bump may shift the agent's action. The goal of the agent is to maximize the cumulative reward, and the interaction is conducted episodically.

**Preference Feedback.** In this experiment, we also assumed access to 100 oracles. Each will generate a preference either based on the standard Bradley-Terry model or based on a linear link function over the trajectory rewards, depending on the underlying unknown preference model. We used $\gamma = 1$ for the Bradley-Terry model, and $\gamma = 0.02$ for the linear model. We also assumed $D = 1$ in this experiment to study the effect of distinguishability: the oracles only compared a pair of trajectories.

**Policy Parameterization.** For all algorithms implemented in our experiment, we used tabular policy softmax parameterization, i.e., each state-action pair $(s, a)$ is equipped with a parameter $\xi_{s,a}$ and the policy $\pi(a|s)$ of taking action $a$ at state $s$ would follow:

$$\pi(a|s) = \frac{\exp(\xi_{s,a})}{\sum_a \exp(\xi_{s,a})}.$$

**Baseline Algorithms.** We also considered four baselines: (1) `RM+PPO` (Ouyang et al., 2022), (2) `DPO` (Rafailov et al., 2024), (3) `Online DPO` (Dong et al., 2024; Guo et al., 2024), and (4) `ZPG` (Zhang & Ying, 2025), where all of them require a known link function to be implemented. Therefore, we let them assume a logistic link function. Note that both `DPO` and `Online DPO` are only designed for deterministic MDPs. In this experiment, due to the noisy reward signal from both the stochastic transition kernel and the heterogeneous reward, all algorithms collected $N = 1,000$ trajectories between policy updates and then queried the preference oracles to evaluate each trajectory. For `ZSPO`, the preferences for different pairs of trajectories and from different oracles were aggregated via a majority vote. The implementation details are similar to the ones used in our robotic experiments as specified in Sec. C.1.

**Performance.** We first compare `ZSPO` to the baselines in the Bradley-Terry model so that the underlying preference link function matches the one used by baselines, as shown in Fig. 4(a). In this setting, all algorithms know the true link function, *except ours*. We observe that even though our proposed `ZSPO` does not know the link function and may suffer from the distinguishability, it has almost the same performance as the best baselines. This confirms the correctness of our design, i.e., `ZSPO` continues to improve the policy with sign-information of value-function difference and does not need to explicitly know or learn the link function. This demonstrates the robustness of `ZSPO` in stochastic general RL problems.

**Link Function Mismatch.** We then experimented with the setting with a link function mismatch, i.e., the true link function is linear while the baseline algorithms adopt a logistic link function, as shown in Fig. 4(b). It shows that `ZSPO` (our algorithm) is more robust compared to the baselines and converges quickly. `DPO` has a poor performance in both cases since the KL regularization to the reference policy restrains the possibility of finding a near-optimal policy. When a link function mismatch exists, `DPO`

and `PPO` suffer severely from the mismatch, where the return of the learned policy is much inferior compared to Fig. 4(a). This is because the intermediate reward, either implicitly or explicitly learned from the preference with a mis-specification, deviates from the true reward function, and thus shifts the optimal policy learned from it. `Online DPO` avoids the drawback of the KL regularization, but still suffers from link function mismatch with a worse final policy than that under `ZSPO`. `ZPG`, on the other hand, has a similar final policy return as `ZSPO` and demonstrates some robustness. This is because both algorithms are based on similar ideas, where the information of value function differences is recovered to estimate the gradient via zeroth-order approximation. However, when the preference is generated from a linear function instead of the logistic function, where both are anti-asymmetric, the sign of the value function difference recovered by `ZPG` is still accurate, and the algorithm is equivalent to `ZSPO` with an inappropriate non-stationary learning rate. Thus, the convergence speed of `ZPG` in this setting is much slower than `ZSPO`, which is due to the mismatch of the link function. We note that both `Online DPO` and `DPO`, designed for deterministic MDPs, suffer amplified inherent performance loss when the link function is specified. The observation from the `GridWorld` environment coincides with the `Gymnasium` environment.

# D DISCUSSIONS

In this section, we provide an additional discussion of the proposed `ZSPO` algorithm. We discuss the distinguishability constant $\varepsilon_D^*$, the zeroth-order algorithm itself, and other related aspects.

## D.1 SIGN-BASED ZEROTH-ORDER OPTIMIZATION ALGORITHM

The sign-based estimator has been understudied in the zeroth-order literature. To be more specific and illustrate the novelty, we consider a setting where the value function $V(\pi_{\boldsymbol{\theta}})$ can be directly queried, similar to the non-convex optimization setting. Classic two-point zeroth-order optimization algorithm (Ghadimi & Lan, 2013), i.e., the zeroth-order stochastic gradient descent (`ZO-SGD`) algorithm, perturbs the current parameter $\boldsymbol{\theta}_t$ with a small distance $\mu$ and a randomly sampled vector $\boldsymbol{v}_t$ from a $d$-dimensional normal distribution to obtain the perturbed parameter $\boldsymbol{\theta}_t' = \boldsymbol{\theta}_t + \mu\boldsymbol{v}_t$. Then, it constructs the gradient estimator based on the difference of the two points as follows:

$$\hat{\boldsymbol{g}}_{t,\text{ZO-SGD}} = \frac{V(\pi_{\boldsymbol{\theta}_t'}) - V(\pi_{\boldsymbol{\theta}_t})}{\mu}\boldsymbol{v}_t,$$

and then proceeds with gradient ascent as $\boldsymbol{\theta}_{t+1} = \boldsymbol{\theta}_t + \alpha\hat{\boldsymbol{g}}_{t,\text{ZO-SGD}}$, where $\alpha$ is the learning rate. Researchers have shown that this version of the zeroth-order method converges to the stationary point with a convergence rate $\sqrt{d/T}$ (Nesterov & Spokoiny, 2017) when the learning rate and perturbation distance satisfy $\alpha = \mu = 1/\sqrt{dT}$. Some researchers also studied the zeroth-order sign gradient descent (`ZO-signSGD`) (Liu et al., 2019) to achieve more robustness against adversarial attacks. This algorithm uses the same gradient estimator $\hat{\boldsymbol{g}}_t$ as `ZO-SGD`, but only uses the sign of the gradient estimator to obtain the parameter for the next iteration, i.e., $\boldsymbol{\theta}_{t+1} = \boldsymbol{\theta}_t + \alpha\,\text{sign}[\hat{\boldsymbol{g}}_{t,\text{ZO-SGD}}]$, where the sign operator here takes the sign of each entry and combine them into a vector in the same dimension as $\hat{\boldsymbol{g}}_t$. The algorithm perturbs the current policy multiple times at each iteration, and achieves the same convergence rate as `ZO-SGD` if the number of perturbations is large. Variants of both algorithms have also been studied, such as variance-reduced gradient and ADMM (Liu et al., 2018b;a).

Our algorithm `ZSPO` is different from both approaches. `ZSPO` uses a different gradient estimator, which uses only the sign of the value function difference as follows:

$$\hat{\boldsymbol{g}}_{t,\text{ZSPO}} = \text{sign}\left[V(\pi_{\boldsymbol{\theta}_t'}) - V(\pi_{\boldsymbol{\theta}_t})\right]\boldsymbol{v}_t,$$

where the sign operates on scalars, where $\boldsymbol{v}_t$ is sampled from a random normal distribution. Then, `ZSPO` uses classic ascent procedure to obtain the parameter for the next iteration $\boldsymbol{\theta}_{t+1} = \boldsymbol{\theta}_t + \alpha\hat{\boldsymbol{g}}_{t,\text{ZSPO}}$. It can be observed from our Corollary 2 that when $N \to \infty$ and $D \to \infty$, `ZSPO` achieves the same convergence rate as classic zeroth-order methods. However, `ZSPO` has multiple advantages. First, it only requires the sign of the value function to construct the gradient estimator, which is the reason it can be applied in our preference-based RL problem with an unknown link function. In general, the sign is much easier to obtain than the full difference information. Second, it may be more suitable for distributed optimization settings since only the component related to the value function to be optimized is the sign information, which can be easily transmitted through channels since it

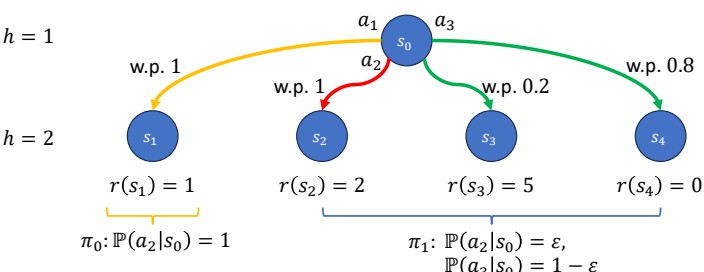

Figure 5: a two-step MDP example: the reward depends on the state at the second planning step with two deterministic actions and one random action. Policy $\pi_0$ selects action $a_1$ deterministically and policy $\pi_1$ randomizes over action $a_2$ and $a_3$. Notice that $V(\pi_1) - V(\pi_0) = \varepsilon \geq 0$.

only consists of a single bit. Moreover, since the gradient estimator is different from the classic zeroth-order stochastic gradient descent, we cannot directly use the smoothing function framework in (Ghadimi & Lan, 2013) to prove its convergence, and therefore, a new framework is developed.

### D.2 DISTINGUISHABILITY FROM PREFERENCE ORACLE

In this section, we discuss the distinguishability constant $\varepsilon_D^*$ for batches of trajectories with size $D$. Specifically, we discuss the regularity of the preference model, the policies to be compared, and the underlying RL problem that affects the distinguishability constants, so that the expected preference based on batches of trajectories aligns with the value function difference.

#### D.2.1 AN EXAMPLE TO DEMONSTRATE FACTORS INFLUENCING DISTINGUISHABILITY

We consider an RL problem example with planning step $H = 2$ and two policies $\pi_0$ and $\pi_1$ as shown in figure 5. The first step $h = 1$ has only one state $s_0$ with three actions $\{a_1, a_2, a_3\}$ and the MDP will always initialize to $s_0$. The second step consists of four states $\{s_1, s_2, s_3, s_4\}$, and each state only has one action. The true reward of the MDP depends on the state on the second planning step, and since the planning step $H = 2$, the transition only happens when the agent chooses an action at state $s_0$ in the first planning step $h = 1$. If action $a_1$ is chosen in $s_0$, the state deterministically transits to $s_1$ with reward $r(s_1) = 1$. If action $a_2$ is chosen, the state deterministically transits to $s_2$ with reward $r(s_2) = 2$. If action $a_3$ is chosen, the state transits to $s_3$ with probability 0.2 and reward $r(s_3) = 5$, or it transits to $s_4$ with probability 0.8 and reward $r(s_4) = 0$.

For simplicity of demonstration, we assume the link function $\sigma(\cdot)$ is the 0-1 step function, i.e., the oracles would deterministically prefer the trajectory batch with a larger average return. Even though this link function is not smooth or strictly increasing, we can construct a series of smooth and strictly increasing link functions, such as logistic functions, to approximate it, and the step function is the limit of such a series of approximating functions. We consider two policies $\pi_0$ and $\pi_1$, where the only difference is the strategy on the first planning step when choosing actions at state $s_0$. Policy $\pi_0$ chooses $a_1$ deterministically, so the state will always transit to $s_1$. Policy $\pi_1$ randomizes between $a_2$ and $a_3$. Specifically, it chooses $a_2$ on state $s_0$ with probability $\varepsilon$ and chooses action $a_3$ with probability $1 - \varepsilon$. We also assume the batch size $D = 1$, i.e., for each policy, we sample one trajectory $\tau_0 \sim \pi_0$ and $\tau_1 \sim \pi_1$ and then query a oracle for feedback.

We can easily verify that the value function of $\pi_0$ is $V(\pi_0) = 1$ since it is the state that deterministically transits to $s_1$ with reward 1. For policy $\pi_1$, we can also calculate its value function:

$$V(\pi_1) = \mathbb{P}(a_2|s_0) \mathbb{P}(s_2|a_2, s_0) r(s_2) + \mathbb{P}(a_3|s_0) (\mathbb{P}(s_3|a_3, s_0) r(s_3) + \mathbb{P}(s_4|a_3, s_0) r(s_4))$$
$$= 1 + \varepsilon.$$

Therefore, policy $\pi_1$ would have a larger value function and therefore a better performance than policy $\pi_0$ with $V(\pi_1) - V(\pi_0) = \varepsilon \geq 0$. We would also hope that the preference probability $\mathbb{P}(\tau_1 \succ \tau_0) \geq 0.5$ since $\tau_1$ is generated from the better policy $\pi_1$ and $\tau_0$ is generated by the worse

policy. We could characterize this event, which is the event that the state transits to $s_2$ or $s_3$ after taking action $a_3$ following policy $\pi_1$, which happens with probability:

$$\mathbb{P}(\tau_1 \succ \tau_0 | \tau_0 \sim \pi_0, \tau_1 \sim \pi_1) = \mathbb{P}(a_2|s_0)\mathbb{P}(s_2|a_2, s_0) + \mathbb{P}(a_3|s_0)\mathbb{P}(s_3|a_3, s_0) = 0.8 \cdot \varepsilon + 0.2.$$

Therefore, when $\varepsilon$ is large, i.e., when $\varepsilon > 3/8$, the two policies $\pi_1$ and $\pi_0$ will be distinguishable for oracles, since the preference probability $\mathbb{P}(\tau_1 \succ \tau_0)$ is larger than half. However, when $\varepsilon$ is small, the two policies may not be distinguishable since the oracles in expectation may prefer the trajectory generated from the worse policy $\pi_0$, and the expected preference over the trajectories is contradictory to the true value function, which measures the authentic quality of the policy. Therefore, we have the distinguishability $\varepsilon_1^* \geq 3/8$. This example demonstrates that the distinguishability constant from oracles could be non-zero, and it comes from the asymmetric randomness of the policy, reward of each state, and MDP transitions. Notice that if we change the link function from a step function to the standard logistic function, we can also verify that to obtain $\mathbb{P}(\tau_1 \succ \tau_0) > 0.5$, it would require $\varepsilon > 0.277$. So the distinguishability constant $\varepsilon_D^*$ is dependent on the link function $\sigma(\cdot)$.

### D.2.2 Characterization of Distinguishability Constant

If the distinguishability constant $\varepsilon_D^*$ is positive, the oracles may not be able to distinguish two policies based on the trajectories sampled from them, which causes mistakes. Therefore, when we implement ZSPO, which uses trajectory batches and oracle preferences to approximate the sign of the value function, we need to be cautious in choosing the perturbation, so that the perturbed policy and the original policy are distinguishable. Therefore, we must understand this constant $\varepsilon_D^*$ and the factors that influence it. In general, the constant $\varepsilon_D^*$ depends on multiple factors such as (i) the transition kernel and reward of the MDP, (ii) the link function, and (iii) the batch size used to query oracles. We emphasize that even though our empirical experiments seem to suffice to only use $D = 1$ and ignore the effect of the distinguishability constant, this is because either the inherent distinguishability is small in these experiments due to the nice and regular structure of the problem with low reward noise, or the agent's policy is still far from optimal, which the effect of distinguishability has not yet kicked in. Nonetheless, this inherent difficulty of preference-based RL under the unknown link function should be recognized and studied due to its important practical implications, as preferences may not offer the correct direction to improve the policy.

**RL Problem.** The limit of distinguishability in the example of figure 5 results from the fact that even though action $a_3$ has the same expected reward as action $a_1$, the probability of obtaining a reward larger than $r(s_1)$ is much smaller than half. In most of the trajectories, the state will transit to $s_4$ with a reward $r(s_4) = 0$, and thus the policy would be less preferred by the oracles. On the other hand, we can revise the transition kernel and reward function so that under action $a_3$, the MDP would transit to state $s_3$ with probability $0.5$ and reward $r(s_3) = 2$, and transits to $s_4$ with probability $0.5$ and reward $r(s_4) = 0$. Then, the probability of preferring the trajectory generated by $\pi_1$ would become $\mathbb{P}(\tau_1 \succ \tau_0) = 0.5(1 + \varepsilon)$ which is larger than half for all $\varepsilon$. Then, when only comparing these two classes of policies, the limit of distinguishability would be $0$ and the preference of the trajectory batches exactly reflects the relationship between value functions.

**Link Function.** Moreover, if we switch the 0-1 step link function to the logistic function under the Bradley-Terry model, the minimum $\varepsilon$ for the probability $\mathbb{P}(\tau_1 \succ \tau_0)$ to reflect the value function difference, i.e., $\mathbb{P}(\tau_1 \succ \tau_0) \geq 1/2$, will also be different, i.e., $0.277$ versus $0.375$. Furthermore, if the link function $\sigma(\cdot)$ is linear on the interval $[-H, H]$, then according to definition 1, we could push the expectation inside the deviation function as follows:

$$\begin{aligned}
\mathbb{E}_{\mathcal{D}_0, \mathcal{D}_1}\left[\varsigma\left(\bar{r}\left(\mathcal{D}_1\right) - \bar{r}\left(\mathcal{D}_0\right)\right)\right] &= \varsigma\left(\mathbb{E}_{\mathcal{D}_1}\left[\bar{r}\left(\mathcal{D}_1\right)\right] - \mathbb{E}_{\mathcal{D}_0}\left[\bar{r}\left(\mathcal{D}_0\right)\right]\right) \\
&= \varsigma\left(V(\pi_1) - V(\pi_0)\right) \\
&\geq \frac{1}{2}\varsigma\left(\frac{V(\pi_1) - V(\pi_0)}{2}\right).
\end{aligned}$$

Therefore, for batch size $D$, the limit of distinguishability $\varepsilon_D^* = 0$ and the expected oracle preference will always point to the policy with a larger value function. In general, for the same batch size $D$, the distinguishability constant $\varepsilon_D^*$ measures the skewness of the link function $\sigma(\cdot)$ compared to the linear function. When the link function is closer to a step function, which is highly non-linear, $\varepsilon_D^*$ may be larger. If the link function is more linear, then $\varepsilon_D^*$ is likely to be smaller since the linearization of the deviation function would be close to the deviation function itself. If we recall the convergence

rate of ZSPO in Theorem 1, the link function controls the trade-off between the distinguishability and the majority vote approximation error. Specifically, to achieve a better approximation error, one would hope the oracles possess a link function which is very close to a step function, so that for the same number of trajectory batches $N$, the approximation error would be smaller. This link function is likely to be non-linear. However, to reduce the distinguishability error, one would prefer a linear link function over the aforementioned non-linear close-to-step function since it helps to estimate the sign of the value function from the oracle's preference.

**Batch Size.** The limit of distinguishability $\varepsilon_D^*$ is dependent on the batch size $D$ of trajectories that we use to query oracles for comparison results. In general, if we consider a pair of trajectory batches $(\mathcal{D}_1, \mathcal{D}_0)$ generated from $\pi_1$ and $\pi_0$ in figure 5 respectively with large enough batch size $D$, the average return $\bar{r}(\mathcal{D}_1)$ of policy $\pi_1$ would be concentrated around its expectation $V(\pi_1)$ with a distribution converging to a normal distribution. Since the return of policy $\pi_0$ is deterministic and $V(\pi_1) - V(\pi_0) = \varepsilon > 0$, by the symmetric nature of the normal distribution, the probability that the oracle prefers batch $\mathcal{D}_0$ over $\mathcal{D}_1$ will be less than half, because the average return $\bar{r}(\mathcal{D}_1) < \bar{r}(\mathcal{D}_0)$. Even though increasing the batch size $D$ is generally helpful to decrease the distinguishability constant $\varepsilon_D^*$, the oracles may have an upper limit for the number of trajectories to aggregate and compare at the same time. This implies that if we want to obtain a better policy, we should also employ better oracles with a larger capacity to aggregate trajectories.

**Structure of Policies.** If the policy $\pi_1$ in figure 5 is a randomization between action $a_1$ and $a_2$ instead of the randomization between action $a_2$ and $a_3$, the probability $\mathbb{P}(\tau_1 \succ \tau_2)$ will be larger than half for all $\varepsilon > 0$. This demonstrates that the structure of the policies being compared by the oracle influences whether one can distinguish a better policy by comparing trajectories. However, in our proposed algorithm ZSPO, we intend to compare the value function between the perturbed policy $\pi_{\theta_t'}$ and the current policy $\pi_{\theta_t}$ using preference feedback, and the structure of the two policies could be arbitrary. Therefore, the convergence result will be dependent on the distinguishability for the most difficult pair of policies, as defined in definition 1.

## D.3 OTHER ASPECTS AND LIMITATIONS

**Distinguishability in Convergence Rate.** Due to the approach that ZSPO uses oracles, i.e., use trajectories generated by different policies to query oracles to infer which policy has a larger value function, the method we propose suffers from the oracle distinguishability limit $\varepsilon_D^*$, which limits the convergence rate of the proposed ZSPO algorithm. The influence may depend on multiple factors that the RL agent cannot control, as specified in section D.2. These factors may include the underlying RL problem itself, the true reward function, and the nature of the oracles being used. To mitigate it, we could either employ oracles of higher quality and let them compare larger batches of trajectories, or we could regularize the policy being explored so that the comparison suffers less from this distinguishability. It also remains an open question whether the way ZSPO uses the preference oracles is the optimal approach to train RL agents from preference feedback.

**Local Convergence.** Our main result in Theorem 1 states the convergence rate for ZSPO to a stationary policy instead of the global optimal policy. The results can be extended to global convergence under regular assumptions in optimization, such as convexity (Boyd & Vandenberghe, 2004) or the Polyak-Łojasiewicz (PL) condition (Karimi et al., 2016). However, these assumptions usually do not hold in reality when general function approximation is considered. It remains an interesting question whether the global convergence of ZSPO can be proved without these assumptions, even for tabular parameterization as in (Mei et al., 2020). We conjecture that combining the analysis in this paper with the convergence analysis of classic policy gradient algorithms, the global convergence rate of ZSPO for tabular parameterization could be proved.

**Practicality in Real-World RL Tasks.** It has been widely believed that zeroth-order optimization methods are not efficient in real-world experiments, since their complexity is usually dependent on the dimension of the parameters, which could be huge in real-world tasks such as large language model finetuning, let alone ZSPO only uses a one-bit feedback. This is observed in tasks that arise in many machine learning fields. However, as counterintuitive as it may be, recent advances such as Mezo (Malladi et al., 2023) have shown the opposite that zeroth-order optimization could also achieve competitive performance in these complex tasks, with the merits of memory and computational efficiency without the need for backpropagation, as long as a parameter-efficient fine-tuning or

training framework is used to efficiently decrease the effective dimension of the optimization problem. This is also partly because the dependence on the number of parameter dimensions is based on the worst-case optimization instance, but practical problems usually have much benign landscapes. These tasks are over-parameterized, and a low-rank finetuning of the full model is sufficient to achieve competitive performance, so the effective dimension of the parameters is much lower. Moreover, in other real-world tasks involving preference-based reinforcement learning, such as autonomous driving and human-robot co-operation, or experiments in online platforms and developments, the gradient of a meaningful objective function simply cannot be obtained without restrictive assumptions, and therefore, one cannot hope to use first-order methods. The same situation is encountered with preference-based RL when the mechanism of preference generation is unknown or variable over time. In these cases, zeroth-order methods such as ZSPO are the only resolution that could make things work. Moreover, as shown in the empirical experiments in this paper, accurately estimating the gradient is usually not necessary to optimize the policy. A policy improvement direction correlated with the gradient direction is empirically enough, as we only use a single pair of trajectories for each policy update in some of our experiments in section 5. Therefore, we conjecture that the full potential of zeroth-order methods in practice is still largely underexplored. We consider evaluating the performance of ZSPO with real preference feedback in real-world tasks like robotics and language models as our future direction.

## E   PROOF OF PROPOSITION 1

We first define the following positive constant:

$$\varepsilon_0 \equiv \frac{4H}{\sqrt{D}} \sqrt{2 \log \left( \frac{2}{\varsigma \left( H/\sqrt{D} \right)} \right)}. \tag{6}$$

Let us consider two arbitrary policies $\pi_0$ and $\pi_1$. Without loss of generality, we assume the value function $V(\pi_1)$ is larger than the other policy $V(\pi_0)$, and their difference is larger by $\varepsilon_0$, i.e., $V(\pi_1) - V(\pi_0) \geq \varepsilon_0$. Let $\mathcal{D}_0$ be a batch of trajectories generated from policy $\pi_0$, and let $\mathcal{D}_1$ be a batch of trajectories generated from policy $\pi_1$ with $|\mathcal{D}_0| = |\mathcal{D}_1| = D$. Since the reward function $r(\cdot)$ is bounded in $[0, H]$, we obtain that $\bar{r}(\mathcal{D}_1) - \bar{r}(\mathcal{D}_0)$ is a sub-Gaussian random variable. Notice that:

$$\mathbb{E}[\bar{r}(\mathcal{D}_1) - \bar{r}(\mathcal{D}_0)] = V(\pi_1) - V(\pi_0).$$

For simplicity, define the random variable $\Delta$ to be the difference of the empirical reward as follows:

$$\Delta = \bar{r}(\mathcal{D}_1) - \bar{r}(\mathcal{D}_0) = \frac{1}{D} \sum_{\tau \in \mathcal{D}_1} r(\tau) - \frac{1}{D} \sum_{\tau \in \mathcal{D}_0} r(\tau).$$

Now, we analyze the expectation of the deviation function for the empirical reward difference in definition 1. First, from assumption 1, the link function is monotonically increasing, so the deviation function is also monotonically increasing, and we have:

$$\mathbb{E}\left[\varsigma(\Delta)\right]$$
$$= \mathbb{E}\left[\varsigma(\Delta) \mathbb{1}_{\Delta \geq \frac{V(\pi_1) - V(\pi_2)}{2}}\right] + \mathbb{E}\left[\varsigma(\Delta) \mathbb{1}_{\Delta \leq \frac{V(\pi_1) - V(\pi_2)}{2}}\right]$$
$$\geq \varsigma\left(\frac{V(\pi_1) - V(\pi_0)}{2}\right) \mathbb{P}\left(\Delta \geq \frac{V(\pi_1) - V(\pi_0)}{2}\right) + \varsigma(-H) \mathbb{P}\left(\Delta \leq \frac{V(\pi_1) - V(\pi_0)}{2}\right)$$
$$\geq \varsigma\left(\frac{V(\pi_1) - V(\pi_0)}{2}\right) \mathbb{P}\left(\Delta \geq \frac{V(\pi_1) - V(\pi_0)}{2}\right) - \frac{1}{2} \mathbb{P}\left(\Delta \leq \frac{V(\pi_1) - V(\pi_0)}{2}\right),$$

where the last inequality uses the fact that the deviation function is bounded below by $-1/2$. By Hoeffding's inequality, we can use concentration to characterize the probability that the empirical reward difference deviates from the value function difference as follows:

$$\mathbb{P}\left(\Delta \leq \frac{V(\pi_1) - V(\pi_0)}{2}\right) \leq \mathbb{P}\left(\bar{r}(\mathcal{D}_1) - \bar{r}(\mathcal{D}_0) - V(\pi_1) + V(\pi_0) \leq -\frac{\varepsilon_0}{2}\right) \leq \exp\left(-\frac{D\varepsilon_0^2}{8H^2}\right),$$

where the first inequality uses the fact that $V(\pi_1) - V(\pi_0) \geq \varepsilon_0$. So we can lower bound the expectation of the deviation function as follows:

$$\mathbb{E}\left[\varsigma\left(\Delta\right)\right] \geq \left(1 - \exp\left(-\frac{D\varepsilon_0^2}{8H^2}\right)\right) \varsigma\left(\frac{V(\pi_1) - V(\pi_0)}{2}\right) - \frac{1}{2}\exp\left(-\frac{D\varepsilon_0^2}{8H^2}\right). \tag{7}$$

Notice that by the construction of $\varepsilon_0$ in equation 6 and the fact that the deviation function $\varsigma(\cdot)$ is upper bounded by $1/2$, we can lower bound $\varepsilon_0$ by:

$$\varepsilon_0 = \frac{4H}{\sqrt{D}}\sqrt{2\log\left(\frac{2}{\varsigma\left(H/\sqrt{D}\right)}\right)} \geq \frac{4H}{\sqrt{D}}\sqrt{2\log 4} \geq \frac{4H}{\sqrt{D}}. \tag{8}$$

Since the value function difference $V(\pi_1) - V(\pi_0)$ is positive, the deviation function in the first term of equation 7 is positive and therefore the first term is lower bounded as follows:

$$\left(1 - \exp\left(-\frac{D\varepsilon_0^2}{8H^2}\right)\right) \varsigma\left(\frac{V(\pi_1) - V(\pi_0)}{2}\right) \geq \left(1 - e^{-2}\right)\varsigma\left(\frac{V(\pi_1) - V(\pi_0)}{2}\right)$$

$$\geq \frac{3}{4}\varsigma\left(\frac{V(\pi_1) - V(\pi_0)}{2}\right).$$

On the other hand, for the second term of equation 7, we want to provide an upper bound and relate it to the deviation function of the value function difference. Therefore, we first characterize a slightly tighter lower bound on $\varepsilon_0^2$ as follows:

$$\varepsilon_0^2 = 32\frac{H^2}{D}\log\left(\frac{2}{\varsigma\left(H/\sqrt{D}\right)}\right) \geq 8\frac{H^2}{D}\log\left(\frac{2}{\varsigma\left(H/\sqrt{D}\right)}\right).$$

The last inequality is because the deviation function $\varsigma(\cdot)$ is upper bounded by $1/2$ and therefore the element inside the logarithm is lower bounded by $4$. Therefore, the logarithm is positive, and we can divide the coefficient by $4$. This implies:

$$\frac{1}{2}\exp\left(-\frac{D\varepsilon_0^2}{8H^2}\right) \leq \frac{1}{2}\exp\left(\log\left(\frac{\varsigma\left(H/\sqrt{D}\right)}{2}\right)\right) = \frac{1}{4}\varsigma\left(\frac{H}{\sqrt{D}}\right).$$

Again, as shown in the equation 8, we have:

$$\frac{H}{\sqrt{D}} \leq \frac{\varepsilon_0}{2}.$$

By the monotonicity of the preference deviation function $\varsigma(\cdot)$, we can upper bound the second term in equation 7 as follows:

$$\frac{1}{2}\exp\left(-\frac{D\varepsilon_0^2}{8H^2}\right) \leq \frac{1}{4}\varsigma\left(\frac{H}{\sqrt{D}}\right) \leq \frac{1}{4}\varsigma\left(\frac{\varepsilon_0}{2}\right) \leq \frac{1}{4}\varsigma\left(\frac{V(\pi_1) - V(\pi_0)}{2}\right),$$

where the last inequality uses our assumption that $V(\pi_1) - V(\pi_0) \geq \varepsilon_0$. Therefore, substitute the bound for both terms back to equation 8, and we have:

$$\mathbb{E}\left[\varsigma\left(\bar{r}\left(\mathcal{D}_1\right) - \bar{r}\left(\mathcal{D}_0\right)\right)\right] \geq \frac{3}{4}\varsigma\left(\frac{V(\pi_1) - V(\pi_0)}{2}\right) - \frac{1}{4}\varsigma\left(\frac{V(\pi_1) - V(\pi_0)}{2}\right)$$

$$\geq \frac{1}{2}\varsigma\left(\frac{V(\pi_1) - V(\pi_0)}{2}\right).$$

Therefore, for any two policies $\pi_1$ and $\pi_0$ whose value function is separated by $\varepsilon_0$, the requirement presented in definition 1 is satisfied, which means the limit of distinguishability $\varepsilon_D^*$ should be even smaller. Then, we can conclude:

$$\varepsilon_D^* \leq \varepsilon_0 = \frac{4H}{\sqrt{D}}\sqrt{2\log\left(\frac{2}{\varsigma\left(H/\sqrt{D}\right)}\right)}.$$

According to the link function smoothness assumption in assumption 2, we have the deviation function $\varsigma(\cdot)$ is also smooth with the same constant $L$. Then, when $D \geq L^2 H^2 / (\sigma'(0))^2$, we have:

$$\varsigma\left(\frac{H}{\sqrt{D}}\right) = \varsigma\left(\frac{H}{\sqrt{D}}\right) - \varsigma(0) \geq \frac{\sigma'(0)H}{\sqrt{D}} - \frac{LH^2}{2D} \geq \frac{\sigma'(0)H}{2\sqrt{D}}.$$

This implies the distinguishability $\varepsilon_D^*$ is upper bounded by:

$$\varepsilon_D^* = \mathcal{O}\left(\frac{H}{\sqrt{D}}\sqrt{\log\left(\frac{D}{\sigma'(0)H}\right)}\right).$$

On the other hand, when we have $D \leq L^2 H^2 / (\sigma'(0))^2$, the smoothness approximation is vacuous, and by the monotonicity of the preference deviation function, we have:

$$\varsigma\left(\frac{H}{\sqrt{D}}\right) \geq \varsigma\left(\frac{\sigma'(0)}{L}\right).$$

This implies the distinguishability $\varepsilon_D^*$ is upper bounded by:

$$\varepsilon_D^* = \mathcal{O}\left(\frac{H}{\sqrt{D}}\sqrt{\log\left(\frac{1}{\varsigma(\sigma'(0)/L)}\right)}\right).$$

Combining both bounds, we conclude that:

$$\varepsilon_D^* = \mathcal{O}\left(\frac{H}{\sqrt{D}}\sqrt{\log\left(\max\left\{\frac{1}{\varsigma(\sigma'(0)/L)}, \frac{D}{\sigma'(0)H}\right\}\right)}\right).$$

# F    PROOF OF THEOREM 1

In this section, we prove our main result on the convergence rate of ZSPO. Before we start, we first provide a few lemmas that will be important for the proof.

## F.1    FUNDAMENTAL LEMMAS

We first present and prove a few fundamental lemmas that will be repeatedly used in the proof of Theorem 1. Since the perturbation vector $v_t$ is sampled from a normal distribution in $\mathbb{R}^d$, we have the following two lemmas to characterize the Euclidean norm:

**Lemma 1** *Let $v \in \mathbb{R}^d$ be a random vector sampled from a $d$-dimensional normal distribution $\mathcal{N}(\mathbf{0}, \boldsymbol{I}_d)$, then we have:*

$$\mathbb{E}\left[\|\boldsymbol{v}\|_2^2\right] = d.$$

**Proof.** Let $v = [v_1, v_2, \cdots, v_d]$. Since the vector $v$ is sampled from $\mathcal{N}(\mathbf{0}, \boldsymbol{I}_d)$, its coordinates $v_1$, $v_2, \cdots, v_d$ are independent of each other and follow the same standard normal distribution $\mathcal{N}(0, 1)$. Therefore, we would have:

$$\mathbb{E}\left[\|\boldsymbol{v}\|_2^2\right] = \mathbb{E}\left[\sum_{i=1}^d v_i^2\right] = d\mathbb{E}\left[v_1^2\right] = d,$$

where the second last step comes from the identically distributed nature of the coordinates, and the last equality is because each coordinate is zero-mean with unit variance. ∎

**Lemma 2** *Let $v \in \mathbb{R}^d$ be a random vector sampled from a $d$-dimensional normal distribution $\mathcal{N}(\mathbf{0}, \boldsymbol{I}_d)$, and let $\boldsymbol{a} \in \mathbb{R}^d$ be any deterministic vector, we have:*

$$\mathbb{E}\left[|\langle \boldsymbol{v}, \boldsymbol{a}\rangle|\right] = \sqrt{\frac{2}{\pi}}\|\boldsymbol{a}\|_2.$$

**Proof.** Let $\boldsymbol{v} = [v_1, v_2, \cdots, v_d]$ and $\boldsymbol{a} = [a_1, a_2, \cdots, a_d]$. Since the vector $\boldsymbol{v}$ is sampled from $\mathcal{N}(\boldsymbol{0}, \boldsymbol{I}_d)$, its coordinates $v_1, v_2, \cdots, v_d$ are independent of each other and follow the same standard normal distribution $\mathcal{N}(0, 1)$. Therefore, we have:

$$\langle \boldsymbol{v}, \boldsymbol{a} \rangle = \sum_{i=1}^{d} a_i v_i \sim \mathcal{N}\left(0, \|\boldsymbol{a}\|_2^2\right),$$

by the property of jointly normal random variables. Therefore, to characterize the expectation of its absolute value, we have:

$$\mathbb{E}\left[|\langle \boldsymbol{v}, \boldsymbol{a} \rangle|\right] = \frac{1}{\sqrt{2\pi}\|\boldsymbol{a}\|_2} \int_{-\infty}^{+\infty} |x| \exp\left(-\frac{x^2}{2\|\boldsymbol{a}\|_2}\right) dx = \frac{2}{\sqrt{2\pi}\|\boldsymbol{a}\|_2} \int_{0}^{+\infty} x \exp\left(-\frac{x^2}{2\|\boldsymbol{a}\|_2^2}\right) dx.$$

Let $u = x^2/(2\|\boldsymbol{a}\|_2^2)$, and substitute it into the integral, we have:

$$\mathbb{E}\left[|\langle \boldsymbol{v}, \boldsymbol{a} \rangle|\right] = \sqrt{\frac{2}{\pi}} \frac{1}{\|\boldsymbol{a}\|_2} \int_{0}^{+\infty} \|\boldsymbol{a}\|_2^2 \exp\left(-u\right) du = \sqrt{\frac{2}{\pi}} \|\boldsymbol{a}\|_2.$$

This concludes the proof of the lemma. ∎

We also assume that both the link function $\sigma(\cdot)$ and the value function $V(\pi_{\boldsymbol{\theta}})$ are $L$-smooth, and therefore the function value difference of two points can be approximated by the first-order Taylor's expansion with controllable error, which is stated in the following lemma.

**Lemma 3 (Lemma 7 in (Liu et al., 2018b))** *For any $L$-smooth function $f : \mathbb{R}^d \to \mathbb{R}$, and any pair of points $x \in \mathbb{R}^d$ and $y \in \mathbb{R}^d$, we have:*

$$|f(y) - f(x) - \langle \nabla f(x), y - x \rangle| \leq \frac{L}{2}\|y - x\|_2^2.$$

### F.2 PROOF OF CONVERGENCE RATE

Now, we follow the Lyapunov drift analysis framework to analyze ZSPO in algorithm 1. For simplicity, let the perturbation distance $\mu_t = \mu$ to be time-homogeneous. Since the value function $V(\pi_{\boldsymbol{\theta}})$ is $L$-smooth, by Lemma 3, we can approximate the value function increment at each gradient step as follows:

$$\begin{aligned}
V(\pi_{\boldsymbol{\theta}_t}) - V(\pi_{\boldsymbol{\theta}_{t+1}}) \leq & \langle -\nabla_{\boldsymbol{\theta}} V(\pi_{\boldsymbol{\theta}_t}), \boldsymbol{\theta}_{t+1} - \boldsymbol{\theta}_t \rangle + \frac{L}{2}\|\boldsymbol{\theta}_{t+1} - \boldsymbol{\theta}_t\|_2^2 \\
= & -\alpha_t \langle \nabla_{\boldsymbol{\theta}} V(\pi_{\boldsymbol{\theta}_t}), \hat{\boldsymbol{g}}_t \rangle + \frac{\alpha_t^2 L}{2}\|\hat{\boldsymbol{g}}_t\|_2^2 \\
= & -\alpha_t \operatorname{sign}\left[\sum_{n=1}^{N} o_{t,n} - \frac{1}{2}\right] \langle \nabla_{\boldsymbol{\theta}} V(\pi_{\boldsymbol{\theta}_t}), \boldsymbol{v}_t \rangle + \frac{\alpha_t^2 L}{2}\mathbb{E}\left[\|\boldsymbol{v}_t\|_2^2\right],
\end{aligned}$$

where the first equality is due to the ascent update in line 9 of algorithm 1 and the second equality is due to the expression of $\hat{\boldsymbol{g}}_t$ in line 8 of algorithm 1 and the fact that the sign is either 1 or $-1$, so it would not affect the norm. By Lemma 1, the expected squared norm of the perturbation vector $\boldsymbol{v}_t$ is exactly the dimension $d$, so we can proceed as:

$$\begin{aligned}
V(\pi_{\boldsymbol{\theta}_t}) - V(\pi_{\boldsymbol{\theta}_{t+1}}) \leq & -\alpha_t \operatorname{sign}\left[\sum_{n=1}^{N} o_{t,n} - \frac{1}{2}\right] \langle \nabla_{\boldsymbol{\theta}} V(\pi_{\boldsymbol{\theta}_t}), \boldsymbol{v}_t \rangle + \frac{\alpha_t^2 L d}{2} \\
= & -\alpha_t \underbrace{\operatorname{sign}\left[V(\pi_{\boldsymbol{\theta}_t'}) - V(\pi_{\boldsymbol{\theta}_t})\right] \langle \nabla_{\boldsymbol{\theta}} V(\pi_{\boldsymbol{\theta}_t}), \boldsymbol{v}_t \rangle}_{D_1} + \frac{\alpha_t^2 L d}{2} \\
& -\alpha_t \underbrace{\left(\operatorname{sign}\left[\sum_{n=1}^{N} o_{t,n} - \frac{1}{2}\right] - \operatorname{sign}\left[V(\pi_{\boldsymbol{\theta}_t'}) - V(\pi_{\boldsymbol{\theta}_t})\right]\right) \langle \nabla_{\boldsymbol{\theta}} V(\pi_{\boldsymbol{\theta}_t}), \boldsymbol{v}_t \rangle}_{D_2},
\end{aligned}$$

where in the last equality, we add and subtract the sign of the value function difference $\text{sign}[V(\pi_{\boldsymbol{\theta}_t'}) - V(\pi_{\boldsymbol{\theta}_t})]$. The first term $D_1$ does not involve any preference feedback. The second term $D_2$ characterizes the difference between the majority vote result and the sign of the value function, which we hope they coincide in most cases. In order to efficiently bound the gradient norm, we would need to extract a negative drift from either $D_1$ or $D_2$, which directly relates to the gradient of the value function. Indeed, it comes from $D_1$. From a high-level, if the perturbation vector $\boldsymbol{v}_t$ has a positive inner product with the gradient, the perturbed policy $\pi_{\boldsymbol{\theta}_t'}$ should have a larger value function, since the parameter shifts towards the gradient ascending direction, then we will have $\text{sign}[V(\pi_{\boldsymbol{\theta}_t'}) - V(\pi_{\boldsymbol{\theta}_t})] \geq 0$ and as a consequence $D_1 \geq 0$. The following lemma relates $D_1$ to the gradient norm:

**Lemma 4** *For* ZSPO, *conditioned on the information filtration $\mathcal{F}_t$ of any time $t$, the negative drift term $D_1$ can be upper bounded as follows:*

$$\mathbb{E}\left[D_1|\mathcal{F}_t\right] = \mathbb{E}_{\boldsymbol{v}_t}\left[\text{sign}\left[V(\pi_{\boldsymbol{\theta}_t + \mu \boldsymbol{v}_t}) - V(\pi_{\boldsymbol{\theta}_t})\right]\langle \nabla_{\boldsymbol{\theta}} V(\pi_{\boldsymbol{\theta}_t}), \boldsymbol{v}_t\rangle\right] \geq \sqrt{\frac{2}{\pi}}\|\nabla_{\boldsymbol{\theta}} V(\pi_{\boldsymbol{\theta}_t})\|_2 - \mu L d.$$

The proof of Lemma 4 is deferred. With its help, we can obtain the following drift upper bound:

$$\mathbb{E}\left[V(\pi_{\boldsymbol{\theta}_t}) - V(\pi_{\boldsymbol{\theta}_{t+1}})|\mathcal{F}_t\right] \leq -\alpha_t\sqrt{\frac{2}{\pi}}\|\nabla_{\boldsymbol{\theta}} V(\pi_{\boldsymbol{\theta}_t})\|_2 + \alpha_t \mu L d + \frac{\alpha_t^2 L d}{2} + \alpha_t\left|\mathbb{E}[D_2|\mathcal{F}_t]\right|.$$

Then, it suffices to bound the error $D_2$ that uses preference feedback to estimate the sign of the value function difference. The following lemma characterizes this error and implies that it will not be larger than the negative drift except for some additional positive terms that can be made small if the number of batches $N$ increases and the perturbation distance $\mu$ is chosen properly.

**Lemma 5** *For* ZSPO, *conditioned on the information filtration $\mathcal{F}_t$ of any time $t$, the preference error term $D_2$ can be upper bounded as follows:*

$$\left|\mathbb{E}\left[D_2|\mathcal{F}_t\right]\right| = \left|\mathbb{E}\left[\left(\text{sign}\left[\sum_{n=1}^{N} o_{t,n} - \frac{1}{2}\right] - \text{sign}\left[V(\pi_{\boldsymbol{\theta}_t'}) - V(\pi_{\boldsymbol{\theta}_t})\right]\right)\langle \nabla_{\boldsymbol{\theta}} V(\pi_{\boldsymbol{\theta}_t}), \boldsymbol{v}_t\rangle\right]\right|$$

$$\leq \frac{2}{e}\sqrt{\frac{2}{\pi}}\|\nabla_{\boldsymbol{\theta}} V(\pi_{\boldsymbol{\theta}_t})\|_2 + 4\left(\mu L d + \frac{\varepsilon_D^*}{\mu}\right) + \frac{8}{\mu}\varsigma^{-1}\left(\sqrt{\frac{2}{N}}\right).$$

With the help of both Lemma 4 and Lemma 5, we can derive the drift upper bound as follows:

$$\mathbb{E}\left[V(\pi_{\boldsymbol{\theta}_t}) - V(\pi_{\boldsymbol{\theta}_{t+1}})|\mathcal{F}_t\right]$$

$$\leq -\left(1 - \frac{1}{e}\right)\sqrt{\frac{2}{\pi}}\alpha_t\|\nabla_{\boldsymbol{\theta}} V(\pi_{\boldsymbol{\theta}_t})\|_2 + 5\alpha_t \mu L d + \frac{\alpha_t^2 L d}{2} + \frac{4\alpha_t \varepsilon_D^*}{\mu} + \frac{8\alpha_t}{\mu}\varsigma^{-1}\left(\sqrt{\frac{2}{N}}\right).$$

Let $c_0 = (1 - e^{-1})\sqrt{2/\pi} > 0$ be the coefficient of the negative drift, and we have:

$$\mathbb{E}\left[V(\pi_{\boldsymbol{\theta}_t}) - V(\pi_{\boldsymbol{\theta}_{t+1}})|\mathcal{F}_t\right] \leq -c_0\alpha_t\|\nabla_{\boldsymbol{\theta}} V(\pi_{\boldsymbol{\theta}_t})\|_2 + \alpha_t^2 L d$$

$$+ 8\alpha_t\left[\mu L d + \frac{\varepsilon_D^*}{\mu} + \frac{1}{\mu}\varsigma^{-1}\left(\sqrt{\frac{2}{N}}\right)\right].$$

Finally, we take the expectation over the filtration $\mathcal{F}_t$ and then take a telescoping sum over the time horizon $t$, and with some manipulation over the terms, we will obtain:

$$\frac{1}{\sum_{\tau=1}^{T} \alpha_\tau}\sum_{t=1}^{T} \alpha_t\mathbb{E}\left[\|\nabla_{\boldsymbol{\theta}} V(\pi_{\boldsymbol{\theta}_t})\|_2\right] \leq \frac{1}{c_0}\left(\frac{V(\boldsymbol{\theta}_1) - V(\boldsymbol{\theta}_{T+1})}{\sum_{\tau=1}^{T} \alpha_\tau} + \frac{\sum_{t=1}^{T} \alpha_t^2}{\sum_{\tau=1}^{T} \alpha_\tau}L d\right)$$

$$+ \frac{8}{c_0}\left[\mu L d + \frac{\varepsilon_D^*}{\mu} + \frac{1}{\mu}\varsigma^{-1}\left(\sqrt{\frac{2}{N}}\right)\right].$$

Since we choose the learning rate $\alpha_t = \Theta(\sqrt{H/dt})$, this implies the following relationship:

$$\sum_{t=1}^{T} \alpha_t = \Theta\left(\sqrt{\frac{HT}{d}}\right), \quad \sum_{t=1}^{T} \alpha_t^2 = \Theta\left(\frac{H \log T}{d}\right).$$

Recall the mechanism of choosing $\boldsymbol{\theta}_R$ for output, we have:

$$\mathbb{E}\left[\|\nabla_{\boldsymbol{\theta}} V(\pi_{\boldsymbol{\theta}_R})\|_2\right] = \mathcal{O}\left(\sqrt{\frac{Hd\log^2 T}{T}} + \frac{1}{\mu}\left(\varsigma^{-1}\left(\sqrt{\frac{2}{N}}\right) + \varepsilon_D^*\right) + \mu d\right).$$

### F.3 PROOF OF LEMMA 4

In this section, we prove the lower bound on the negative drift term $D_1$. Notice that $D_1$ does not involve any randomness from the preference feedback mechanism, and therefore the randomness only comes from both $\boldsymbol{\theta}_t$ and the choice of $\boldsymbol{v}_t$. We take a conditional expectation over the filtration $\mathcal{F}_t$, where $\boldsymbol{\theta}_t$ is viewed as a conditionally given, then the only randomness is the choice of $\boldsymbol{v}_t$:

$$\mathbb{E}[D_1|\mathcal{F}_t] = \mathbb{E}_{\boldsymbol{v}_t}\left[\text{sign}\left[V(\pi_{\boldsymbol{\theta}_t'}) - V(\pi_{\boldsymbol{\theta}_t})\right]\langle \nabla_{\boldsymbol{\theta}} V(\pi_{\boldsymbol{\theta}_t}), \boldsymbol{v}_t\rangle\right].$$

According to Lemma 3, the value function difference can be approximated by its linearization, and therefore neglecting the approximation error, the sign of the value function difference will be the same as the sign of $\langle \nabla_{\boldsymbol{\theta}} V(\pi_{\boldsymbol{\theta}_t}), \boldsymbol{v}_t\rangle$. This motivates us the separate $\boldsymbol{v}_t$ into the following three events depending on how close the sampled perturbation vector is to the gradient of the value function. Define:

$$\mathcal{E}_{\boldsymbol{v}_t,+} = \left\{\langle \nabla_{\boldsymbol{\theta}} V(\pi_{\boldsymbol{\theta}_t}), \boldsymbol{v}_t\rangle \geq \frac{\mu L}{2}\|\boldsymbol{v}_t\|_2^2\right\},$$

$$\mathcal{E}_{\boldsymbol{v}_t,-} = \left\{\langle \nabla_{\boldsymbol{\theta}} V(\pi_{\boldsymbol{\theta}_t}), \boldsymbol{v}_t\rangle \leq -\frac{\mu L}{2}\|\boldsymbol{v}_t\|_2^2\right\},$$

$$\mathcal{E}_{\boldsymbol{v}_t,0} = \left\{|\langle \nabla_{\boldsymbol{\theta}} V(\pi_{\boldsymbol{\theta}_t}), \boldsymbol{v}_t\rangle| \leq \frac{\mu L}{2}\|\boldsymbol{v}_t\|_2^2\right\}.$$

We call them the positive event, the negative event, and the margin event, respectively, since the inner product of the gradient and perturbation vector $\boldsymbol{v}_t$ would be positive, negative, and close to 0. On the positive event $\mathcal{E}_{\boldsymbol{v}_t,+}$, we use Lemma 3 to lower bound the value function difference as follows:

$$\begin{aligned}
V(\pi_{\boldsymbol{\theta}_t'}) - V(\pi_{\boldsymbol{\theta}_t}) \geq &\langle \nabla_{\boldsymbol{\theta}} V(\pi_{\boldsymbol{\theta}_t}), \boldsymbol{\theta}_t' - \boldsymbol{\theta}_t\rangle - \frac{L}{2}\|\boldsymbol{\theta}_t' - \boldsymbol{\theta}_t\|_2^2 \\
\geq &\mu\langle \nabla_{\boldsymbol{\theta}} V(\pi_{\boldsymbol{\theta}_t}), \boldsymbol{v}_t\rangle - \frac{\mu^2 L}{2}\|\boldsymbol{v}_t\|_2^2 \\
\geq &\frac{\mu^2 L}{2}\|\boldsymbol{v}_t\|_2^2 - \frac{\mu^2 L}{2}\|\boldsymbol{v}_t\|_2^2 \\
= &0,
\end{aligned}$$

where the last inequality is due to the definition of the positive event $\mathcal{E}_{\boldsymbol{v}_t,+}$. Since the sign of the inner product is also positive on this event, we can conclude that:

$$\text{sign}\left[V(\pi_{\boldsymbol{\theta}_t'}) - V(\pi_{\boldsymbol{\theta}_t})\right]\langle \nabla_{\boldsymbol{\theta}} V(\pi_{\boldsymbol{\theta}_t}), \boldsymbol{v}_t\rangle = \langle \nabla_{\boldsymbol{\theta}} V(\pi_{\boldsymbol{\theta}_t}), \boldsymbol{v}_t\rangle = |\langle \nabla_{\boldsymbol{\theta}} V(\pi_{\boldsymbol{\theta}_t}), \boldsymbol{v}_t\rangle|.$$

Similarly, on the negative event $\mathcal{E}_{\boldsymbol{v}_t,-}$, we also use Lemma 3 to upper bound the value function difference as:

$$\begin{aligned}
V(\pi_{\boldsymbol{\theta}_t'}) - V(\pi_{\boldsymbol{\theta}_t}) \leq &\langle \nabla_{\boldsymbol{\theta}} V(\pi_{\boldsymbol{\theta}_t}), \boldsymbol{\theta}_t' - \boldsymbol{\theta}_t\rangle + \frac{L}{2}\|\boldsymbol{\theta}_t' - \boldsymbol{\theta}_t\|_2^2 \\
\leq &\mu\langle \nabla_{\boldsymbol{\theta}} V(\pi_{\boldsymbol{\theta}_t}), \boldsymbol{v}_t\rangle + \frac{\mu^2 L}{2}\|\boldsymbol{v}_t\|_2^2 \\
\leq &-\frac{\mu^2 L}{2}\|\boldsymbol{v}_t\|_2^2 + \frac{\mu^2 L}{2}\|\boldsymbol{v}_t\|_2^2 \\
= &0,
\end{aligned}$$

where the last inequality is due to the definition of the negative event $\mathcal{E}_{\boldsymbol{v}_t,-}$. So the sign of the value function is negative, and we also have the inner product being negative, so overall, the product of the value function sign and the inner product is still positive, i.e.,

$$\text{sign}\left[V(\pi_{\boldsymbol{\theta}_t'}) - V(\pi_{\boldsymbol{\theta}_t})\right]\langle \nabla_{\boldsymbol{\theta}} V(\pi_{\boldsymbol{\theta}_t}), \boldsymbol{v}_t\rangle = -\langle \nabla_{\boldsymbol{\theta}} V(\pi_{\boldsymbol{\theta}_t}), \boldsymbol{v}_t\rangle = |\langle \nabla_{\boldsymbol{\theta}} V(\pi_{\boldsymbol{\theta}_t}), \boldsymbol{v}_t\rangle|.$$

Now, we analyze the negative drift term $D_1$ under separated events as follows. Since on the filtration $\mathcal{F}_t$, the variable $\boldsymbol{\theta}_t$ is given and the randomness only comes from sampling the vector $\boldsymbol{v}_t$, so we omit the filtration and denote the expectation over $\boldsymbol{v}_t$.

$$
\begin{aligned}
\mathbb{E}[D_1|\mathcal{F}_t] =& \mathbb{E}_{\boldsymbol{v}_t}\left[D_1 \mathbb{1}_{\mathcal{E}_{\boldsymbol{v}_t,+}}\right] + \mathbb{E}_{\boldsymbol{v}_t}\left[D_1 \mathbb{1}_{\mathcal{E}_{\boldsymbol{v}_t,-}}\right] + \mathbb{E}_{\boldsymbol{v}_t}\left[D_1 \mathbb{1}_{\mathcal{E}_{\boldsymbol{v}_t,0}}\right] \\
=& \mathbb{E}_{\boldsymbol{v}_t}\left[|\langle \nabla_{\boldsymbol{\theta}} V(\pi_{\boldsymbol{\theta}_t}), \boldsymbol{v}_t\rangle| \mathbb{1}_{\mathcal{E}^{\complement}_{\boldsymbol{v}_t,0}}\right] + \mathbb{E}_{\boldsymbol{v}_t}\left[D_1 \mathbb{1}_{\mathcal{E}_{\boldsymbol{v}_t,0}}\right] \\
=& \mathbb{E}_{\boldsymbol{v}_t}\left[|\langle \nabla_{\boldsymbol{\theta}} V(\pi_{\boldsymbol{\theta}_t}), \boldsymbol{v}_t\rangle|\right] - \mathbb{E}_{\boldsymbol{v}_t}\left[|\langle \nabla_{\boldsymbol{\theta}} V(\pi_{\boldsymbol{\theta}_t}), \boldsymbol{v}_t\rangle| \mathbb{1}_{\mathcal{E}_{\boldsymbol{v}_t,0}}\right] + \mathbb{E}_{\boldsymbol{v}_t}\left[D_1 \mathbb{1}_{\mathcal{E}_{\boldsymbol{v}_t,0}}\right] \\
=& \sqrt{\frac{2}{\pi}} \|\nabla_{\boldsymbol{\theta}} V(\pi_{\boldsymbol{\theta}_t})\|_2 - \mathbb{E}_{\boldsymbol{v}_t}\left[|\langle \nabla_{\boldsymbol{\theta}} V(\pi_{\boldsymbol{\theta}_t}), \boldsymbol{v}_t\rangle| \mathbb{1}_{\mathcal{E}_{\boldsymbol{v}_t,0}}\right] + \mathbb{E}_{\boldsymbol{v}_t}\left[D_1 \mathbb{1}_{\mathcal{E}_{\boldsymbol{v}_t,0}}\right] \\
\geq& \sqrt{\frac{2}{\pi}} \|\nabla_{\boldsymbol{\theta}} V(\pi_{\boldsymbol{\theta}_t})\|_2 - \mathbb{E}_{\boldsymbol{v}_t}\left[|\langle \nabla_{\boldsymbol{\theta}} V(\pi_{\boldsymbol{\theta}_t}), \boldsymbol{v}_t\rangle| \mathbb{1}_{\mathcal{E}_{\boldsymbol{v}_t,0}}\right] - \mathbb{E}_{\boldsymbol{v}_t}\left[|D_1| \mathbb{1}_{\mathcal{E}_{\boldsymbol{v}_t,0}}\right],
\end{aligned}
$$

where in the second last step, we use Lemma 2 to relate the inner product to the gradient norm of the value function, and in the last inequality, we used the fact that the absolute value is no less than the original value. Then, it remains to analyze the additional terms. Notice that the sign of the value function difference in $D_1$ is either $-1$ or $1$, so we have:

$$
\begin{aligned}
\mathbb{E}_{\boldsymbol{v}_t}\left[|D_1| \mathbb{1}_{\mathcal{E}_{\boldsymbol{v}_t,0}}\right] =& \mathbb{E}_{\boldsymbol{v}_t}\left[\left|\text{sign}\left[V(\pi_{\boldsymbol{\theta}_t'}) - V(\pi_{\boldsymbol{\theta}_t})\right] \langle \nabla_{\boldsymbol{\theta}} V(\pi_{\boldsymbol{\theta}_t}), \boldsymbol{v}_t\rangle\right| \mathbb{1}_{\mathcal{E}_{\boldsymbol{v}_t,0}}\right] \\
=& \mathbb{E}_{\boldsymbol{v}_t}\left[|\langle \nabla_{\boldsymbol{\theta}} V(\pi_{\boldsymbol{\theta}_t}), \boldsymbol{v}_t\rangle| \mathbb{1}_{\mathcal{E}_{\boldsymbol{v}_t,0}}\right],
\end{aligned}
$$

which is the same as the other additional term. By the definition of the margin event $\mathcal{E}_{\boldsymbol{v}_t,0}$, we have:

$$
\mathbb{E}_{\boldsymbol{v}_t}\left[|\langle \nabla_{\boldsymbol{\theta}} V(\pi_{\boldsymbol{\theta}_t}), \boldsymbol{v}_t\rangle| \mathbb{1}_{\mathcal{E}_{\boldsymbol{v}_t,0}}\right] \leq \frac{\mu L}{2} \mathbb{E}\left[\|\boldsymbol{v}_t\|_2^2\right] = \frac{\mu L d}{2},
$$

where the last equality uses Lemma 1. Substituting it back to the lower bound of $D_1$, we conclude:

$$
\mathbb{E}[D_1|\mathcal{F}_t] \geq \sqrt{\frac{2}{\pi}} \|\nabla_{\boldsymbol{\theta}} V(\pi_{\boldsymbol{\theta}_t})\|_2 - \mu L d.
$$

### F.4 PROOF OF LEMMA 5

In this section, we prove Lemma 5 to bound $D_2$, which characterizes the approximation error of preference feedback with majority vote to estimate the value function sign. Notice that the randomness of $D_2$ comes from four sources: the current policy parameter $\boldsymbol{\theta}_t$, the perturbation distance $\boldsymbol{v}_t$, the trajectory batches $\mathcal{D}_{n,1}$ and $\mathcal{D}_{n,0}$, and the preference feedback $o_{t,n}$. Therefore, we first take a conditional expectation over the first two sources to fix the policies $\pi_{\boldsymbol{\theta}_t}$ and $\pi_{\boldsymbol{\theta}_t'}$ and analyze $D_2$ as:

$$
\begin{aligned}
|\mathbb{E}[D_2|\boldsymbol{\theta}_t', \boldsymbol{\theta}_t]| \leq& \left|\mathbb{E}\left[\text{sign}\left[\sum_{n=1}^{N} o_{t,n} - \frac{1}{2}\right] - \text{sign}\left[V(\pi_{\boldsymbol{\theta}_t'}) - V(\pi_{\boldsymbol{\theta}_t})\right]\middle| \boldsymbol{\theta}_t', \boldsymbol{\theta}_t\right]\right| |\langle \nabla_{\boldsymbol{\theta}} V(\pi_{\boldsymbol{\theta}_t}), \boldsymbol{v}_t\rangle| \\
=& 2\mathbb{P}\left(\text{sign}\left[\sum_{n=1}^{N} o_{t,n} - \frac{1}{2}\right] \neq \text{sign}\left[V(\pi_{\boldsymbol{\theta}_t'}) - V(\pi_{\boldsymbol{\theta}_t})\right]\right) |\langle \nabla_{\boldsymbol{\theta}} V(\pi_{\boldsymbol{\theta}_t}), \boldsymbol{v}_t\rangle|,
\end{aligned}
$$

where in the last step, we notice that the term will only be non-zero if the two signs inside the expectation disagree with each other. Then, it is sufficient to study the event where the signs of the majority vote and the value function difference disagree. Again, this probability would depend on how large the gap in the value function difference is. If the value function difference is bounded away from 0, then the oracles would find it easier to distinguish the two policies and thus it is more likely that the sign of the majority vote will be the same as the sign of the value function. So we consider the following three events:

$$
\mathcal{E}_{\boldsymbol{v}_t,+,\varepsilon_D^*} = \left\{\langle \nabla_{\boldsymbol{\theta}} V(\pi_{\boldsymbol{\theta}_t}), \boldsymbol{v}_t\rangle \geq \mu L \|\boldsymbol{v}_t\|_2^2 + \frac{\varepsilon_D^*}{\mu}\right\},
$$

$$
\mathcal{E}_{\boldsymbol{v}_t,-,\varepsilon_D^*} = \left\{\langle \nabla_{\boldsymbol{\theta}} V(\pi_{\boldsymbol{\theta}_t}), \boldsymbol{v}_t\rangle \leq -\mu L \|\boldsymbol{v}_t\|_2^2 - \frac{\varepsilon_D^*}{\mu}\right\},
$$

$$
\mathcal{E}_{\boldsymbol{v}_t,0,\varepsilon_D^*} = \left\{|\langle \nabla_{\boldsymbol{\theta}} V(\pi_{\boldsymbol{\theta}_t}), \boldsymbol{v}_t\rangle| \leq \mu L \|\boldsymbol{v}_t\|_2^2 + \frac{\varepsilon_D^*}{\mu}\right\}.
$$

We abuse the name and call the three events the positive event, the negative event, and the margin event, respectively. On the positive event $\mathcal{E}_{\boldsymbol{v}_t, +, \varepsilon_D^*}$, we can lower bound the value function difference according to Lemma 3 as follows:

$$V(\pi_{\boldsymbol{\theta}_t'}) - V(\pi_{\boldsymbol{\theta}_t}) \geq \langle \nabla_{\boldsymbol{\theta}} V(\pi_{\boldsymbol{\theta}_t}), \boldsymbol{\theta}_t' - \boldsymbol{\theta}_t \rangle - \frac{L}{2} \|\boldsymbol{\theta}_t' - \boldsymbol{\theta}_t\|_2^2$$

$$= \mu \langle \nabla_{\boldsymbol{\theta}} V(\pi_{\boldsymbol{\theta}_t}), \boldsymbol{v}_t \rangle - \frac{\mu^2 L}{2} \|\boldsymbol{v}_t\|_2^2$$

$$\geq \frac{\mu^2 L}{2} \|\boldsymbol{v}_t\|_2^2 + \varepsilon_D^*.$$

Notice that this implies the value function difference is positive and thus the sign is positive, so $D_2$ will be non-zero over this event if the following event happens:

$$\left\{ \text{sign} \left[ \sum_{n=1}^N o_{t,n} - \frac{1}{2} \right] = -1 \right\} \subset \left\{ \sum_{n=1}^N o_{t,n} \leq \frac{N}{2} \right\}.$$

Notice that $o_{t,n}$ is the preference feedback for the $n$-th pair of trajectory batches. Therefore, it is a Bernoulli random variable with expectation $p_t$ as follows:

$$p_t = \mathbb{E}_{\mathcal{D}_1 \sim \pi_{\boldsymbol{\theta}_t'}, \mathcal{D}_2 \sim \pi_{\boldsymbol{\theta}_t}} \left[ \sigma \left( \bar{r}(\mathcal{D}_1) - \bar{r}(\mathcal{D}_0) \right) \right] = \frac{1}{2} + \mathbb{E}_{\mathcal{D}_1 \sim \pi_{\boldsymbol{\theta}_t'}, \mathcal{D}_2 \sim \pi_{\boldsymbol{\theta}_t}} \left[ \varsigma \left( \bar{r}(\mathcal{D}_1) - \bar{r}(\mathcal{D}_0) \right) \right],$$

where the last equality uses the definition of the deviation function. Recall that we not only obtained the value function difference is larger than 0, we also obtained $V(\pi_{\boldsymbol{\theta}_t'}) - V(\pi_{\boldsymbol{\theta}_t}) \geq \varepsilon_D^*$, and by the definition of the distinguishability $\varepsilon_D^*$, the expected deviation function over batches should be lower bounded by the deviation function of the value function difference. This is because the value function difference is large enough so that oracles can distinguish the better one from the two in expectation when looking at trajectory batches. Therefore, the deviation from half can be bounded as:

$$p_t - \frac{1}{2} = \mathbb{E}_{\mathcal{D}_1 \sim \pi_{\boldsymbol{\theta}_t'}, \mathcal{D}_2 \sim \pi_{\boldsymbol{\theta}_t}} \left[ \varsigma \left( \bar{r}(\mathcal{D}_1) - \bar{r}(\mathcal{D}_0) \right) \right] \geq \frac{1}{2} \varsigma \left( \frac{V(\pi_{\boldsymbol{\theta}_t'}) - V(\pi_{\boldsymbol{\theta}_t})}{2} \right).$$

We can also revisit the lower bound of the value function difference to obtain a finer one using Lemma 3 as follows:

$$V(\pi_{\boldsymbol{\theta}_t'}) - V(\pi_{\boldsymbol{\theta}_t}) \geq \langle \nabla_{\boldsymbol{\theta}} V(\pi_{\boldsymbol{\theta}_t}), \boldsymbol{\theta}_t' - \boldsymbol{\theta}_t \rangle - \frac{L}{2} \|\boldsymbol{\theta}_t' - \boldsymbol{\theta}_t\|_2^2$$

$$= \frac{\mu \langle \nabla_{\boldsymbol{\theta}} V(\pi_{\boldsymbol{\theta}_t}), \boldsymbol{v}_t \rangle}{2} + \frac{\mu \langle \nabla_{\boldsymbol{\theta}} V(\pi_{\boldsymbol{\theta}_t}), \boldsymbol{v}_t \rangle}{2} - \frac{\mu^2 L}{2} \|\boldsymbol{v}_t\|_2^2$$

$$\geq \frac{\mu \langle \nabla_{\boldsymbol{\theta}} V(\pi_{\boldsymbol{\theta}_t}), \boldsymbol{v}_t \rangle}{2} + \frac{\varepsilon_D^*}{2},$$

where we used the definition of the positive event $\mathcal{E}_{\boldsymbol{v}_t, +, \varepsilon_D^*}$ in the last inequality. So substitute it into the deviation of $p_t$, we can have:

$$p_t - \frac{1}{2} \geq \frac{1}{2} \varsigma \left( \frac{\mu \langle \nabla_{\boldsymbol{\theta}} V(\pi_{\boldsymbol{\theta}_t}), \boldsymbol{v}_t \rangle}{4} + \frac{\varepsilon_D^*}{4} \right).$$

Therefore, by Hoeffding's concentration inequality, we can bound the probability that $D_2$ is non-zero on the positive event as follows:

$$\mathbb{P} \left( \text{sign} \left[ \sum_{n=1}^N o_{t,n} - \frac{1}{2} \right] = -1 \right) = \mathbb{P} \left( \sum_{n=1}^N o_{t,n} \leq \frac{N}{2} \right)$$

$$\leq \exp \left( -\frac{N}{2} \varsigma^2 \left( \frac{\mu \langle \nabla_{\boldsymbol{\theta}} V(\pi_{\boldsymbol{\theta}_t}), \boldsymbol{v}_t \rangle}{4} + \frac{\varepsilon_D^*}{4} \right) \right).$$

So, on the positive event, we can upper bound $D_2$ as follows:

$$|\mathbb{E}[D_2 | \boldsymbol{\theta}_t', \boldsymbol{\theta}_t]| \leq 2 \mathbb{P} \left( \text{sign} \left[ \sum_{n=1}^N o_{t,n} - \frac{1}{2} \right] = -1 \right) |\langle \nabla_{\boldsymbol{\theta}} V(\pi_{\boldsymbol{\theta}_t}), \boldsymbol{v}_t \rangle|$$

$$\leq 2 \exp \left( -\frac{N}{2} \varsigma^2 \left( \frac{\mu \langle \nabla_{\boldsymbol{\theta}} V(\pi_{\boldsymbol{\theta}_t}), \boldsymbol{v}_t \rangle}{4} + \frac{\varepsilon_D^*}{4} \right) \right) |\langle \nabla_{\boldsymbol{\theta}} V(\pi_{\boldsymbol{\theta}_t}), \boldsymbol{v}_t \rangle|$$

On the negative event $\mathcal{E}_{\boldsymbol{v}_t,-,\varepsilon_D^*}$, we can also perform the same analysis. Using the anti-symmetric nature of the preference deviation function, we can obtain the same upper bound as follows:

$$|\mathbb{E}\left[D_2|\boldsymbol{\theta}_t',\boldsymbol{\theta}_t\right]| \leq 2\exp\left(-\frac{N}{2}\varsigma^2\left(\frac{\mu\left|\langle\nabla_{\boldsymbol{\theta}}V(\pi_{\boldsymbol{\theta}_t}),\boldsymbol{v}_t\rangle\right|}{4}+\frac{\varepsilon_D^*}{4}\right)\right)\left|\langle\nabla_{\boldsymbol{\theta}}V(\pi_{\boldsymbol{\theta}_t}),\boldsymbol{v}_t\rangle\right|.$$

On the margin event $\mathcal{E}_{\boldsymbol{v}_t,0,\varepsilon_D^*}$, the absolute value of the inner product between the gradient and the perturbation vector is upper bounded even though the sign of the value function difference may disagree, so we will have:

$$|\mathbb{E}\left[D_2|\boldsymbol{\theta}_t',\boldsymbol{\theta}_t\right]| \leq 2\mathbb{P}\left(\mathrm{sign}\left[\sum_{n=1}^N o_{t,n}-\frac{1}{2}\right]\neq\mathrm{sign}\left[V(\pi_{\boldsymbol{\theta}_t'})-V(\pi_{\boldsymbol{\theta}_t})\right]\right)\left|\langle\nabla_{\boldsymbol{\theta}}V(\pi_{\boldsymbol{\theta}_t}),\boldsymbol{v}_t\rangle\right|$$

$$\leq 2\left(\mu L\|\boldsymbol{v}_t\|_2^2+\frac{\varepsilon_D^*}{\mu}\right),$$

where in the last inequality, we use the definition of the margin event. Finally, we can combine the three events and take an expectation over the perturbation vector $\boldsymbol{v}_t$ as follows:

$$|\mathbb{E}\left[D_2|\mathcal{F}_t\right]| \leq \mathbb{E}_{\boldsymbol{v}_t}\left[|\mathbb{E}\left[D_2|\boldsymbol{\theta}_t',\boldsymbol{\theta}_t\right]|\right]$$

$$= \mathbb{E}_{\boldsymbol{v}_t}\left[|\mathbb{E}\left[D_2|\boldsymbol{\theta}_t',\boldsymbol{\theta}_t\right]|\,\mathbb{1}_{\mathcal{E}_{\boldsymbol{v}_t,0,\varepsilon_D^*}^{\complement}}\right]+\mathbb{E}_{\boldsymbol{v}_t}\left[|\mathbb{E}\left[D_2|\boldsymbol{\theta}_t',\boldsymbol{\theta}_t\right]|\,\mathbb{1}_{\mathcal{E}_{\boldsymbol{v}_t,0,\varepsilon_D^*}}\right].$$

On the margin event $\mathcal{E}_{\boldsymbol{v}_t,0,\varepsilon_D^*}$, we have the expected absolute value of $D_2$ bounded as follows:

$$\mathbb{E}_{\boldsymbol{v}_t}\left[|\mathbb{E}\left[D_2|\boldsymbol{\theta}_t',\boldsymbol{\theta}_t\right]|\,\mathbb{1}_{\mathcal{E}_{\boldsymbol{v}_t,0,\varepsilon_D^*}}\right] \leq \mathbb{E}_{\boldsymbol{v}_t}\left[2\left(\mu L\|\boldsymbol{v}_t\|_2^2+\frac{\varepsilon_D^*}{\mu}\right)\mathbb{1}_{\mathcal{E}_{\boldsymbol{v}_t,0,\varepsilon_D^*}}\right]$$

$$\leq 2\left(\mu L\mathbb{E}\left[\|\boldsymbol{v}_t\|_2^2\right]+\frac{\varepsilon_D^*}{\mu}\right)$$

$$= 2\left(\mu Ld+\frac{\varepsilon_D^*}{\mu}\right),$$

where the last equality uses Lemma 1. On the event $\mathcal{E}_{\boldsymbol{v}_t,0,\varepsilon_D^*}^{\complement}=\mathcal{E}_{\boldsymbol{v}_t,+,\varepsilon_D^*}\cup\mathcal{E}_{\boldsymbol{v}_t,-,\varepsilon_D^*}$, we use the following notation to denote the exponential term in the upper bound of $D_2$ for simplicity:

$$f_t\left(\boldsymbol{v}_t\right) = \exp\left(-\frac{N}{2}\varsigma^2\left(\frac{\mu\left|\langle\nabla_{\boldsymbol{\theta}}V(\pi_{\boldsymbol{\theta}_t}),\boldsymbol{v}_t\rangle\right|}{4}+\frac{\varepsilon_D^*}{4}\right)\right)\leq 1.$$

Then, the expectation over the positive and negative events $\mathcal{E}_{\boldsymbol{v}_t,0,\varepsilon_D^*}^{\complement}$ can be bounded as follows:

$$\mathbb{E}_{\boldsymbol{v}_t}\left[|\mathbb{E}\left[D_2|\boldsymbol{\theta}_t',\boldsymbol{\theta}_t\right]|\,\mathbb{1}_{\mathcal{E}_{\boldsymbol{v}_t,0,\varepsilon_D^*}^{\complement}}\right]$$

$$\leq 2\mathbb{E}_{\boldsymbol{v}_t}\left[f_t(\boldsymbol{v}_t)\left|\langle\nabla_{\boldsymbol{\theta}}V(\pi_{\boldsymbol{\theta}_t}),\boldsymbol{v}_t\rangle\right|\mathbb{1}_{\mathcal{E}_{\boldsymbol{v}_t,0,\varepsilon_D^*}^{\complement}}\right]$$

$$= 2\mathbb{E}_{\boldsymbol{v}_t}\left[f_t(\boldsymbol{v}_t)\left|\langle\nabla_{\boldsymbol{\theta}}V(\pi_{\boldsymbol{\theta}_t}),\boldsymbol{v}_t\rangle\right|\right]-2\mathbb{E}_{\boldsymbol{v}_t}\left[f_t(\boldsymbol{v}_t)\left|\langle\nabla_{\boldsymbol{\theta}}V(\pi_{\boldsymbol{\theta}_t}),\boldsymbol{v}_t\rangle\right|\mathbb{1}_{\mathcal{E}_{\boldsymbol{v}_t,0,\varepsilon_D^*}}\right]$$

$$\leq 2\mathbb{E}_{\boldsymbol{v}_t}\left[f_t(\boldsymbol{v}_t)\left|\langle\nabla_{\boldsymbol{\theta}}V(\pi_{\boldsymbol{\theta}_t}),\boldsymbol{v}_t\rangle\right|\right]+2\mathbb{E}_{\boldsymbol{v}_t}\left[\left|\langle\nabla_{\boldsymbol{\theta}}V(\pi_{\boldsymbol{\theta}_t}),\boldsymbol{v}_t\rangle\right|\mathbb{1}_{\mathcal{E}_{\boldsymbol{v}_t,0,\varepsilon_D^*}}\right]$$

$$\leq 2\mathbb{E}_{\boldsymbol{v}_t}\left[f_t(\boldsymbol{v}_t)\left|\langle\nabla_{\boldsymbol{\theta}}V(\pi_{\boldsymbol{\theta}_t}),\boldsymbol{v}_t\rangle\right|\right]+2\left(\mu L\mathbb{E}\left[\|\boldsymbol{v}_t\|_2^2\right]+\frac{\varepsilon_D^*}{\mu}\right)$$

$$= 2\mathbb{E}_{\boldsymbol{v}_t}\left[f_t(\boldsymbol{v}_t)\left|\langle\nabla_{\boldsymbol{\theta}}V(\pi_{\boldsymbol{\theta}_t}),\boldsymbol{v}_t\rangle\right|\right]+2\left(\mu Ld+\frac{\varepsilon_D^*}{\mu}\right),$$

where in the second last inequality, we used $f_t(\boldsymbol{v}_t)\leq 1$ and in the last inequality, we used the definition of the margin event $\mathcal{E}_{\boldsymbol{v}_t,0,\varepsilon_D^*}$. The last equality is due to Lemma 1. Then, collecting the two terms, we conclude:

$$|\mathbb{E}\left[D_2|\mathcal{F}_t\right]| \leq 2\mathbb{E}_{\boldsymbol{v}_t}\left[f_t(\boldsymbol{v}_t)\left|\langle\nabla_{\boldsymbol{\theta}}V(\pi_{\boldsymbol{\theta}_t}),\boldsymbol{v}_t\rangle\right|\right]+4\left(\mu Ld+\frac{\varepsilon_D^*}{\mu}\right).$$

Then, it remains to construct an upper bound for the first term. Let $w_t = \langle \nabla_{\boldsymbol{\theta}} V(\pi_{\boldsymbol{\theta}_t}), \boldsymbol{v}_t \rangle$, and we use $D_3$ to denote the first term as follows:

$$D_3 = \mathbb{E}_{\boldsymbol{v}_t}\left[ f_t(\boldsymbol{v}_t) \left| \langle \nabla_{\boldsymbol{\theta}} V(\pi_{\boldsymbol{\theta}_t}), \boldsymbol{v}_t \rangle \right| \right] = \mathbb{E}_{\boldsymbol{v}_t}\left[ \exp\left( -\frac{N}{2}\varsigma^2 \left( \frac{\mu|w_t|}{4} + \frac{\varepsilon_D^*}{4} \right) \right) |w_t| \right]$$

$$\leq \mathbb{E}_{\boldsymbol{v}_t}\left[ \exp\left( -\frac{N}{2}\varsigma^2 \left( \frac{\mu|w_t|}{4} \right) \right) |w_t| \right],$$

where we used the fact that $\varepsilon_D^* \geq 0$ and $\varsigma(\cdot)$ is monotonically increasing. We also separate the event into two cases depending on $|w_t|$ as follows:

$$D_3 \leq \mathbb{E}_{\boldsymbol{v}_t}\left[ \exp\left( -\frac{N}{2}\varsigma^2 \left( \frac{\mu|w_t|}{4} \right) \right) |w_t| \mathbb{1}_{|w_t| \geq \frac{4}{\mu}\varsigma^{-1}\left( \sqrt{\frac{2}{N}} \right)} \right]$$

$$+ \mathbb{E}_{\boldsymbol{v}_t}\left[ \exp\left( -\frac{N}{2}\varsigma^2 \left( \frac{\mu|w_t|}{4} \right) \right) |w_t| \mathbb{1}_{|w_t| \leq \frac{4}{\mu}\varsigma^{-1}\left( \sqrt{\frac{2}{N}} \right)} \right].$$

In the first event where $|w_t|$ is large, we can obtain that the exponential is upper bounded as:

$$\exp\left( -\frac{N}{2}\varsigma^2 \left( \frac{\mu|w_t|}{4} \right) \right) \leq \frac{1}{e}.$$

So we can upper bound the first term as follows:

$$\mathbb{E}_{\boldsymbol{v}_t}\left[ \exp\left( -\frac{N}{2}\varsigma^2 \left( \frac{\mu|w_t|}{4} \right) \right) |w_t| \mathbb{1}_{|w_t| \geq \frac{4}{\mu}\varsigma^{-1}\left( \sqrt{\frac{2}{N}} \right)} \right] \leq \frac{1}{e}\mathbb{E}_{\boldsymbol{v}_t}\left[ \left| \langle \nabla_{\boldsymbol{\theta}} V(\pi_{\boldsymbol{\theta}_t}), \boldsymbol{v}_t \rangle \right| \right]$$

$$= \frac{1}{e}\sqrt{\frac{2}{\pi}}\|\nabla_{\boldsymbol{\theta}} V(\pi_{\boldsymbol{\theta}_t})\|_2,$$

where the last equality uses Lemma 2. In the second event, we know that $|w_t|$ is upper bounded and the exponential is smaller than 1, so we obtain:

$$\mathbb{E}_{\boldsymbol{v}_t}\left[ \exp\left( -\frac{N}{2}\varsigma^2 \left( \frac{\mu|w_t|}{4} \right) \right) |w_t| \mathbb{1}_{|w_t| \leq \frac{4}{\mu}\varsigma^{-1}\left( \sqrt{\frac{2}{N}} \right)} \right] \leq \frac{4}{\mu}\varsigma^{-1}\left( \sqrt{\frac{2}{N}} \right).$$

Finally, summarizing the two cases immediately gives us an upper bound on $D_3$ as follows:

$$D_3 \leq \frac{4}{\mu}\varsigma^{-1}\left( \sqrt{\frac{2}{N}} \right) + \frac{1}{e}\sqrt{\frac{2}{\pi}}\|\nabla_{\boldsymbol{\theta}} V(\pi_{\boldsymbol{\theta}_t})\|_2.$$

And it naturally leads to the bound on the approximation error term $D_2$ as follows:

$$|\mathbb{E}[D_2|\mathcal{F}_t]| \leq 2D_3 + 4\left( \mu L d + \frac{\varepsilon_D^*}{\mu} \right)$$

$$\leq \frac{2}{e}\sqrt{\frac{2}{\pi}}\|\nabla_{\boldsymbol{\theta}} V(\pi_{\boldsymbol{\theta}_t})\|_2 + 4\left( \mu L d + \frac{\varepsilon_D^*}{\mu} + \frac{2}{\mu}\varsigma^{-1}\left( \sqrt{\frac{2}{N}} \right) \right).$$

## G  PROOF OF COROLLARY 2

In this section, we show that with the smoothness assumption on the link function, we can use a perturbation distance $\mu$ that doesn't require the knowledge of the link function itself or the distinguishability $\varepsilon_D^*$ to achieve a good convergence guarantee. First, combining Theorem 1 and Proposition 1, we have:

$$\mathbb{E}\left[\|\nabla_{\boldsymbol{\theta}} V(\pi_{\boldsymbol{\theta}_R})\|_2\right] = \widetilde{\mathcal{O}}\left( \sqrt{\frac{Hd}{T}} + \frac{1}{\mu}\varsigma^{-1}\left( \sqrt{\frac{2}{N}} \right) + \frac{H}{\mu\sqrt{D}} + \mu d \right).$$

Since the link function is smooth, the deviation function $\varsigma(\cdot)$ is also smooth with Lipchitz constant $L$, then we have for any points $x \in \mathbb{R}$,

$$\left| \left( \varsigma^{-1} \right)' (x) - \left( \varsigma^{-1} \right)' (0) \right| = \left| \frac{1}{\varsigma'(x)} - \frac{1}{\varsigma'(0)} \right| = \frac{|\varsigma'(x) - \varsigma'(0)|}{|\varsigma'(x)|\,|\varsigma'(0)|} \leq \frac{L|x|}{|\varsigma'(x)|\,\sigma'(0)}.$$

In the last step, we used the assumption 2 that the derivative of the link function at 0 is positive. And on the other hand, we have by the smoothness of $\varsigma(\cdot)$, we have $|\varsigma'(x) - \varsigma'(0)| = |\varsigma'(x) - \sigma'(0)| \leq L|x|$, and this implies $|\varsigma'(x)| \geq \sigma'(0) - L|x| \geq \sigma'(0)/2$ when $|x| \leq \varsigma'(0)/(2L)$. Therefore, we can conclude that in the $\sigma'(0)/(2L)$ neighborhood of the origin, we have:

$$\left| \left( \varsigma^{-1} \right)' (x) - \left( \sigma^{-1} \right)' (0) \right| \leq \frac{2L}{(\varsigma'(0))^2} |x|,$$

so the inverse of deviation function is smooth with parameter $2L/(\sigma'(0))^2$. Then, suppose the number of batches $N$ is large enough such that the argument is inside the neighborhood of the origin, i.e., $N \geq 8L^2/(\sigma'(0))^2$, then by Lemma 3, we have:

$$\varsigma^{-1} \left( \sqrt{\frac{2}{N}} \right) = \varsigma^{-1} \left( \sqrt{\frac{2}{N}} \right) - \varsigma^{-1}(0) \leq \left( \varsigma^{-1} \right)'(0) \sqrt{\frac{2}{N}} + \frac{L}{(\sigma'(0))^2} \frac{2}{N} = \mathcal{O} \left( \frac{1}{\sigma'(0)\sqrt{N}} \right).$$

Therefore, if we select the perturbation distance $\mu$ as in Corollary 2 independent of the link function and the distinguishability, i.e., $\mu^2 = \Theta(d^{-1} \max\{1/\sqrt{N}, H/\sqrt{D}\})$, we have:

$$\mathbb{E}\left[ \|\nabla_{\boldsymbol{\theta}} V(\pi_{\boldsymbol{\theta}_R})\|_2 \right] = \widetilde{\mathcal{O}} \left( \sqrt{\frac{Hd}{T}} + \max\left\{ \frac{1}{\sigma'(0)}, 1 \right\} \frac{\sqrt{d}}{N^{\frac{1}{4}}} + \frac{\sqrt{Hd}}{D^{\frac{1}{4}}} \right).$$

