# OpenReview forum: "Provable Policy Optimization for Reinforcement Learning from Trajectory Preferences with an Unknown Link Function"
_ICLR.cc/2026/Conference — Submitted to ICLR 2026_

### Official Review · Reviewer_HMb6 · 2025-10-25

**Soundness:** 3
**Presentation:** 4
**Contribution:** 3
**Rating:** 6
**Confidence:** 3

**Summary:**

This paper proposes a zeroth-order optimization algorithm for preference-based RL with an unknown link function. The algorithm updates the policy parameters toward a direction that increases the value function by using random perturbations and majority votes. The authors provide convergence guarantees to a stationary point and experiments demonstrating that the algorithm outperforms baselines when the link function is misspecified.

**Strengths:**

- The proposed algorithm is simple to implement and directly optimizes the policy without relying on a reward model, similar to DPO and NLHF.

- The theoretical analysis is thorough and mathematically sound.

- The empirical results are also promising: the algorithm performs better than baselines that assume a BT model under misspecified settings, over all three environments.

**Weaknesses:**

- The motivation could be stronger. The authors could further discuss why it is important to consider broader classes of link functions beyond the BT model, given that the BT model performs well empirically and has theoretical justification through the Borda rule in social choice theory.
- Moreover, while the linear link function is used to demonstrate misspecification in experiments, it is unclear how practical this model is. It is also unclear how much performance degradation occurs in the proposed algorithm (compared to others) if the true link function were sigmoid.
- Scalability to high-dimensional settings such as LLMs is unclear. It seems unlikely to me that the proposed approach would scale efficiently.
- The bound converges to 0 as $D \rightarrow \infty$, but practically it is difficult to use large $D$ in online settings.

**Questions:**

- In line 161, the global optimizer $\theta^*$ is defined but never used. It seems that no theoretical results establish convergence to a global optimizer.
- In experiments, what value of $D$ was used? I only found $D=1$ in the stochastic gridworld example.

---

> ### Author Response · Authors · 2025-11-21
> **Response to Reviewer HMb6 (1/2)**
>
> We thank the reviewer for their time reviewing our paper and for the valuable comments. We are excited that the reviewer considers our theoretical analysis solid and acknowledges the novelty of the unknown link function setting. We comment on the concerns of the reviewer as follows:
>
> # Motivation of the Unknown Link Function beyond BT
> We agree with the reviewer that the Bradley-Terry model has a strong root in random utility and social choice theory, including the connections to Borda-type scoring rules under Gumbel noise. Even though the Bradley-Terry model has been useful in practice, there are still calibration and misspecification issues reported. Our point is not to dismiss BT, but to **handle settings where the effective link induced by heterogeneous annotators, LLM judges, or mixed protocols is unknown and may deviate from any fixed parametric family**.
>
> In practice, human annotators from crowd-sourcing platforms are typically uncalibrated and have heterogeneous preferences, as they come from diverse backgrounds and hold varying opinions, resulting in different shapes and varying “temperature”. If we use LLMs as judges, their internal scoring function may also depend on the context, prompts, and system settings, which creates heterogeneity as well. In most applications, the preference oracles are **mixtures of different types of annotators**, and this usually makes **the effective link function of preference oracles not Bradley-Terry and unknown**.
>
> At the same time, **other link functions also connect to random-utility models**, e.g., if the noise for random utility is normally distributed, it gives the *probit* model, where the link function is a normal CDF. The discussion on different link functions studied in the literature and their connection to social choice theory is in Appendix B. The purpose of our paper is to develop general methods that remain valid under any admissible link function and are robust to mis-specifications of any particular parametric form, i.e., BT, probit, etc.
>
> # Linear Link Function in Experiments
>
> We chose the linear link function in the experiment for two reasons.
> 1. The linear link function is also connected to the random-utility model. If the noise difference follows a uniform distribution, the pairwise choice probability is the CDF of a uniform distribution, which is the linear link function. This provides a clean and interpretable non-logistic example within the random-utility framework.
> 2. Our aim to use the linear link function in the experiment is to **stress-test BT-based methods under a qualitatively different link function**. The linear link exposes precisely the kind of misspecification where inverting an incorrect BT $\sigma$ can produce large magnitude bias, while a sign-based method is invariant. Our experiment results demonstrate that indeed BT-based methods suffer from link function mismatch while ZSPO remains robust.
>
> We also conducted an ablation experiment by changing the underlying link function back to the Bradley-Terry model, as shown in Fig.2(a). The performance of BT-based baselines such as RM+PPO, Online DPO, and ZPG significantly improves in terms of final return because there is no mis-specification. As expected, our ZSPO does not outperform baselines when BT is correctly specified, but remains competitive, demonstrating its robustness and invariance. This also answers the reviewer’s question of degradation: **ZSPO does not degrade catastrophically when the true link is sigmoid**.
>
> # Global Optimizer and Global Convergence
> You are correct that the global optimizer $\theta*$ is not used since our theory only establishes convergence to a stationary point, not the global optimizer $\theta*$. It is only used in our proof to analyze the Lyapunov function, and we removed it from the main text.
>
> For most policy-gradient-style methods with general policy parameterization, showing the convergence to the stationary point is standard due to the **non-convex nature** of this class of RL problems. To establish convergence to the global optimizer, one would require some **additional assumptions, such as convexity or the PL condition** over the function parameterization. Considering the difficulty and generality of the setting, we focus on the convergence to a stationary point and a meaningful result showing the effectiveness of the proposed algorithm to a wider range of RL problems.

---

> > ### Author Response · Authors · 2025-11-21
> > **Response to Reviewer HMb6 (2/2)**
> >
> > # Scalability to high-dimensional settings
> >
> > We agree that naively applying ZSPO to full LLM parameter spaces is not immediately practical, and we do not intend to overclaim otherwise. Our paper’s theory and experiment focus on a **generic recipe and theoretical guidance** to solve preference-based RL with an unknown link function. For high-dimensional models in practice, ZSPO should be viewed as a **conceptual building block** instead of a plug-and-play large-scale system in its current form.
> >
> > In our conclusion and Appendix D.3, we discussed **combining ZSPO with efficient zeroth-order techniques** such as MeZO with PEFT for LLMs, which has been quite successful and delivers competitive performance with memory and computation efficiency. In practice, we believe that achieving the scalability of ZSPO would take a few more steps. First, we need to confine the perturbation subspaces to be low-dimensional using LoRA or PEFT-type optimization framework to reduce the effective dimension of the problem; second, we may use block-wise update and parallel implementation to improve the policy iteration efficiency; and finally, we could combine ZSPO with variance-reduction methods and modern optimization techniques tailored to large models. Inspired by the success of similar zeroth-order methods like MeZo in LLMs, we see a clear potential to extend our theoretical ZSPO framework to large-scale real-world applications, such as robotics and LLMs, which we explicitly identify as future work.
> >
> > # The Role of batch size D and Practical Choices
> >
> > In theory, the dependence of $D$ appears through the distinguishability term $ \varepsilon_{D}* $, i.e., a larger $D$ reduces the distinguishability $\varepsilon_D*$ and thus tightens the convergence rate. However, this is a **sufficient condition** theoretically, and not a practical prescription. Depending on the RL problem instance, the distinguishability term $\varepsilon_D^*$ could itself be very small even if we only use a small $D$, or in more extreme cases, when we use $D=1$. In our revised experiments, **we deliberately use $D=1$**, i.e., each iteration uses a single trajectory per policy to query preference. **ZSPO still converges and exhibits robustness** against a mis-specified link function. This shows that for many practical problems, especially experiments with deterministic transitions, *the problem instances are much nicer than the theoretical worst-case, which does not require using a large $D$ for convergence*.
> >
> > The large $D$ discussion is mainly about understanding how distinguishability may affect the consistency between trajectory preferences and policy preferences in general, which affects all algorithms in the unknown link function setting. It may not be a requirement for practical uses. The same principle also applies to the choice of $N$ in practice, where a large $N$ helps to reduce variance in ascent direction estimation. In practice, like our experiments, we chose $N=1$ and ZSPO still converges, which means large $N$ may not be a must in many applications. Also, we conducted an ablation over $N$ in Fig.2(b) showing that increasing $N$ would help reduce variance and improve the stability and convergence speed of ZSPO.
> >
> > ---
> > We hope these clarifications and additional experiments address the reviewer’s concerns about motivation beyond BT, the role of the linear link, scalability, and the nature of convergence guarantees.

---

> > > ### Author Response · Authors · 2025-11-25
> > >
> > > Dear Reviewer HMb6,
> > >
> > > Thank you for your time reviewing our paper and the positive feedback with thoughtful comments. We want to follow up to see whether our rebuttal has addressed your concerns. Please don't hesitate to let us know if you have any other questions/comments. Thanks!
> > >
> > > best,
> > >
> > > Authors

---

### Official Review · Reviewer_2LSp · 2025-11-01

**Soundness:** 3
**Presentation:** 2
**Contribution:** 2
**Rating:** 4
**Confidence:** 3

**Summary:**

This work extends the standard Bradley–Terry model for preference-based RL by proposing a method that does not assume knowledge of the link function.
The core algorithm, Zeroth-Order Sign Policy Optimization (ZSPO), achieves independence from the explicit link function by only estimating the sign of the value difference. The authors theoretically establish ZSPO's convergence to a stationary policy under smoothness and distinguishability assumptions. Empirical evaluation on classic control benchmarks shows the method's strong robustness when the link function is misspecified.

**Strengths:**

•  Addresses a critical, well-motivated problem by successfully generalizing preference-based Reinforcement Learning to remove the dependence on a known link function.

•  The paper features a clear and insightful articulation of the deep connection between dueling bandit theory and preference-based RL, which helps situate the proposed method within the broader theoretical landscape.

**Weaknesses:**

• The core theoretical framework and results appear to be a minor adaptation of the Zhang & Ying (2024)'s ZPG framework, rather than offering significant conceptual originality.
-	Zhang, Qining, and Lei Ying. "Zeroth-order policy gradient for reinforcement learning from human feedback without reward inference." arXiv preprint arXiv:2409.17401 (2024).
• The reliance on a majority-vote mechanism requiring an extremely large number of rollouts (e.g., $N=1000$) raises serious concerns about practical inefficiency and high computational cost, making it potentially infeasible for realistic RLHF applications.
• The comparison across algorithms is questionable by the use of different preference oracles (e.g., BT versus linear) for each method. This weakens the direct comparability of the empirical results.
• Although the experiments demonstrate empirical performance improvement and strong robustness to link-function misspecification, the paper provides no corresponding theoretical analysis to explain the underlying mechanism. Specifically, it is unclear how the proposed method mitigates optimization bias or guarantees convergence stability under such misspecification.

**Questions:**

•  Can the authors clarify what theoretical elements are genuinely novel beyond those inherited from ZPG?
•  How does the majority-vote approximation error behave in smaller N regimes (e.g., N=32,64) relevant to real RLHF scenarios?
•  Why are different preference oracles used for different algorithms? Wouldn’t this lead to unfair comparisons?

---

> ### Author Response · Authors · 2025-11-21
> **Response to Reviewer 2LSp (1/2)**
>
> We thank the reviewer for their time reviewing our paper and for the valuable comments. We are excited that the reviewer recognized that our work addresses a well-motivated problem. We comment on the concerns of the reviewer as follows:
>
> # Novelty Compared to ZPG
>
> We want to emphasize that the main purpose of this paper is to **answer a fundamental question** in RL from preference feedback that Zhang & Ying (2024)'s ZPG framework **fails to address**: *can RL agents learn from preferences without knowing the link function?* ZSPO brings an affirmative answer to this important question, which itself has significant conceptual originality. While we indeed build on the zeroth-order perspective of that work, our problem formulation, estimator, and analysis differ in several fundamental ways:
>
> 1. **Different problem setting** (unknown $\sigma(\cdot)$ vs known $\sigma(\cdot)$): ZPG assumes the link function $\sigma(\cdot)$ is **known** and crucially **relies on its inverse to recover the numerical value differences** from preference probabilities. In our paper, we explicitly consider the unknown link function setting and propose to estimate the sign signal, and we show convergence without access to the link function.
>
> 2. **Different estimator** (sign vs magnitude): ZPG estimates the magnitude of value differences by inverting the link function, which is impossible in the unknown link function setting studied in this paper. Our ZSPO only estimates the sign of the value difference to tackle the unknown link function setting, which contains weaker information. But we can still show convergence similar to ZPG.
>
> 3. **New challenges** (distinguishability): Our convergence analysis in the unknown link function setting introduces the preference distinguishability constant $\varepsilon_D^*$ that captures the bottleneck of optimizing the expected value function from preferences. This concept captures how well the unknown link function $\sigma(\cdot)$ and trajectory batch size $D$ allow preferences to distinguish policies, and we characterize how it affects the convergence rate. This quantity and *the study of its trade-off in this paper are absent in ZPG*, which assumes a fixed and known link function.
>
> 4. **Different proof frameworks:** Due to the sign function in ZSPO, the proof framework developed in ZPG cannot be used for the analysis of ZSPO, since we are not trying to estimate the gradient from zeroth-order methods; we are only estimating a direction that is positively correlated with the gradient. This requires a more delicate analysis of the negative drift for the Lyapunov function, and technical decomposition and control of error terms are not the same as we discussed in Sec. 4.3.
>
> Given the major differences in the setting, challenge, and analysis framework stated above, we believe it is **NOT accurate** to regard ZSPO as “a minor adaptation of Zhang & Ying (2024)'s ZPG framework”.
>
> # Practicality of Batched Rollout and Comparison
>
> We want to point out that in practice, we may **NOT require a large $D$ or a large $N$** for the number of rollouts for ZSPO to deliver a good performance. Typically, we never used $N=1000$ for our Gymnasium experiments. In our revised experiments for the Gymnasium environments, we only use $N=D=1$, i.e., one rollout per policy iteration, and the preference oracle compares one pair of trajectories every time. Yet, ZSPO still demonstrates **the best performance compared to baselines** when there exists a link function mis-specification. In the ablation study where the underlying link function is logistic, which matches the BT-based baselines, ZSPO still has a similar and competitive performance.
>
> We also conducted an ablation study on $N$ and discovered that even though $N=1$ is enough for ZSPO to converge and deliver a good performance, using a larger $N$ would help it to **converge much faster with a more stable training dynamic**, which is consistent with the theoretical term involving $N$ in Corollary 2. This confirms that the large N in Theorem 1 is a sufficient condition for all instances. We can use smaller values in practice.
>
> In theory, for stochastic MDPs, a large $D$ for the batched preference is required to ensure the distribution of return concentrates around the expectation to exhibit some “regularity”, so that trajectory preference will be consistent with the policy preference with a small distinguishability $\varepsilon_D^*$. This is **necessary for any algorithm to learn the optimal policy** in the unknown link function setting. We discussed this unique challenge of the unknown link function setting in detail in Appendix D.2. But in practice,  the required $D$ and $N$ for a specific RL problem depend on the specific problem instance. Much **smaller $N$ and $D$ are possible** if the distribution of the return is “regular” (e.g., single-modal) with a small variance. We discussed these aspects in Appendix D.2.

---

> > ### Author Response · Authors · 2025-11-21
> > **Response to Reviewer 2LSp (2/2)**
> >
> > # Experimental Link Function Clarification
> > We want to clarify a key point regarding the experiment setup in the review: “The comparison across algorithms is questionable by the use of different preference oracles (e.g., BT versus linear) for each method”.
> >
> > In fact, all baselines in our experiment use **the same preference oracle with the linear function** in Fig.1, where we ran our experiments with 10 randomly chosen seeds and report the mean plus standard error over seeds. In our ablation study shown in Fig.2(a), we changed the underlying link function to logistic, and every algorithm obtained the preference feedback from a Bradley-Terry model, matching the baseline algorithms as well. In this case, ZSPO still has a competitive performance. *The difference between ZSPO and baselines is only in what link function each algorithm assumes, not what the real link function generating the preference is*. For all baselines, they assume a logistic link function, and for ZSPO, it does not need to assume any $\sigma$ since it only uses the sign from the majority vote.
> >
> > # Theoretical Explanation of Robustness to Link Function
> >
> > Our theoretical results have already captured the underlying mechanism of why ZSPO exhibits robustness to link function mis-specification, and how it mitigates bias to guarantee stability. The reason is that we only use **the sign of the function difference instead of its magnitude**, which is **a robust estimator invariant of the link function shape**. First, as long as the link function is monotone with $\sigma’(0) > 0$ as in our Assumption 2, the sign of the reward difference between two trajectories will coincide with the sign of the expected preference from oracles. Subject to the distinguishability analysis resulting in an error $\varepsilon_D^*$, the majority vote mechanism in ZSPO makes sure the sign estimator is correct with high probability, and the convergence rate is $\mu^{-1}\sqrt{2/N}$ as shown in Corollary 2. This characterizes how accurately the majority vote approximates the population-level sign. Both terms apply under any true link function satisfying assumption 2, including the non-BT link functions. Methods like ZPG and DPO rely on recovering or using numerical reward differences via the known link function. Their gradient estimators are systematically biased in magnitude and even direction. ZSPO, by contrast, discards magnitude and uses only the sign, which is invariant under monotone reparameterizations of the true link function.
> >
> > # Behavior of Majority-vote error in small N
> >
> > Theoretical behavior of the majority vote error is captured in the third term of Theorem 1 via $\varsigma^{-1}(\sqrt{2/N})$, which decays as N increases. We complement this with the new N-ablation study. As described before, our ablation with $N=1,4,16$ shows that ZSPO already converges for $N=1$ and large $N$ improves speed and reduces variance.
> >
> > ---
> > We hope these clarifications and new experiments address the reviewer’s concerns regarding novelty, practicality, fairness of comparisons, and the theoretical basis for robustness under link-function misspecification. We would appreciate it if the reviewer could reevaluate our paper and the rating based on our rebuttal and the new empirical results.

---

> > > ### Author Response · Authors · 2025-11-25
> > > **Reminder**
> > >
> > > Dear Reviewer 2LSp,
> > >
> > > Thank you for your time reviewing our paper. We want to follow up to see whether our rebuttal has addressed your concerns. We would very much appreciate it if the reviewer would consider re-evaluating their score based on our response. Please don't hesitate to let us know if you have any other questions/comments. Thanks!
> > >
> > > best,
> > >
> > > Authors

---

### Official Review · Reviewer_C5xV · 2025-11-01

**Soundness:** 3
**Presentation:** 3
**Contribution:** 2
**Rating:** 4
**Confidence:** 5

**Summary:**

This paper addresses the problem of **preference-based reinforcement learning (PbRL)** under the assumption that the **link function (σ)** — which defines the relationship between trajectory reward differences and preference probabilities — is **unknown or potentially misspecified**.

To tackle this, the authors propose **ZSPO (Zeroth-Order Sign Policy Optimization)**, an algorithm that enables provable policy improvement without explicit knowledge of σ.

ZSPO estimates only the **sign of the value difference** between two policies and determines an ascent direction based on this binary (±1) signal.

The method thus relies solely on **directional information** instead of full reward magnitude or an assumed link model, effectively leveraging **1-bit preference feedback** for policy updates.

Notably, while ZSPO is described as a **zeroth-order, gradient-free** algorithm, it is not entirely gradient-free in implementation.

The update rule still follows a **gradient-ascent-like structure**, where the estimated sign signal acts as a coarse directional proxy for ∇θV(πθ).

In this sense, ZSPO avoids explicit gradient computation or backpropagation but retains the overall framework of gradient-based policy optimization.

Theoretically, the paper provides convergence guarantees for ZSPO under general MDP settings, showing that the expected policy gradient norm decreases polynomially with respect to the number of iterations and trajectory comparisons.

Empirically, across CartPole-v1, HalfCheetah-v5, and Hopper-v5, the method demonstrates strong robustness under **link function misspecification** (e.g., true link linear vs. assumed logistic), outperforming baselines including RM+PPO, Online DPO, ZPG, and Evolution Strategies in both stability and final performance.

**Strengths:**

1. **Novel Theoretical Formulation** — Unlike most prior works in preference-based RL that assume a known link function (typically the logistic Bradley–Terry model), this paper explicitly formulates the problem under *unknown link functions* and provides the first formal convergence guarantee in this setting.
2. **Simple but Powerful Algorithmic Insight** — The paper shows that effective policy improvement is achievable using only the **sign** of value differences, discarding magnitude information. This leads to a minimal yet expressive framework for **1-bit feedback–based policy optimization**.
3. **Mathematical Rigor** — The analysis is well-structured, offering explicit convergence rates (Theorem 1 and Corollary 1), a clear definition of the distinguishability constant ε\*_D, and an interpretable sample-complexity bound. The proof framework extends Lyapunov-drift analysis into a zeroth-order optimization context, which is an elegant theoretical contribution.
4. **Empirical Persuasiveness** — The experiments systematically simulate link-function misspecification across three Gymnasium control environments, demonstrating that ZSPO remains consistently robust compared to all baselines.
5. **Clear Relation to Prior Work** — The paper effectively situates ZSPO within the landscape of existing preference-based RL methods such as DPO, ZPG, ES, and RM+PPO, providing a fair and transparent comparative evaluation.

**Weaknesses:**

1. **Complexity of Online Preference Collection** — The algorithm requires multiple batch-level trajectory comparisons and majority-vote queries per iteration, which may be impractical or expensive in real human-in-the-loop systems.
2. **Discontinuity in Policy Updates** — Since the update direction is derived from 1-bit sign feedback, it can be highly sensitive to noise. Although convergence is theoretically ensured, the variance of updates may be substantial in practice.
3. **Limited Experimental Scope** — All experiments rely on synthetic preference oracles; no evaluation with real human or LLM feedback is included, which limits claims of real-world applicability.
4. **Baseline Comparability Issues** — Some baselines differ in setup: RM+PPO uses semi-offline reward modeling, while DPO includes KL regularization. Hence, the comparisons are not entirely controlled.
5. **Lack of Qualitative Analysis** — The paper could benefit from a visualization or sensitivity study showing how different link-function shapes (e.g., steep, flat, linear) influence learning dynamics.

**Questions:**

1. **Practical Scope of Link-Function Uncertainty** — In what real scenarios (e.g., human crowd evaluations, noisy LLM feedback) does the “unknown link function” assumption most accurately apply?
2. **Stability of Policy Updates** — Since 1-bit sign gradients might cause oscillation or overshooting in policy space, have you explored adaptive step-size or momentum strategies to mitigate instability?
3. **Human-in-the-Loop Applicability** — When extending to actual human feedback, how could the number of comparisons per iteration (N, D) be reduced? Would active preference sampling or uncertainty-driven querying help?
4. **Partial-Information Extensions** — If partial knowledge of the link function is available (e.g., monotonicity or σ′(0)), can the method be modified to accelerate convergence beyond the purely sign-based approach?
5. **Policy-Label Feedback (Relation to PPL Work)** — How might ZSPO adapt if behavior-policy labels are provided, as in online PPL settings? Could incorporating such distributional information improve gradient-direction accuracy?
6. **Comparison with Pebble’s Online PPO** — The Pebble framework (Wirth et al., 2021; Wang et al., 2023) also conducts online preference-based learning using PPO within a fully interactive loop. Given that RM+PPO in this paper follows a semi-offline setup, a comparison with Pebble’s *online* PPO (which shares the same data collection and update loop structure) could provide a more direct empirical baseline.

    Could the authors clarify why this comparison was not included, and whether differences in data collection protocol or link-function assumptions made it infeasible?

---

> ### Author Response · Authors · 2025-11-21
> **Response to Reviewer C5xV (1/2)**
>
> We thank the reviewer for their time reviewing our paper and for the valuable comments. We are excited that the reviewer appreciates the novelty of our theoretical formulation with an unknown link function, the simplicity, and the power of our algorithm. We next address the concerns of the reviewer:
>
> # Practicality of Batched Online Preference Collection
> We agree that our theoretical results require large batch sizes $D$ and many majority votes $N$ per update, which is because the theoretical bounds have to hold for the worst-case instances. However, **our revised experiments show that ZSPO works in the tested environments with $N=D=1$**, i.e., every policy iteration only generates a single trajectory pair for comparison and uses it for policy improvement. We observe that ZSPO remains stable, matches BT-based baselines when the true link is logistic, and continues to **outperform them when the link is misspecified**. We also included a new ablation over $N$, showing that **with more majority votes, ZSPO converges faster** and has a better performance. This demonstrates that large N and D in Theorem 1 are sufficient conditions.
>
> Moreover, in many applications, the preference oracle need not be a real human annotator. Our paper mainly focuses on designing algorithms that use any preference oracle efficiently to learn a good policy, and how to reduce the cost of collecting preferences is a different important aspect that is complementary to our work.
>
> # Improving the Stability of ZSPO in Practice
>
> Our theory shows that, on average, the sign direction remains positively correlated with the true policy gradient, which is sufficient for convergence guarantees with 1-bit feedback. For theoretical analysis and the simplicity to demonstrate how we use the sign gradient to tackle the unknown link function setting, we chose to present the algorithm in a simple SGD framework. We agree that this variant of sign-based updates can, in practice, be noisy. However, ZSPO’s update rule is entirely compatible with modern optimization methods, such as adaptive optimizers, simply by viewing the zeroth-order policy ascent direction $\hat{g}_t$ in Algorithm. 1 as the gradient of the optimizer. In our experiments, we **indeed used the AdamW optimizer, which produces a training curve as stable as or more stable than baselines**. We stated this implementation in Appendix C. Conceptually, if we reuse the comparisons from previous policy iterations, ZSPO can also be combined with uncertainty-based strategic querying to focus comparisons where the sign is most uncertain, thereby reducing the number of required comparisons.
>
> # Unknown Link Function
>
> In practice, human annotators from crowd-sourcing platforms are typically uncalibrated and have heterogeneous preferences, as they come from diverse backgrounds and hold varying opinions, resulting in different shapes and varying “temperature”. If we use LLMs as judges, their internal scoring function may also depend on the context, prompts, and system settings, which creates heterogeneity as well. In most applications, the preference oracles are **mixtures of different types of annotators, and this usually makes the effective link function of preference oracles not Bradley-Terry and unknown**. Usually, in real-world human-involvement experiments in social science, the human annotators must be calibrated before giving preferences, but this is typically more costly. Even though the Bradley-Terry model has been useful in practice, there are still calibration and misspecification issues reported. We also conducted an ablation experiment by changing the underlying link function from Bradley-Terry to linear, and the performance of **most algorithms in practice suffers a performance loss**. This justifies why we should study the unknown link function setting.
>
> # Preference Oracles in Experiments
>
> We agree that real human or LLM feedback is an important direction, and we explicitly list this as future work.. Our goal in this paper is to first provide a **theoretical and algorithmic foundation** for policy optimization under unknown link functions, and to evaluate the approach in a controlled setting where the true link is given so that **mis-specification can be precisely captured and studied**. Our algorithm framework is fully compatible with existing RLHF pipelines with human raters of LLMs-as-judge, which we view as orthogonal engineering work rather than the core theoretical contribution of this paper. This is similar to how many RLHF methods are initially studied with synthetic “human” oracles in the dueling bandit and preference-based RL literature before full-scale deployment in robotics and LLM tasks.

---

> ### Author Response · Authors · 2025-11-21
> **Response to Reviewer C5xV (2/2)**
>
> # Baselines and Comparability
>
> Our goal in experiments is to compare the performance of ZSPO to algorithms that are commonly used in the preference-based RL literature, and that, like ZSPO, **do not require reward inference**, under a shared trajectory and preference query budget. This allows us to demonstrate the impact of link-function mis-specification and the robustness of ZSPO.  We agree that the baselines may differ in some setups, but we view the differences as intrinsic to these methods rather than an artifact of our experiment setup. All methods are given the same preference and sample budget, and the difference is how they use these resources.
>
> We also appreciate the reviewer’s suggestion on comparing with Pebble; however, it is still an RL algorithm that **builds around online reward modeling, conceptually similar to RM+PPO**. The motivation to remove the reward model has been explained in the literature, such as [1]. Since the main focus of our paper is on algorithms without reward inference, we chose to concentrate on comparisons of algorithms, such as ZPG and DPO, that share this design philosophy. Therefore, given the limited space and time, we feel comparing with Pebble is not necessary in this paper.
>
> [1] An, et al. 2023 “Direct Preference-based Policy Optimization without Reward Modeling”
>
> # Partial Information of Link Function
>
> If the partial information, such as a lower bound on $\sigma’(0)$, is known, one could use this information to improve the learning rate schedule via Corollary 1,  or use the local linear approximation of the link function to approximate the reward difference when the trajectories being compared have similar rewards. This information may also be helpful in the gradient estimation, especially when the sign has large uncertainty and additional structure is needed to stabilize updates.
>
> # Other Comments
>
> We also want to point out that ZSPO is a zeroth-order (gradient-free) method, because it does not obtain and use the first-order gradient information to optimize the value function. The information ZSPO gathers is only binary feedback from preference oracles, and they are not directly related to the gradient. Whether the algorithm follows a gradient-ascent-like structure is not what distinguishes a zeroth-order method from a first-order method.
>
> ---
> We sincerely hope our rebuttal addresses the reviewer’s concerns. We would appreciate it if the reviewer could re-evaluate our paper and the ratings based on our rebuttal and the newly added experimental results.

---

> > ### Author Response · Authors · 2025-11-25
> > **Reminder**
> >
> > Dear Reviewer C5xV,
> >
> > Thank you for your time reviewing our paper. We want to follow up to see whether our rebuttal has addressed your concerns. We would very much appreciate it if the reviewer would consider re-evaluating their score based on our response. Please don't hesitate to let us know if you have any other questions/comments. Thanks!
> >
> > best,
> >
> > Authors

---

### Official Review · Reviewer_evWw · 2025-11-04

**Soundness:** 3
**Presentation:** 3
**Contribution:** 2
**Rating:** 4
**Confidence:** 4

**Summary:**

The paper presents a zero order preference based algorithm that does not require the specification of the link function, i.e., the function that connects the preference with the underlying rewards. Common algorithms use the Terry-Bradly model (a sigmoid) which can limit the performance of these algorithms in the case of preference misspecification as argued by the authors. The algorithm is a zero-order algorithm that directly searches in the parameter space of the policy. It chooses a random update direction and then computes via the preference feedback whether this direction improved performance or not. The main contribution of the paper is the theoertical analyis showing convergence bounds in terms of the gradient norm of the algorithm. Moreover, the algorithm is compared against baselines on 3 continuous control tasks.

**Strengths:**

- the theoretical analysis seems sound (but I could not check every detail)
- The new algorithm has proofable convergence
- Using preference-based RL without specification of the link-function has not been studied so far in the literature

**Weaknesses:**

- The assumptions made by the algorithm are very far from beeing practical. The algorithm requires that we generate D trajectories for both policies N times in order to compute a single gradient update. In practice, it will be very hard for humans to compare D trajectories. Most practical RLHF algorithms compare trajectories instead of policies, i.e., they can learn from the comparison of single trajectories instead of a batch of trajectories.
- The black-box manner of the algorithm also brings severe limitations (which is however also comparable to the recent ZPG algorithm). As the preference compares policy parameters and not trajectories, it is for example very hard for the algorithm to take random initial states into account. This will only work if we massively increase the number of trajectories D in the comparison.
- The experimental evaluation is not convincing. It looks very noisy. Its unclear how many seeds have been used, but from the plot it seems way too small to make a proper statistics. Authors should use at least 10 (better 20) seeds to get better statistics
- The experimental setting is also not fully clear to me. Are the baselines evaluated in a similar manner then the algorithm (use a batch of trajectories for the preference comparison) or are they applied to single trajectory pairs? Algorithms such as DPO are not black box, so they can leverage single trajectory comparisons in a much more straightforward way. It is an unfair comparison if these algorithms are evaluated only on preferences on trajectory batches as they do not have the same limitations as the presented algorithm.
- More ablations should be performed. For example, it would be insightful if the derived bounds hold at least approximately (i.e. by showing how the performance changes with number of comparisons N and batch size D). It would also be good to show experiments with different ground truth link functions, in particular, if the ground truth link function is indeed the terry-bradley model, how would the algorithm perform against the baselines that use the terry-bradley assumptions.

**Questions:**

- Please specifiy what exactly a "linear link function" is (used for the experiments)
- The link function is not formally defined in the beginning. That would help the understanding
- It would be helpful to see the number of trajectories or number of samples on the x axis in the results instead of the number of iterations
-

---

> ### Author Response · Authors · 2025-11-21
> **Response to Reviewer evWw (1/2)**
>
> We thank the reviewer for their time reviewing our paper and for the valuable comments. We are glad that the reviewer finds our theoretical analysis solid and acknowledges the novelty of the unknown link-function setting. We comment on the concerns of the reviewer as follows:
>
> # Practicality of Batched Rollout and Comparison
>
> The theoretical guarantees of ZSPO are for worst-case instances with concentration analysis, so they must hold uniformly for all problems. In such analysis, large $D$ and $N$ appear as sufficient conditions. In practice, most RL problems are not worst-case problems, and the required $D$ and $N$ for a specific RL problem depend on the instance, i.e., regularity of reward and noise level in the environment, which could be much smaller. The practicality of ZSPO should be evaluated by its experimental performance. We ran our Gymnasium experiments with **$N=D=1$ under the single-trajectory comparison setting, and ZSPO is still the best** among baselines. This shows that much smaller $N$ and $D$ are possible in practice if the distribution of the return is “regular” (e.g., single-modal) with a small variance. We discussed these aspects in Appendix D.2.
>
> # Black-Box Manner of ZSPO and Single Trajectory Comparison
>
> **In practice, ZSPO could still work with a single trajectory comparison** and still has a competitive performance, as shown in our updated experiments with $N=D=1$. Moreover, we allow random initial states in all experiments, showing that we do **NOT** need to “massively increase the number of trajectories $D$ in the comparison” to achieve a good performance. The reason is that the gymnasium environments are mostly deterministic, and the return usually has a regular distribution with low noise. So the policy value difference could be reliably inferred from the single trajectory difference with a small error when the perturbation distance is chosen properly.
>
> Moreover, we want to point out an overly simplified viewpoint in the review that DPO (as not a black box method) could learn a good policy from single-trajectory comparisons. The reason is that **DPO only assumes a deterministic MDP**. **The argument is NOT true in stochastic MDPs, which is the main focus of this paper**. The optimal policy for a deterministic MDP is the policy that generates the highest-reward trajectory. Therefore, if we learn the preference over trajectories from single-trajectory comparisons, we can optimize the policy by maximizing its occurrence. This relationship directly builds a path from trajectory comparison to policy comparison. However, in more general and challenging stochastic MDPs, the optimal policy is defined as the policy that, on average, generates the highest return, not the policy that has the largest chance of generating the highest-reward trajectory. The distinguishability issue studied in this paper naturally occurs. Therefore, even if we learn the ranking over all trajectories from single-trajectory comparisons, we still need a method to average the trajectories out to evaluate the value of the policies and identify which one has a higher return on average. In this case, **learning from single-trajectory comparisons is not enough to learn the best policy**, and we ultimately need policy preference oracles and have to resort to black-box methods due to the stochastic transition. In general, the notion of batched comparison and the batch size $D$ arises from this **fundamental bottleneck in preference-based RL, which affects all algorithms**. It is **NOT a weakness of ZSPO**. Using a large batch $D$ and $N$ for obtaining preference could be required for all preference-based link-function-agnostic RL algorithms in such worst-case settings.
>
> # Experimental Evaluation and Clarification
>
> As suggested by the reviewer, we conducted additional experiments with *10 randomly chosen seeds* and report the mean plus standard error over seeds, where we *changed the x-axis to the number of trajectories*. Specifically, we only used **$N=1$ roll-outs per policy iteration and $D=1$ trajectories per batch** to obtain preferences. In this setting, all algorithms **use a single-trajectory pair for comparison**, which is the best preference oracle for baseline algorithms like DPO, as suggested by the reviewer. However, the relative empirical performance ordering of algorithms is unchanged: for all environments with an unknown linear link function generating the preferences, **ZSPO has the best performance**. We want to emphasize that the empirical results are still consistent with our theoretical analysis, because the classic control and MuJoCo environments mostly have deterministic transitions with regular deterministic reward functions, resulting in a very mild preference-value distinguishability issue and low variance. This allows ZSPO to also learn from single-pair preference to infer the value difference well enough.

---

> ### Author Response · Authors · 2025-11-21
> **Response to Reviewer evWw (2/2)**
>
> # Ablation Studies
> As suggested by the reviewer, we conducted two ablation studies.
> 1. We **changed the true link function from a linear link function to the BT model** in the CartPole environment to investigate the performance of baseline algorithms without link function mis-specification, and the robustness of ZSPO. The results are shown in Fig.2(a) in the updated version, where we observe that most algorithms’ performance improved significantly compared to the case with a mis-specified link function, but ZSPO’s performance remains competitive. This demonstrates the importance of studying link-function mis-specification and the robustness of ZSPO. More studies are put in Appendix C.
> 2. We studied **ZSPO with a different number of roll-outs $N$ per iteration** to investigate how $N$ influences the variance and the convergence rate as proof-of-concept to our theoretical results. The results are shown in Fig.2(b), where we observe that as $N$ increases, ZSPO enjoys a faster convergence rate and an improved final performance. This coincides with our theory.
>
> # Other Comments
> We revised the paper to specify the linear link function in the experiment section and define the link function in the introduction as a function mapping reward difference to probability.
>
> ---
> We sincerely hope our rebuttal addresses the reviewer’s concerns, especially about using large batches of rollouts $D$ and $N$ for comparison, in the sense that it is **not a weakness of ZSPO, but a universal bottleneck** in the unknown link function with stochastic MDPs, which can be characterized by the preference distinguishability that affects all algorithms. ZSPO provides a theoretical guarantee for the worst-case RL instances, but in practice, when problem instances have nice structures, **we can use $D=N=1$ in ZSPO as our experiments suggest**.
>
> We would appreciate it if the reviewer could reevaluate our paper and the rating based on our rebuttal and the new empirical results.

---

> > ### Author Response · Authors · 2025-11-25
> > **Reminder**
> >
> > Dear Reviewer evWw,
> >
> > Thank you for your time reviewing our paper. We want to follow up to see whether our rebuttal has addressed your concerns. We would very much appreciate it if the reviewer would consider re-evaluating their score based on our response. Please don't hesitate to let us know if you have any other questions/comments. Thanks!
> >
> > best,
> >
> > Authors

---

### Author Response · Authors · 2025-11-21
**General Response to Reviewers**

We thank all the reviewers for their time reviewing our paper, and we address the common concerns. We have revised our manuscript accordingly.

# Empirical Experiments and Clarifications
As suggested by the reviewers, we conducted additional experiments with 10 randomly chosen seeds and report the training return curve and best policy performance as mean $\pm$ std in Fig.1 in the revision, plotted against the number of trajectories sampled. Specifically, we only used $N=1$ roll-outs per policy iteration and $D=1$ trajectories per batch to obtain preferences. Hence, each comparison is based on a **single pair of trajectories**, ensuring consistency across algorithms. The link function generating the preference is linear, and baseline algorithms assume a logistic link function internally, to compare the performances when there is a link function mis-specification. The complete results and discussion can be found in Section. 5 of our revised paper. We outline the average return of the learned policy as follows:

|Environments|CartPole|HalfCheetah|Hopper|
|-|-|-|-|
|ZSPO(Ours)|493|1484|515|
|ZPG|413|1068|457|
|RM+PPO|260|226|488|
|Online DPO|438|60|264|
|ES|210|778|403|

We observe that the relative empirical performance ordering of algorithms is unchanged compared to our initial submission. For all environments with a link function mismatch, **ZSPO converges even with small $N$ and $D$, and its average return performance is significantly better than all baselines**, demonstrating its robustness to the link function mis-specification.

# Ablations
We also conducted two additional ablation studies as suggested by the reviewers, shown in Fig.2 of our revised paper.
1. We tested all algorithms when the **true link function $\sigma$ is indeed logistic**, matching the BT model assumed by baselines in CartPole. The results are shown in Fig.2(a). We observe that baseline algorithms improved significantly when we removed the link function mismatch. Still, ZSPO demonstrated a competitive performance. This ablation study emphasizes the importance of recognizing the link function mis-specification, which could drastically hinder the performance of preference-based RL algorithms, and showcases the robustness of ZSPO.
2.  We studied the performance of ZSPO **with a different number of roll-outs $N$ per iteration**, to reduce the variance in value function sign estimation and improve the convergence rate. The results are shown in Fig.2(b). We observe that as $N$ increases, ZSPO enjoys a faster convergence rate and an improved final policy performance, which is consistent with our theoretical analysis.

# Practicality of Large Batch Roll-outs and Comparisons

We would like to point out that the theoretical guarantees are for the worst-case instance, applicable to all problems, while most instances in practice are not worst-case. In our updated experiments, ZSPO works well with small numbers of rollouts and batches, which shows its practicality.

Our updated experiments show **ZSPO works well with very small $N$ and $D$**. We intentionally used $N=1$ and $D=1$, and ZSPO still converges with the best performance under a link function mismatch, so neither large $N$ nor large $D$ is necessary in practice. We want to emphasize that the empirical finding is still consistent with our theory. The convergence rate of ZSPO in Theorem 1 depends on two additional terms: the distinguishability error and the approximation error. In the worst case, we would need $N$ and $D$ to be large enough for the concentration to occur, to reduce these two terms. But in practice, when environments like Gymnasium typically have deterministic transitions with regular deterministic reward functions, the preference-value distinguishability issue is much milder and the variance of return is smaller. This allows ZSPO to also learn from single-pair preference to infer the value difference well enough. For example, in our deterministic environment studies, a single trajectory is sufficient.

For stochastic MDPs, a large $D$ is necessary to ensure the distribution of return concentrates to exhibit some “regularity”, so that trajectory preference will be consistent with the policy preference with a small distinguishability $\varepsilon_D^*$. This is necessary for any algorithm to learn the optimal policy with an unknown link function. We discussed this unique challenge in detail in Appendix D.2. On the other hand, a large $N$ for the number of rollouts helps with variance reduction, which is sometimes required when the environmental return is noisy. Therefore, **large $N$ and $D$ are both necessary for the convergence in RL problems with higher stochasticity and noisy observation, independent of the used algorithm**. In practice, the numbers depend on the hardness of the problem and could be much reduced. For example, $N=D=1$ is enough for our Gym experiments.

---

We hope our response can address the reviewers' concerns.

---

### Author Response · Authors · 2025-12-01
**Summary of Reviewer Comments and Our Rebuttal (1/2)**

Dear Area Chair,

Thank you for your time and effort in overseeing the review process for our submission. Below, we summarize the reviewers’ main concerns and how our rebuttal and revision have addressed their comments/questions. Overall, **the reviewers acknowledged the paper’s theoretical contribution**: providing a provable convergence framework for policy optimization from trajectory preferences under an unknown link function and going beyond the standard Bradley–Terry (BT) assumption. They also noted that the empirical results are promising. The main concerns can be grouped into three points as follows.
1. The practicality of ZSPO given the theoretical dependence on the large comparison batch size $D$ and the majority vote size $N$,
2. The novelty relative to prior work, i.e., ZPG,
3. The clarity and fairness of the experimental evaluation.

In the rebuttal, we addressed these issues via **additional experiments, explicit ablations, and clearer separation and clarification between worst-case theoretical conditions and typical empirical requirements**. We hope this summary helps inform your assessment.

# Practicality of Large Preference Comparison Batch Size
Reviewers (evWw, C5xV, 2LSp, HMb6) raised concerns that ZSPO’s use of one-bit feedback and its theoretical requirement for large $N$ and $D$ could limit practical applicability. We clarified that large $N$ and $D$ arise as sufficient conditions needed to establish convergence guarantees for worst-case stochastic MDP instances under an unknown link function. We further discussed that this issue reflects a broader, algorithm-dependent bottleneck for link-function-agnostic preference-based RL in stochastic environments, and is not specific to ZSPO.

Importantly, we emphasized that **large $N$ and $D$ are NOT necessary in typical practical settings**. Empirically, we revised all Gymnasium experiments to use single-trajectory, single-comparison feedback, i.e., $N=D=1$. ZSPO still matches or exceeds baseline performance under link-function mismatch (Fig. 1). We also clarified an implementation detail relevant to stability: ZSPO is fully compatible with modern optimizers, and all experiments used AdamW, which yields stable learning curves comparable to baselines. The details are added to Appendix C.

# Experimental Fairness and Ablation Studies
Reviewers (evWw, C5xV, 2LSp) requested additional experiments to support the results: more random seeds, confirmation that all methods share the same preference oracle, and ablations on $N$ and link-function misspecification. In response, we re-ran the experiments with 10 seeds with $N=D=1$ and confirmed that in this setting, all baselines used the same preference oracle with a linear link function. Our results demonstrated that **ZSPO still outperformed baselines** when there is a link-function mis-specification. We also conducted ablation studies when the link function is correctly specified, and ZSPO remained competitive compared to baselines that explicitly use the link function. We then studied how the batch size $N$ influences the performance and showed that ZSPO could converge with $N=1$, but increasing $N$ improved the convergence speed and stability, consistent with our theoretical characterization. We revised our section 5 based on the new results.

# Motivation Beyond the Bradley-Terry Model
Reviewers (C5xV, HMb6) asked for a further justification for considering non-BT unknown link functions. We explained the motivation to realistic settings with heterogeneous or uncalibrated human and LLM raters, whose effective link functions are mixtures of various monotonic link functions, and thus could be unknown and outside the BT class. We also noted that alternative link functions (e.g., probit, linear) **correspond to standard random-utility models** under different noise assumptions (Gaussian vs. uniform), supporting the theoretical generality of the setting. In the experiments, we used the linear link function to stress-test BT-based methods under a qualitatively link-function mis-specification, which showed that BT-dependent baselines can suffer a substantial performance drop, whereas ZSPO remains robust.

---

> ### Author Response · Authors · 2025-12-01
> **Summary of Reviewer Comments and Our Rebuttal (2/2)**
>
> # Novelty Relative to Prior ZPG Framework
> Reviewer 2LSp suggested ZSPO might be an incremental adaptation of the previously published ZPG method. We respectfully clarified that this comment is **NOT an accurate assessment**, and highlighted four substantial distinctions:
> 1. Problem setting: ZSPO operates under an unknown link function, whereas ZPG requires a known one;
> 2. Estimator: ZSPO uses only the sign of value-function differences, rather than the magnitude in ZPG;
> 3. New proof framework: we designed a new Lyapunov-drift analysis specific to sign-based updates, replacing ZPG’s magnitude-based gradient estimation.
> 4. New theoretical characterization: we studied the preference-distinguishability constant $\varepsilon_D^*$ unique to the unknown link function setting and quantified how it influences the convergence.
>
> Collectively, these constitute a distinct conceptual problem formulation and original analysis framework beyond ZPG.
>
> # Robustness to Link-Function Mis-Specification
> Reviewer 2LSp sought a clearer explanation of why ZSPO remains stable under a link-function mis-specification. We emphasized that the sign estimator in ZSPO's update is **invariant under a monotonic transformation of the true link function**, making it agnostic to the link function's shape. This mechanism is captured by our theory and validated by the new ablations, i.e., ZSPO outperforms baselines with a linear-link-function oracles (mis-specification), and remains competitive with a BT oracle (no mis-specification).
>
> # Scalability and Global Optimality
> Reviewer HMb6 asked about the potential for ZSPO to apply in high-dimensional problems, such as LLMs, and whether it is possible to show global convergence. We clarified that ZSPO serves as a **conceptual and theoretical foundation**. The scalability to large-scale models could be achieved through integration with existing efficient zeroth-order techniques (e.g., MeZO) and parameter-efficient fine-tuning methods (LoRA/PEFT). These extensions are outlined explicitly as future work and discussed in Appendix D.3. Regarding optimality, we clarified that our paper focused on convergence to the stationary point due to the non-convex nature of RL problems with general policy parameterization. To establish convergence to the global optimizer, one would **require additional assumptions**, such as convexity or the PL condition over the policy parameterization.
>
> ---
> Sincerely,
>
> Authors of Submission 15628

---

### Meta-Review · Area_Chair_XHCy · 2026-01-07

**Summary:**

This paper studies PbRL when the link function is unknown, rather than assuming a Bradley–Terry / logistic model.
The paper proposes ZSPO, a zeroth-order policy optimization method that avoids estimating reward differences from preferences and instead tries to infer only the sign of the value difference between the current policy and a randomly perturbed policy. Concretely, each iteration perturbs parameters, collects paired trajectory batches under the two policies, uses preference feedback plus a majority vote to estimate the sign of improvement, and then updates the policy in a direction intended to be positively correlated with the true policy gradient.

On the theory side, the analysis establishes convergence to a stationary point in terms of expected gradient norm under smoothness and “distinguishability” assumptions, with rates that depend on the dimension, horizon, the number of comparisons, and a distinguishability quantity. Empirically, the paper evaluates on a small set of control benchmarks with synthetic preference oracles, showing the performances when the true preference model differs from the logistic model assumed by several baselines.

**Reviewer Concerns:**

Multiple reviewers were concerned that the algorithm and/or the theory implies large numbers of rollouts and preference queries, trajectory batches and majority votes, per update, which is likely to be hard to reconcile with real human-in-the-loop settings and potentially expensive even with synthetic or LLM judges. While the rebuttal argues the large-N/D requirements are worst-case and provides updated experiments with small N/D, the gap between the theory’s sufficient conditions and realistic deployment remains a central unresolved issue.

One reviewer viewed the theoretical framework in the paper as a relatively immediate adaptation of prior ZPG-style analysis, with the main change being “sign” rather than “magnitude” estimation. The authors argue this induces genuinely different estimators and proof structure, but from the reviews there is still not strong consensus that the conceptual advance is large relative to the closest predecessor.

Reviewers noted initially unclear seed counts, noisy plots, limited task coverage (only a few benchmarks), etc. The rebuttal adds seeds and ablations and clarifies parts of the setup, which helps, but the overall empirical story is still narrow and does not convincingly establish where this method is competitive beyond small benchmark settings.

The paper discusses potential extensions toward high-dimensional settings (e.g., large policies/LLMs) but provides no direct evidence. Given the zeroth-order nature and the cost of preference queries, reviewers appear to be unconvinced the approach is close to practical at scale.

**Reviewer Scores:**

Reviewer evWw: Likely small upward adjustment (or no change) (4 => 4/5) because several concrete experimental concerns (seed count, x-axis clarity, some ablations, and protocol clarification) were addressed. The main practicality critique would likely remain.

Reviewer C5xV: I would expect no change (4 =>  4). The rebuttal helps on stability/implementation details and clarifies aspects of preference collection, but the reviewer’s broader concerns about real-world applicability ,i.e., synthetic oracle only, missing more direct baselines like Pebble-style comparisons likely persist.

Reviewer 2LSp: Most likely no change (4 => 4). The rebuttal responds directly to novelty and fairness points, but whether it convincingly resolves the minor adaptation of ZPG perception is unclear, and that was a central driver of their score.

Reviewer HMb6: Likely no change (6 => 6). The rebuttal addresses motivation and includes an ablation with a logistic oracle, but scalability and practicality concerns would probably remain, and the reviewer was already in the "weak accept" region.

Overall, the AC believes this is a borderline paper.

---

### Decision · Program_Chairs · 2026-01-26

Reject